# Host control of persistent Epstein–Barr virus infection

Axel Schmidt[1✉], T. Madhusankha Alawathurage[1], Friederike S. David[1,2],
Yosuke Ogawa[3,4,5], Leonard Frach[1,6], Sylvia Richter[1], Merle Schaefer[1], Carina M. Mathey[1],
Sabrina K. Henne[1], Japan COVID-19 Task Force*, Andreas J. Forstner[1,7], Alexander T. Dilthey[8,9],
Anne-Katrin Pröbstel[10,11,12,13,14,15], Kaan Boztug[16,17,18,19], Markus M. Nöthen[1], Ho Namkoong[20],
Yukinori Okada[3,5,21,22,23], Eva C. Beins[1] & Kerstin U. Ludwig[1✉]

Epstein–Barr virus (EBV) infects approximately 90–95% of the global population[1,2] and persists in B cells as a lifelong infection[3]. Previous EBV infection is associated with autoimmune and neoplastic disease[4]. Still, the biological basis of host control during EBV persistence remains unclear. Here we report the identification of non-genetic and genetic factors that are associated with EBV control during persistent infection. Using blood-based genome sequence data from 486,315 UK Biobank and 336,123 All of Us participants, we identified short-read pairs mapping to the EBV genome in 16.2% and 21.8% of individuals, respectively. EBV read detection (EBVread⁺) reflects increased viral load in blood cells, as shown by orthogonal measurements, and was associated with HIV infection, immunosuppressive drug intake and current smoking. Genome-wide analyses of EBVread⁺ identified strong associations at the major histocompatibility complex (MHC), including 54 independent human leukocyte antigen (HLA) alleles of MHC classes I and II, and at 27 genomic regions outside MHC. Epistasis with distinct HLA alleles of MHC class I was observed at the *ERAP2* locus. Analysis of individuals with EBV-associated diseases[4] revealed a higher polygenic burden of EBVread⁺ for HLA alleles at MHC class I in multiple sclerosis (driven by HLA-A*02:01) and at MHC class II in rheumatoid arthritis. Phenome-wide analyses identified a polygenic overlap of EBVread⁺ with inflammatory bowel disease, hypothyroidism and type 1 diabetes. Our study establishes by-products of human genome sequencing as a surrogate marker of EBV viral load. This will facilitate investigation and treatment for EBV and other persistent viral infections.

EBV (human herpesvirus 4) is a DNA virus that infects approximately 90–95% of the global population[1,2]. Primary EBV infection usually occurs in childhood and remains asymptomatic or mild. From adolescence onwards, it can cause infectious mononucleosis[5]. EBV enters the host via the oropharyngeal epithelium and infects naive B cells. These differentiate into long-lived memory B cells that become part of the circulation, thereby establishing persistent infection[3,6]. Occasionally, EBV-infected memory B cells reactivate to produce new infectious virions[7].

EBV infection is a risk factor for various neoplasms (for example, Hodgkin and non-Hodgkin lymphoma and multiple sclerosis)[4,8,9]. Although EBV seropositivity is a prerequisite for multiple sclerosis[10], only some individuals infected with EBV develop the disease, following a prodromal phase[11]. Furthermore, although multiple sclerosis risk is significantly elevated post-infectious mononucleosis, many patients with multiple sclerosis did not have a severe primary EBV infection[12]. Thus, multiple sclerosis may arise secondary to inefficient EBV

[1]Institute of Human Genetics, School of Medicine, University of Bonn and University Hospital Bonn, Bonn, Germany. [2]Department of Psychiatry and Psychotherapy, University of Marburg, Marburg, Germany. [3]Department of Genome Informatics, Graduate School of Medicine, The University of Tokyo, Tokyo, Japan. [4]Department of Pediatrics, Graduate School of Medicine, The University of Tokyo, Tokyo, Japan. [5]Laboratory for Systems Genetics, RIKEN Center for Integrative Medical Sciences, Yokohama, Japan. [6]Department of Clinical, Educational and Health Psychology, Division of Psychology and Language Sciences, Faculty of Brain Sciences, University College London, London, UK. [7]Institute of Neuroscience and Medicine (INM-1), Research Center Jülich, Jülich, Germany. [8]Institute of Medical Microbiology and Hospital Hygiene, Heinrich Heine University Düsseldorf, Düsseldorf, Germany. [9]Center for Digital Medicine, Heinrich Heine University Düsseldorf, Düsseldorf, Germany. [10]Center of Neurology, Department of Neuroimmunology, University Hospital and University Bonn, Bonn, Germany. [11]German Center for Neurodegenerative Diseases (DZNE), Bonn, Germany. [12]Department of Neurology, University Hospital of Basel and University of Basel, Basel, Switzerland. [13]Department of Biomedicine, University Hospital of Basel and University of Basel, Basel, Switzerland. [14]Department of Clinical Research, University Hospital of Basel and University of Basel, Basel, Switzerland. [15]Research Center for Clinical Neuroimmunology and Neuroscience Basel, University Hospital of Basel and University of Basel, Basel, Switzerland. [16]Clinic for Pediatric Immunology and Rheumatology, Center for Pediatrics and Adolescent Medicine, University Hospital Bonn, Bonn, Germany. [17]St. Anna Children's Cancer Research Institute, Vienna, Austria. [18]CeMM Research Center for Molecular Medicine of the Austrian Academy of Sciences, Vienna, Austria. [19]Department of Pediatrics and Adolescent Medicine, Medical University of Vienna, Vienna, Austria. [20]Department of Infectious Diseases, Keio University School of Medicine, Tokyo, Japan. [21]Department of Statistical Genetics, Graduate School of Medicine, The University of Osaka, Suita, Japan. [22]Laboratory of Statistical Immunology, Immunology Frontier Research Center (WPI-IFReC), The University of Osaka, Suita, Japan. [23]Premium Research Institute for Human Metaverse Medicine (WPI-PRIMe), The University of Osaka, Suita, Japan. *A list of authors and their affiliations appears online. ✉e-mail: axel.schmidt@ukbonn.de; kerstin.ludwig@uni-bonn.de

immune control during the prodromal phase, as indicated by high EBV viral load[11]. Similar mechanisms might be implicated in other EBV-associated autoimmune disorders, as suggested by elevated EBV viral loads in systemic lupus erythematosus[13] and rheumatoid arthritis[14]. In EBV-associated cancers, the importance of proper EBV immune control has been demonstrated by studies of inborn errors of immunity (IEIs): patients with IEIs involving impaired T and natural killer (NK) cell cytotoxicity have elevated EBV viral loads in blood[15], and an increased risk for B cell-derived EBV-positive lymphomas[16]. Individuals with human immunodeficiency virus (HIV) or immunosuppression also show impaired EBV control[17] and an increased incidence of EBV-positive lymphomas[18,19]. Still, despite its presumed clinical relevance, data on immune control during persistent EBV infection are limited.

Research into the biological basis of immune control of persistent EBV infection is hampered by a lack of direct measurements of EBV viral load in large immunocompetent cohorts, and limited knowledge regarding the role of serological factors in the control of EBV[20].

To address this, we exploited the fact that EBV DNA in memory B cells is sequenced as a by-product of genome sequencing (GS) of human peripheral blood[21]. Using blood-based GS data from the UK Biobank (UKB)[22] and All of Us (AoU)[23] together with orthogonal data, we demonstrated that short-read pairs mapping to the EBV genome (EBV reads) in GS data are a surrogate measure for increased EBV viral load. EBV read prevalence was increased in immunosuppressed individuals; in current smokers; and in samples obtained in winter. Strong genetic associations were found for the MHC locus and 27 loci outside MHC, which were broadly consistent across the two biobanks. Downstream analyses suggested candidate genes, and highlighted pathways and cell types relevant for EBV immunity. Investigations of EBV-associated diseases generated novel hypotheses regarding mechanisms in multiple sclerosis and rheumatoid arthritis, and phenome-wide analyses identified novel diseases for which host control of EBV viral load might be pathophysiologically relevant.

## EBV reads are present in GS data from biobanks

We retrieved EBV reads from the GS data of 490,293 UKB participants[24] (Methods; Fig. 1a and Supplementary Notes 1 and 2). During quality control (QC), 51 library-preparation plates showed evidence of contamination and were excluded (Methods; Extended Data Fig. 1 and Supplementary Fig. 1). Aggregated EBV reads of the remaining 486,315 individuals (UKB QC cohort) were evenly distributed across the EBV genome (Fig. 1b). EBV read distribution was zero inflated, that is, no EBV reads were observed in $n = 407,544$ individuals (83.8%, denoted as 'EBVread⁻'; Fig. 1c). Of the 78,771 individuals with detected EBV reads ('EBVread⁺', 16.2%), 61.9% had EBV read count = 1. Further analysis of coverage and sequence data (Methods) confirmed that EBV reads from this group reflect true signals (Extended Data Fig. 2 and Supplementary Table 1).

EBV reads were also extracted from the blood-based GS data of 336,123 ethnically diverse individuals from AoU[23] (AoU QC cohort; Methods). EBV read distribution was similar to that in UKB, but a lower fraction of individuals had EBV read count = 1 ($n = 37,901$ out of 73,137 EBVread⁺ individuals, 51.8%; Extended Data Fig. 3). Overall, 21.8% of the AoU QC cohort were EBVread⁺, although this varied across ancestries (Supplementary Table 2). For European (EUR) cohorts, the fraction of EBVread⁺ individuals was comparable in AoU (17.6%) and UKB (15.8%; UKB EUR cohort; Fig. 1a; Methods). Whether the residual difference is due to ancestry-specific mechanisms of EBV control or characteristics such as a higher average GS coverage in AoU (Supplementary Table 2) awaits elucidation. In our data, the EBVread⁺ fraction is higher than in smaller GS (14.0%)[21] or diagnostic quantitative PCR (qPCR; 11.03%)[18] studies of immunocompetent individuals. This might be attributable to differences in cohort composition and/or strict cut-offs used in clinical settings.

## EBVread⁺ status reflects increased EBV viral load in blood cells

We then assessed the relevance of GS-based EBVread⁺ to EBV biology. First, we investigated how well EBVread⁺ matches EBV seropositivity. In a UKB subcohort with available serology data (UKB serology cohort; $n = 9,281$), 491 individuals were EBVsero⁻ and 8,790 EBVsero⁺, based on previous definitions[1]. EBV reads were observed in 0.61% of EBVsero⁻ and 16.38% of EBVsero⁺ individuals (sensitivity of 16.4% and specificity of 99.4%; Fig. 1d). Second, we investigated whether EBV read detection reflects high viral load in blood cells. Therefore, we (1) simulated GS and compared modelled versus observed outcomes; (2) measured viral load via qPCR in samples from two small, independent cohorts with GS data[25,26]; and (3) correlated EBV read counts with EBV gene expression from blood-based RNA sequencing (RNA-seq; Japan COVID-19 Task Force[26] (JCTF); Methods).

The simulation reproduced the EBV read distribution observed in UKB or AoU, including the zero inflation (Extended Data Fig. 4), and was compatible with an underlying log-normal distribution, as reported for HIV-1 viral load[27]. In the qPCR analysis, EBV read counts showed a positive correlation with EBV DNA detection, and a negative correlation with Cp (crossing point) values (Fig. 1e,f and Extended Data Fig. 2). In 1,010 individuals from the JCTF, the fraction of individuals with detected EBV transcripts was higher among EBVread⁺ than among EBVread⁻ samples (Fig. 1g). Together, this provides evidence that EBVread⁺ represents an approximation of elevated EBV viral load within human blood cells.

## EBVread⁺ is associated with decreased EBV control during persistence

To determine which phase of the EBV life cycle is reflected by EBVread⁺, we investigated correlations between EBV read counts and (1) individual EBV transcript counts from the JCTF cohort; and (2) four individual EBV antibody levels (EA-D, EBNA-1, ZEBRA and VCA-p18, all IgG; median fluorescence intensity (MFI) values)[1]. In step one, the strongest EBVread⁺ correlations were with transcripts of *A73* ($\rho = 0.47$, Spearman's rank correlation), *BARFO* ($\rho = 0.42$) and *RPMS1* ($\rho = 0.43$; Fig. 1g). All three belong to the *BART* gene cluster that is associated with latency[4]. We also observed correlations with transcripts of some lytic genes, particularly from the same genomic region.

Step two was performed in 7,338 EUR EBVsero⁺ individuals from the UKB serology cohort, with presumed persistent (not primary) EBV infection given their age at recruitment (Methods). The strongest correlation was observed with IgG levels to VCA-p18 ($\rho = 0.12$, $P < 2.2 \times 10^{-16}$), followed by IgG levels to ZEBRA and EBNA-1 (Extended Data Fig. 5). Although VCA-p18 is a lytic-phase antigen, IgG to VCA-p18 is detectable during persistent EBV infection[28] and increased titres are found in individuals with high EBV viral load in blood[29,30]. Thus, higher viral load in blood cells, as measured by EBVread⁺, might correlate with ongoing lytic activity. This aligns with the 'germinal centre model of EBV persistence'[7], in which the latently infected memory B cell pool in blood is maintained in equilibrium by lytic reactivation events in lymphoid tissues (Extended Data Fig. 2). However, our data suggest an extension to this model, as some reactivation might occur within blood, as recently also demonstrated in individuals with systemic lupus erythematosus[31].

## Non-genetic factors and sex contribute to EBVread⁺

Next, we investigated the influence of non-genetic factors and sex on EBVread⁺, with the aim to (1) identify those factors; (2) enable exclusion from further analysis of all individuals whose EBV read count was probably determined exogenously; and (3) control for these factors in subsequent analyses. Whenever possible, we minimized overfitting

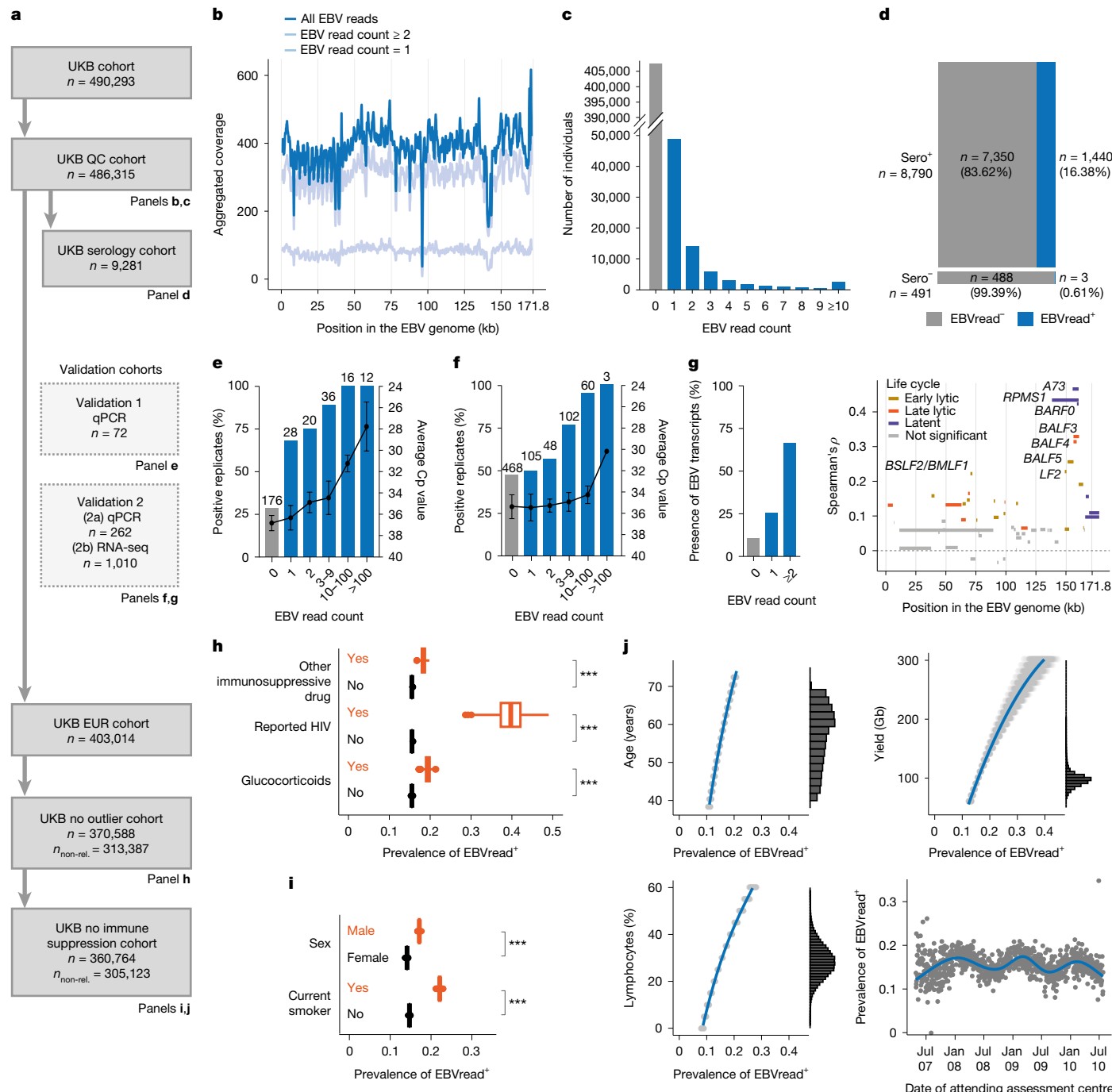

**Fig. 1 | Analysis of EBV reads in blood-based GS data. a**, Flowchart of UKB cohort definitions and respective sizes, created by consecutive steps. Technical validation of EBV reads was performed by qPCR in two independent cohorts. non-rel., non-related. **b**, Cumulative read coverage across the EBV genome in the UKB QC cohort (line smoothed, 500-bp rolling window), for all individuals (dark blue) and split by EBV read count group (light blue). **c**, Number of individuals within EBV read count groups (maximum of 27,639 reads) in the UKB QC cohort. **d**, In a subcohort with available EBV serology data (UKB serology cohort), detection of EBV reads (EBVread⁺) was highly specific for being EBV seropositive (sero⁺). **e,f**, qPCR validated (validation 1 (**e**) and validation 2 (**f**)) EBV reads as a measure of EBV viral load, based on increasing fraction of positive replicates (bars, left y axis) and decreasing average crossing point (Cp) value for positive replicates (points, right y axis, inversely scaled; data are presented as mean, with the error bars denoting standard deviation). The number of replicates is given above the bars. **g**, Paired GS–RNA-seq data of 1,010 samples showed positive association between the presence of EBV transcripts and EBV read

counts (validation 2; left), largely driven by expression of genes from the BART gene cluster (right; colours according to EBV stage in which the gene is primarily expressed). The dotted line indicates a Spearman's ρ of 0. **h**, In the UKB no outlier cohort (non-related), immune-modulating factors significantly increased the prevalence of EBVread⁺ individuals. **i,j**, Following the exclusion of immunocompromised individuals (UKB no immune supp. cohort, non-related), male sex and current smoking status were both associated with EBVread⁺ (**i**), as were older age, lymphocyte percentage, sequencing yield and winter sampling (**j**). Estimates and corresponding distributions were obtained using marginalization and bootstrapping (n = 1,000), except for sampling day, where raw EBVread⁺ prevalences were taken for analysis. Estimated distributions are shown as boxplots (**h,i**) or individual data points (**j**, grey). The boxplots show the median (thick line), 25th and 75th percentiles (box), largest–smallest values no further from the box than 1.5 times the interquartile range (whiskers) and outliers (points; **h,i**). ***Consistent across 1,000 bootstrap replicates.

by using one biobank for discovery and the other for replication (Supplementary Note 1).

First, we assessed 11,111 SNOMED concept IDs and their association with EBVread[+] in the AoU QC cohort (Methods). Initial test statistics were highly inflated, with HIV positivity and smoking showing the strongest associations (Supplementary Table 3 and Supplementary Fig. 2). When the analysis was conditioned on these two traits, inflation was largely resolved, although some residual associations with several immune-related SNOMED concepts remained (Supplementary Note 3).

To quantify the effect of HIV or immunosuppression on EBVread[+] and identify additional contributors, we investigated non-related individuals of EUR ancestry. Individuals with outlier blood count measurements and those in the top EBV read count percentile were excluded, given the high prevalence of pathophysiological processes in this group, which probably drive EBV abundance (Supplementary Fig. 3 and Supplementary Note 4). In this UKB no outlier cohort, 48,771 of 313,387 individuals were EBVread[+] (that is, 15.6%; with an expected standard deviation (s.d.) of 0.1% based on bootstrapping). HIV infection and immune-modulatory drugs significantly increased the likelihood of EBVread[+]. The highest probability was for reported HIV infection (39.7%, s.d. = 3.5%;), followed by intake of glucocorticoids (19.4%, s.d. = 0.7%) or other immunosuppressive drugs (18.3%, s.d. = 0.5%).

We then excluded from the UKB no outlier cohort individuals with reported HIV infection, or current use of glucocorticoids or other immunosuppressive drugs ('UKB no immune supp. cohort'; Fig. 1a), and performed variable selection on a set of predefined covariates to identify further contributing factors in immunocompetent individuals (non-related individuals; 47,234 EBVread[+] and 257,899 EBVread[−]; Methods; Supplementary Table 4). EBV reads were more frequent in male individuals than in female individuals (17.1%, s.d. = 0.1% versus 14.1%, s.d. = 0.1%) and in current smokers than in current non-smokers (22.1%, s.d. = 0.3% versus 14.7%, s.d. = 0.1%; Fig. 1i). Former smoking status alone was not identified as a relevant predictor of EBVread[+]. Other selected variables were increasing age, GS yield and lymphocyte percentage, all of which were positively correlated with EBVread[+] (Fig. 1j). EBV read detection was also more probable in samples collected in winter (Fig. 1j). This seasonality effect was confirmed in AoU (Extended Data Fig. 3) and requires further investigation. A plausible hypothesis is that seasonal infections during winter, such as co-infections with respiratory viruses, drive EBVread[+]. This would be consistent with observations of a higher prevalence of EBVread[+] in the JCTF, whose participants were infected with SARS-CoV-2 around the time of sampling (39.2% EBVread[+]; Supplementary Table 5, Supplementary Fig. 4 and Supplementary Note 5). Together, the identified factors might also contribute to cross-biobank and cross-ancestry differences in EBVread[+] prevalence.

## Common variants in and outside of MHC contribute to EBVread[+]

To identify associations between common genetic variants and EBVread[+], we performed a genome-wide association study (GWAS) using related individuals from the UKB no immune supp. cohort (Fig. 1a) and imputed data (Methods). Variants at 28 loci showed genome-wide significance (Fig. 2a), including a long-range association at the MHC locus and additional associations at 27 non-MHC loci (Methods; Table 1 and Supplementary Tables 6 and 7). The heritability estimate for EBVread[+] for all common variants outside the MHC region was 2.04% (standard error of the mean (s.e.m.) = 0.44%; linkage disequilibrium score regression[32]).

At the non-MHC loci, gene prioritization approaches (Methods) highlighted genes implicated in immune processes (for example, *ERAP2* and *EOMES*), known IEIs (for example, *CD70*, *IKZF3* and *CTLA4*) and genes of pharmacological relevance (for example, *SLAMF7*, inhibited by elotuzumab; Supplementary Table 8). Non-MHC lead variants

were also associated with a broad range of phenotypes in OpenTargets (Supplementary Table 9), although the extent varied across loci. While some loci showed high pleiotropy (more than 100 associated phenotypes, for example, loci including *SH2B3*, *PTPN22* and *IRF1*), other lead variants had only few associations at the same significance threshold, suggesting a more specific role in EBV control (for example, *ILDR1* and *CMC1*). Finemapping with SuSiE[33] identified potentially causative variants at four loci (Table 1 and Supplementary Table 10), including three missense variants with posterior inclusion probability (PIP) scores > 0.1, and one non-coding variant, rs531660643, at PIP > 0.95 (rs531660643). The latter is a splice quantitative trait locus (QTL) for *BCL3* (whole blood, GTEx v8), which is involved in B cell fate and NF-κB regulation[34].

At the MHC region, the immunologically relevant variants are alleles of HLA genes ('HLA alleles'), which determine the repertoire of antigens that can be presented to the immune system. On the basis of the imputed HLA alleles[22], 116 different classical HLA alleles were associated with EBVread[+] (Methods; Supplementary Table 7). The lowest *P* value was for the MHC class II (MHC-II) allele HLA-DRB1*04:04 (beta = 0.79, s.e.m. = 0.02), which is associated with increased rheumatoid arthritis risk[35]. The next most significant HLA alleles were HLA-A*02:01 (beta = −0.31, s.e.m. = 0.01), which decreases risk for multiple sclerosis[36], EBV[+] Hodgkin lymphoma[37] and endemic Burkitt lymphoma[38], and HLA-B*14:02 (beta = −0.68, s.e.m. = 0.02). After iterative conditional analyses, 54 independent alleles from MHC-I and MHC-II remained with genome-wide significance (Methods; Fig. 2b and Supplementary Table 7).

Given previous evidence for epistatic effects between HLA alleles and genes involved in antigen processing, for example, *ERAP2* (ref. 39) and *ERAP1* (ref. 40), we conducted an interaction analysis between the 54 conditionally independent HLA alleles and the top three non-MHC loci (Methods). After correction for multiple testing, three significant interactions were identified between the *ERAP2* lead variant rs2548225 and HLA alleles of MHC-I (that is, HLA-A*02:01, HLA-B*40:01 and HLA-B*15:01; Fig. 2c and Supplementary Table 11). This is functionally plausible, as *ERAP2* encodes an aminopeptidase that trims peptides within the endoplasmic reticulum before loading onto MHC-I[41]. The rs2548225 risk allele tags *ERAP2* haplotypes that are characterized by splice variants, which render *ERAP2* mRNA non-functional[41].

Finally, we aimed to replicate the UKB-based EBVread[+] GWAS results in 184,948 individuals of EUR ancestry from the AoU no outlier cohort. Of the 116 associated HLA alleles, 106 were matched to HLA alleles in AoU (Methods). Of these, 100 showed *P* < 0.05 and a consistent effect direction in both datasets (Supplementary Table 7). For the 54 conditionally independent HLA alleles, 46 of the 52 that were available in AoU were replicated, as were lead variants at 25 of the 27 non-MHC loci (at *P* < 0.05; Supplementary Table 6). No meta-analysis was performed due to missing or different covariates, for example, the lack of blood count data in AoU (Supplementary Note 4).

## Associated GWAS loci for EBVread[+] are specific for increased EBV viral load

To explore whether the identified loci are specific for EBV viral load, we compared effect sizes of lead variants from the EBVread[+] GWAS to GWAS data for memory B cell abundance[42] and human herpesvirus 7 (HHV7). For memory B cell abundance, no significant Spearman's correlation was observed for non-MHC loci (Extended Data Fig. 6; no MHC data provided). However, a genome-wide significant association was observed for the EBVread[+] lead variant at the 13q33.3 locus comprising *TNFSF13B*, which is implicated in memory B cell survival[43] (Supplementary Table 12). For HHV7, we extracted reads from UKB and calculated effect sizes as for EBV (Methods; Supplementary Fig. 5 and Supplementary Note 6). No significant Spearman's correlations were found for the EBVread[+] non-MHC loci or HLA alleles (Fig. 2d),

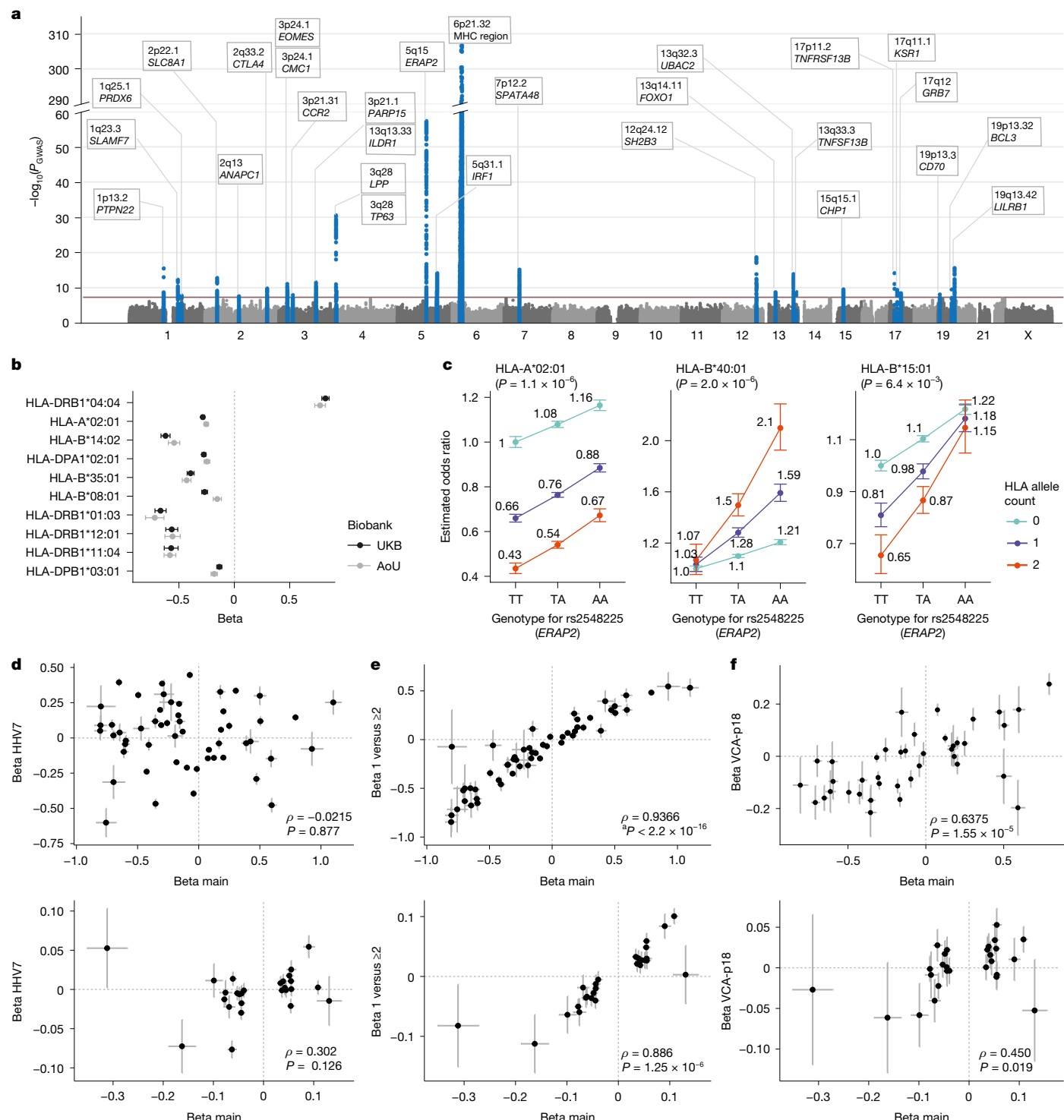

**Fig. 2 | Genetic analyses of EBVread+. a**, Manhattan plot for GWAS on EBVread+ (statistical test: regenie single-variant association testing, adjusted for covariates; see Methods) from the UKB no immune supp. cohort ($n_{(EBVread^+)}$ = 56,180 versus $n_{(EBVread^-)}$ = 304,103). Variants at genome-wide significant loci ($P_{uncorrected}$ < 5 × 10⁻⁸, red line) are highlighted in blue. Each locus is annotated with a chromosomal band and the closest gene. **b**, Forest plot for the top 10 conditionally independent HLA alleles (UKB no immune supp. cohort n = 360,764 and AoU no outlier (EUR) n = 184,948). The points reflect effect sizes calculated with regenie as described in panel **a**, from unconditioned analyses (estimated beta values; error bars represent 95% confidence intervals (unadjusted)). **c**, The top three most significant epistatic interactions, all between the *ERAP2* lead variant and three HLA alleles from MHC-I, from the UKB no immune supp. cohort

non-related subset (n = 304,523). Odds ratios and 95% confidence intervals (unadjusted) are based on the fit of the interaction models (Supplementary Note 11). Note that exact sample numbers for the UKB no immune supp. cohort used in **a–c** vary due to respective missing data. **d–f**, Effect sizes from 54 conditionally independent HLA alleles (top panels) and lead variants at 27 non-MHC loci (bottom panels) from the EBVread+ GWAS (beta main) were plotted against the same measures of additional phenotypes: HHV7read+ (**d**), EBV read = 1 versus EBV read ≥ 2 (**e**), and VCA-p18 IgG levels (**f**; data taken from ref. 44). Spearman correlation coefficients (ρ) and respective two-sided P values (P) are provided. ᵃExact P value is not available due to computational limits. Data are presented as betas and standard error. Sample sizes used to calculate correlation of effect sizes are given in Supplementary Table 12.

**Table 1 | Overview of 27 non-MHC loci associated with EBVread⁺ in UKB**

| Locus | Lead variant for EBVread⁺ (effect allele) | $P$ GWAS EBVread⁺ | Beta±s.e. | Candidate genes[a] | Potentially functionally relevant variant[b] |
|---|---|---|---|---|---|
| 1p13.2 | rs2476601 (A) | $3.30×10^{-16}$ | 0.090±0.011 | *PTPN22*, *PHFT1*, *DCLRE1B* and *AP4B1* | *PTPN22*, p.Trp620Arg |
| 1q23.3 | rs3766370 (C) | $5.00×10^{-13}$ | 0.051±0.007 | *SLAMF7* and *LY9* | – |
| –1q25.1 | rs1539255 (T) | $1.78×10^{-8}$ | −0.039±0.007 | *PRDX6* | – |
| 2p22.1 | rs62149448 (T) | $1.69×10^{-13}$ | −0.063±0.009 | *SLC8A1* | – |
| 2q13 | rs1345202 (T) | $2.52×10^{-8}$ | −0.044±0.008 | (*ANAPC1*) | – |
| 2q33.2 | rs231775 (A) | $1.87×10^{-10}$ | −0.045±0.007 | *CTLA4* | – |
| 3p24.1_EOMES | rs1491190814 (ATT) | $8.19×10^{-12}$ | −0.047±0.007 | *EOMES* | – |
| 3p24.1_CMC1 | rs74533039 (C) | $4.01×10^{-10}$ | 0.055±0.009 | *CMC1* and *AZI2* | – |
| 3p21.31 | rs1473413616 (CA) | $1.23×10^{-8}$ | −0.041±0.007 | *CCR2*, *CCR3* and *CCR5* | – |
| 3q13.33 | rs9828869 (T) | $1.30×10^{-8}$ | 0.042±0.008 | *ILDR1* | – |
| 3q21.1 | rs1106346 (A) | $3.20×10^{-12}$ | −0.048±0.007 | *PARP14* and *PARP15* | – |
| 3q28_LPP | rs13098877 (C) | $2.94×10^{-31}$ | −0.078±0.007 | *LPP* | – |
| 3q28*_TP63 | rs16864734 (G) | $8.67×10^{-10}$ | −0.069±0.012 | *TP63* | – |
| 5q15 | rs2548225 (A) | $5.20×10^{-58}$ | 0.109±0.007 | *ERAP1*, *ERAP2* and *LNPEP* | – |
| 5q31.1 | rs766751473 (TGTGATACCCCAA) | $7.27×10^{-15}$ | −0.053±0.007 | *P4HA2*, *IRF1*, *SLC22A4*, *SLC22A5*, *RAD50* and *PDLIM4* | – |
| 7p12.2 | rs1379182 (T) | $6.84×10^{-16}$ | 0.055±0.007 | *ZPBP* and *SPATA48* | – |
| 12q24.12 | rs7310615 (C) | $2.16×10^{-19}$ | −0.061±0.007 | *SH2B3*, *PHETA1* and *ALDH2* | *SH2B3*, p.Trp262Arg |
| 13q14.11* | rs75289402 (T) | $1.84×10^{-9}$ | 0.042±0.007 | (*FOXO1*) | – |
| 13q32.3 | rs701537 (A) | $1.32×10^{-14}$ | 0.054±0.007 | *UBAC2*, *GPR18* and *GPR183* | – |
| 13q33.3 | rs150861794 (C) | $1.55×10^{-9}$ | −0.163±0.027 | (*TNFSF13B*) | – |
| 15q15.1 | rs796756304 (C) | $2.79×10^{-10}$ | 0.051±0.008 | *NUSAP1* | – |
| 17p11.2 | rs34557412 (A) | $7.29×10^{-15}$ | −0.312±0.040 | *TNFRSF13B* | *TNFRSF13B*, p.Cys104Arg |
| 17q11.1 | rs884186 (A) | $4.09×10^{-10}$ | −0.076±0.012 | *KSR1* | – |
| 17q12 | rs9910678 (T) | $3.50×10^{-9}$ | −0.099±0.017 | *GRB7*, *GSDMB*, *ORMDL3* and *IKZF3* | – |
| 19p13.3 | rs344585 (C) | $8.08×10^{-9}$ | 0.039±0.007 | *CD70* | – |
| 19q13.32 | rs531660643 (G) | $3.08×10^{-10}$ | 0.143±0.023 | *BCL3* | rs531660643 (splice QTL) |
| 19q13.42 | rs111711612 (C) | $2.56×10^{-16}$ | 0.055±0.007 | *LILRB1* | – |

*Failed replication in AoU.

[a]Genes are listed if they were identified by two out of four different gene prioritization approaches (see Supplementary Table 13). If no gene was prioritized, the gene closest to the lead variant is listed in brackets. Underlined genes are the effector gene for lead variants (or variants with $r^2 > 0.7$) in single-cell eQTL data from PBMCs (OneK1K). [b]For missense variants with PIP > 0.1, or non-coding variants with PIP > 0.95.

although six of the non-MHC loci had $P < 0.05$ and a consistent direction of effect. For two of these (*SLC8A1* and *PTPN22*), colocalization analyses indicated shared causal variants (posterior probability (H4) > 0.5; Supplementary Table 13).

We then created a case–control definition in UKB that captures viral load rather than viral susceptibility, by excluding individuals with EBV read count = 0 (that is, almost all seronegative individuals). Effect sizes from an analysis of EBV read count = 1 versus EBV read count ≥ 2 were highly correlated with those of our main GWAS (non-MHC loci: Spearman's $\rho = 0.93$, $P = 6.2 × 10^{-7}$; HLA alleles: $\rho = 0.94$, $P < 2.2 × 10^{-16}$; Fig. 2e). Similar results were obtained in other case–control definitions within EBVread⁺ individuals and in additional comparisons of (1) EBV read count = 0 versus EBV read count = 1, and (2) female and male participants (Extended Data Fig. 6 and Supplementary Table 12).

Finally, we analysed GWAS summary statistics of four EBV antibody levels[44]. Consistent with the aforementioned correlation of EBVread counts with IgG antibody levels, effect sizes of lead variants for EBVread⁺ and VCA-p18 IgG levels were strongly correlated, particularly for the HLA alleles ($\rho = 0.64$, $P = 1.55 × 10^{-5}$; Fig. 2f). These findings suggest that the genetic associations with EBVread⁺ reflect specific EBV viral load-associated factors.

## Gene-based analyses suggest an enrichment of IEI genes

We then performed gene-based analyses to capture additional biology and enable systematic downstream analyses, using EBVread⁺ summary statistics for common variants and exome sequencing data for rare variants.

Common variants were assigned to individual genes, and gene-based $P$ values were calculated (MAGMA[45]; see Methods; without MHC region). Of 63 genes that remained significant after Bonferroni correction (Supplementary Table 14), ten were located outside of genome-wide significant loci and thus represent additional candidate genes. Nine of the 63 genes were IEI genes, including four (*IKZF3*, *NFKB1*, *CTLA4* and *CD70*) that predispose to severe clinical phenotypes post-EBV infection, including persistent EBV viraemia, EBV-associated lymphoproliferation and/or EBV-driven lymphoma (Fig. 3a and Supplementary Note 7). Formal testing using MAGMA gene set enrichment (Methods) showed that IEI genes ($n = 456$) were strongly enriched for association with EBVread⁺ ($P = 4.66 × 10^{-6}$, beta = 0.19, s.e.m. = 0.04). When considering 14 genes that cause monogenic EBV-driven lymphoproliferative diseases[15], the effect size increased (beta = 0.35, s.e.m. = 0.22, $P = 0.055$; Supplementary Table 14).

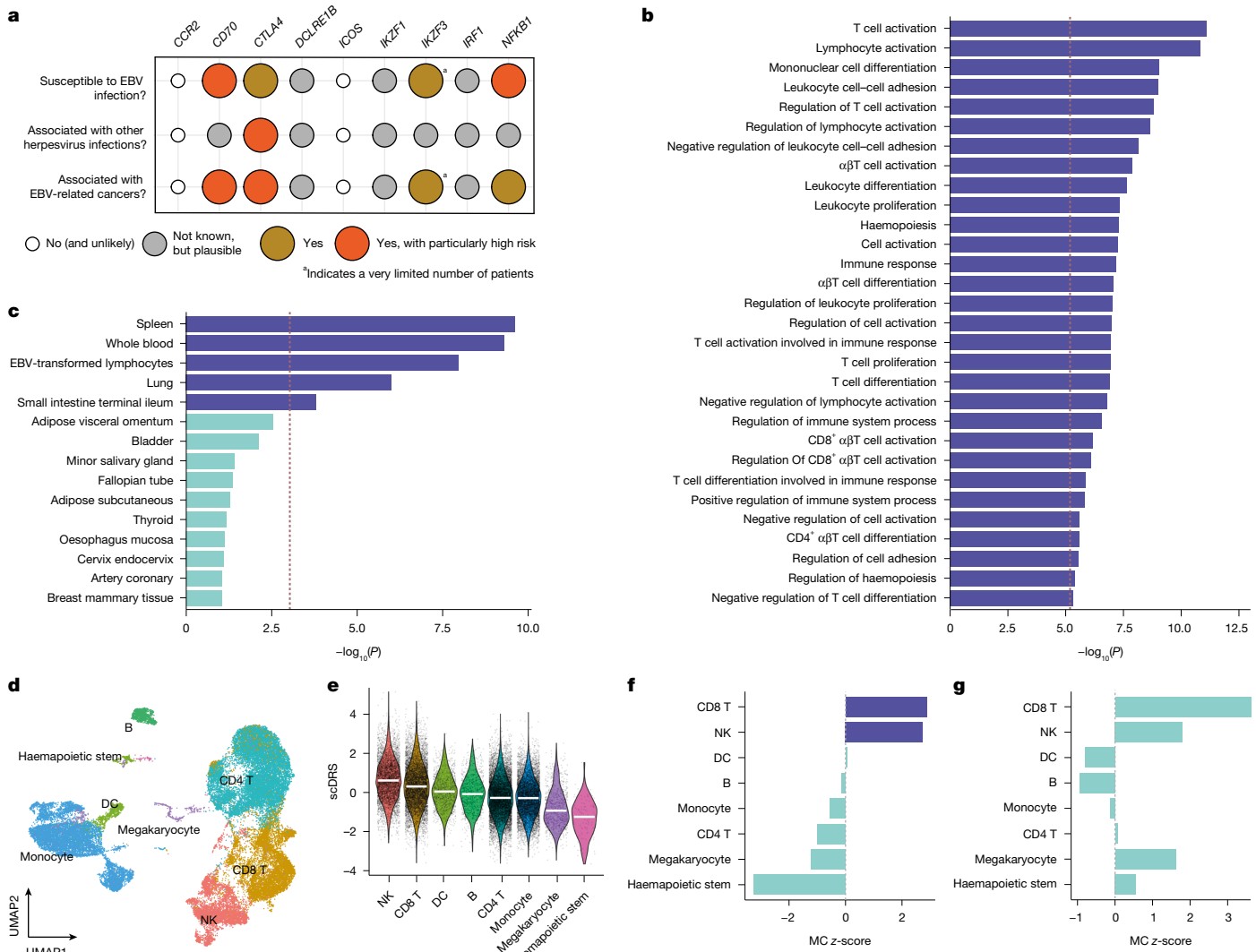

**Fig. 3 | Characterization of non-MHC risk loci associated with EBVread⁺.**
**a**, Nine genes underlying IEIs showed significant enrichment of EBVread⁺-associated common variants. Clinical information regarding EBV infection and associated outcomes for these IEIs were retrieved from literature (see Supplementary Note 7). **b**, Bar plot of $-\log_{10}(P)$ from MAGMA gene set analysis (one-sided) are shown for those Gene Ontology Biological Processes terms that remained significant after Bonferroni correction (dashed line: $P < 6.5 \times 10^{-6}$; $n = 7,743$). **c**, As in panel **b**, for the top 15 most significantly enriched GTEx tissues based on gene expression levels (MAGMA gene property test, one-sided; dashed line: $P = 9.2 \times 10^{-4}$, Bonferroni; $n = 54$). Tissues sorted by $P$ values, with the purple colour indicating significant enrichment. **d**, Uniform manifold approximation and projection (UMAP) representation plot of the PBMC single-cell RNA-seq data[50], coloured by cluster labels of cell-type annotation level 1. DC, dendritic cell. **e**, Distribution of normalized single-cell disease relevance scores (scDRS) across cell types of annotation level 1, presented in descending order based on median scDRS (white bar). Higher scores indicate cells with excess expression of genes implicated by the EBVread⁺ GWAS. **f,g**, Results of the Monte Carlo (MC)-based statistical inference cell-type association (**f**) and within-cell type heterogeneity with scDRS (**g**), based on EBVread⁺. The bar colours represent significance, with the purple colour indicating a multiple comparison-adjusted false discovery rate (FDR) < 0.05.

The aggregate effect of rare variants was captured by gene-based collapsing analyses, based on exome sequencing data (minor allele frequency < 0.01; gene-based association analysis of rare variants (RVAS$_{gene}$); Methods). Twenty-eight genes within the MHC locus and one non-MHC gene (*TNFRSF13B*) were test-wide significant in at least one of four variant pathogenicity definitions ($P_{gene} < 8.86 \times 10^{-7}$; Methods; Supplementary Table 15). The *TNFRSF13B* signal was driven by p.Cys104Arg ($P_{without\ p.Cys104Arg} = 0.087$), which is associated with common variable immunodeficiency, tonsillectomy and ear surgery[46,47].

Intersecting both analyses (MAGMA and RVAS$_{gene}$, each at $P < 0.01$) showed 24 genes with evidence from common and rare variants (Supplementary Table 16). These included seven genes whose rare variant enrichment was driven by putative loss-of-function variants (*PTPN22, GP1BA, CD226, C6orf222, ZNF284, CHD4* and *HKR1*), all of which are

strong novel candidate genes for host control of persistent EBV infection.

## Identification of candidate pathways and effector cell types

We then used the gene-based association statistics for common variants, to obtain insights into effector pathways, tissues and cell types[48]. Using Gene Ontology Biological Processes, we identified 30 test-wide significant pathways (Fig. 3b and Supplementary Table 17). These encompassed various immune processes, for example, T cell activation and differentiation, thus supporting the established role of T cells in EBV control[49]. In expression data from 54 tissues available in GTEx v8, five (that is, spleen, whole blood, EBV-transformed lymphocytes,

lung and terminal ileum) were identified as potential effector tissues (Fig. 3c). For non-blood tissues, we hypothesize that tissue-resident leukocytes are partially responsible for the observed enrichments. For blood, the enrichment was further elucidated using a gene expression dataset from peripheral blood mononuclear cells (PBMCs)[50] and the single-cell disease relevance score approach[51] (scDRS; Methods). Within eight major cell types (annotation level 1), we observed significant enrichments in CD8[+] T cells, consistent with their role in eliminating EBV-infected B cells[49] and NK cells (Fig. 3d,e). At a more fine-grained annotation (level 2, 21 cell types; Methods), the highest average scDRS was observed in the small cell cluster annotated as NK$_{bright}$ cells. Furthermore, support was generated for NK$_{dim}$ and memory CD8[+] T cells, both of which have similar enrichment $P$ values, albeit for much larger cell numbers (Extended Data Fig. 7).

We also mapped lead variants (or proxies thereof) to cell-type-specific *cis*-expression QTL (eQTL) data from PBMCs[52] (OneK1K project; Methods), and identified 18 variant–gene–cell-type associations. Most were for *ERAP2*, with consistent direction of effect in multiple cell types, including CD8[+] T and NK cells. Additional cell-type-specific eQTL effects were found for *CTLA4* and *CMC1* in S100B-positive CD8[+] T cells and *SLC22A5* in NK cells (Supplementary Table 18).

## EBVread[+] has a polygenic architecture

We then evaluated whether an aggregated genetic risk score (GRS) improves risk prediction for EBVread[+] compared with a baseline model (including age and sex), and is transferable across cohorts and ancestries. First, we assigned individuals from the UKB no outlier cohort (EUR) to one of three cohorts: (1) UKB serology target cohort (individuals for whom serology data were available), (2) UKB disease target cohort (individuals with EBV-associated diseases[4]), or (3) UKB base cohort (remaining individuals; Methods). In the UKB base cohort, we generated six GRSs, using either imputed HLA alleles (three GRSs: HLA all, HLA MHC-I and HLA MHC-II) or genotyped singe-nucleotide polymorphisms (SNPs; all, SNPs in MHC and SNPs outside of MHC; Methods).

We then applied these GRSs to the UKB serology target cohort and found that the GRSs encompassing all HLA alleles (HLA all) best explained EBVread[+] according to Nagelkerke $R^2$ (improvement over the base model: $\Delta R^2 = 0.080 \pm 0.009$ s.d.). HLA MHC-I and HLA MHC-II GRS, which represent uncorrelated predictors (Extended Data Fig. 8), performed similarly well when compared to each other (Fig. 4a). The three GRSs based on HLA alleles outperformed SNP-based GRSs, although the GRSs using SNPs outside of MHC (SNP wo MHC) captured independent genetic risk (Fig. 4a). We therefore proceeded with HLA all, HLA MHC-I, HLA MHC-II and SNP wo MHC, none of which differed between EBVsero[+] and EBVsero[−] groups (Fig. 4b) and which were positively correlated with observed EBV read counts in the serology cohort (Fig. 4c and Extended Data Fig. 8).

To analyse transferability, we applied similar GRSs within the AoU no outlier cohort, which was stratified by genetic ancestry (Methods). In the EUR subcohort, which had the highest genetic similarity to the UKB base cohort, improvements in Nagelkerke $R^2$ values compared with the baseline model were similar to our results from UKB, with HLA all best explaining EBVread[+] ($\Delta R^2 = 0.072 \pm 0.002$ s.d.; Fig. 4d and Extended Data Fig. 8). Similarly, HLA all showed the largest improvements in Nagelkerke $R^2$ in each of the five non-EUR ancestry groups, despite differences in absolute values (Fig. 4d). In the African ($\Delta R^2 = 0.055 \pm 0.002$ s.d.) and admixed American ($\Delta R^2 = 0.065 \pm 0.002$ s.d.) groups, predictive performance was similar to that of the AoU EUR subcohort (Fig. 4d). This demonstrates some degree of transferability for the GRS comprising all HLA alleles. In all ancestry groups, the SNP-based GRS was least predictive, but was again similar between the EUR subcohorts of UKB and AoU (Extended Data Fig. 8). These results provide evidence for a polygenic component to EBV viral load that is largely driven by the MHC region and can be transferred across ancestries when calculated based on HLA alleles.

## GRSs associate with EBV-associated and novel diseases

The four selected GRSs were then applied to the UKB disease target cohorts (infectious mononucleosis, Hodgkin lymphoma, multiple sclerosis, rheumatoid arthritis, non-Hodgkin lymphoma, systemic lupus erythematosus and/or Sjögren disease; see above). Highly significant associations were found for an elevated HLA MHC-I GRS in multiple sclerosis and an elevated HLA MHC-II GRS in rheumatoid arthritis (Fig. 4e). For multiple sclerosis, this effect was attenuated when HLA-A*02:01 was excluded from the GRS ($P_{\text{HLA MHC-I}} = 3.09 \times 10^{-5}$, $P_{\text{without HLA-A*02:01}} = 0.031$). By contrast, exclusion of HLA-DRB1*04:04, which is a risk factor for rheumatoid arthritis[35] and was the most significant HLA allele in the EBVread[+] GWAS, from the HLA MHC-II GRS did not attenuate the association of this GRS with rheumatoid arthritis. At $P < 0.1$, we also observed a lower HLA all GRS in individuals with non-Hodgkin lymphoma, and a lower HLA MHC-I GRS in rheumatoid arthritis (Fig. 4e).

We then conducted a phenome-wide association study (PheWAS) in the EUR AoU QC cohort using 1,751 PheCodes. With the exception of Sjögren disease, these PheCodes included all of the aforementioned EBV-associated diseases (Methods; Fig. 4f and Extended Data Fig. 8). At $P < 0.001$, the PheWas replicated all four significant associations identified in UKB. This approach also identified novel candidate diseases associated with EBV host control: the strongest associations were found for type 1 diabetes (beta = 0.176, s.e. = 0.023 for HLA MHC-II), inflammatory bowel disease (beta = −0.14, s.e. = 0.018 for HLA all, and beta = −0.112, s.e. = 0.018 for HLA MHC-II) and hypothyroidism (beta = −0.043, s.e. = 0.008 for HLA MHC-I, and beta = 0.037, s.e. = 0.007 for SNP wo MHC; Supplementary Table 19).

## Suggestive causal effects of EBVread[+] are driven by variants in MHC region

To investigate whether EBVread[+] as an exposure has a causal effect, we performed two-sample Mendelian randomization (2SMR; Methods) for the five diseases with strong evidence for epidemiological association (that is, multiple sclerosis, rheumatoid arthritis, Hodgkin lymphoma, non-Hodgkin lymphoma and systemic lupus erythematosus)[4], and three diseases identified by our PheWAS (that is, inflammatory bowel disease, type 1 diabetes and hypothyroidism). For multiple sclerosis, we tested both case–control status and disease course severity (Supplementary Table 20). We found suggestive evidence for causal effects of EBVread[+] on rheumatoid arthritis (beta$_{\text{wMed}}$ = 0.192, s.e. = 0.053) and type 1 diabetes (beta$_{\text{wMed}}$ = 0.620, s.e. = 0.062), which were consistent across six estimators including two that are robust to pleiotropy (Methods; Supplementary Table 21, Supplementary Fig. 6 and Supplementary Note 8). However, the effects on both outcomes were driven by variants in the MHC region (Supplementary Table 22). Attributing causality is thus problematic, given the unknown extent of pleiotropic effects of MHC variants, and the limited heritability of EBVread[+] attributed to non-MHC variants. No evidence for an EBVread[+] causal effect was found for the other seven tested outcomes (Supplementary Table 21) or the negative control trait (Methods).

## Discussion

This study is one of the first to demonstrate that GS-based EBVreads are a highly specific proxy for elevated EBV viral load in blood cells. Using this measure, we identified associations between EBVread[+] and several non-genetic factors, including current smoking as well as sex. Smoking is also a risk factor for several EBV-associated diseases[53–55],

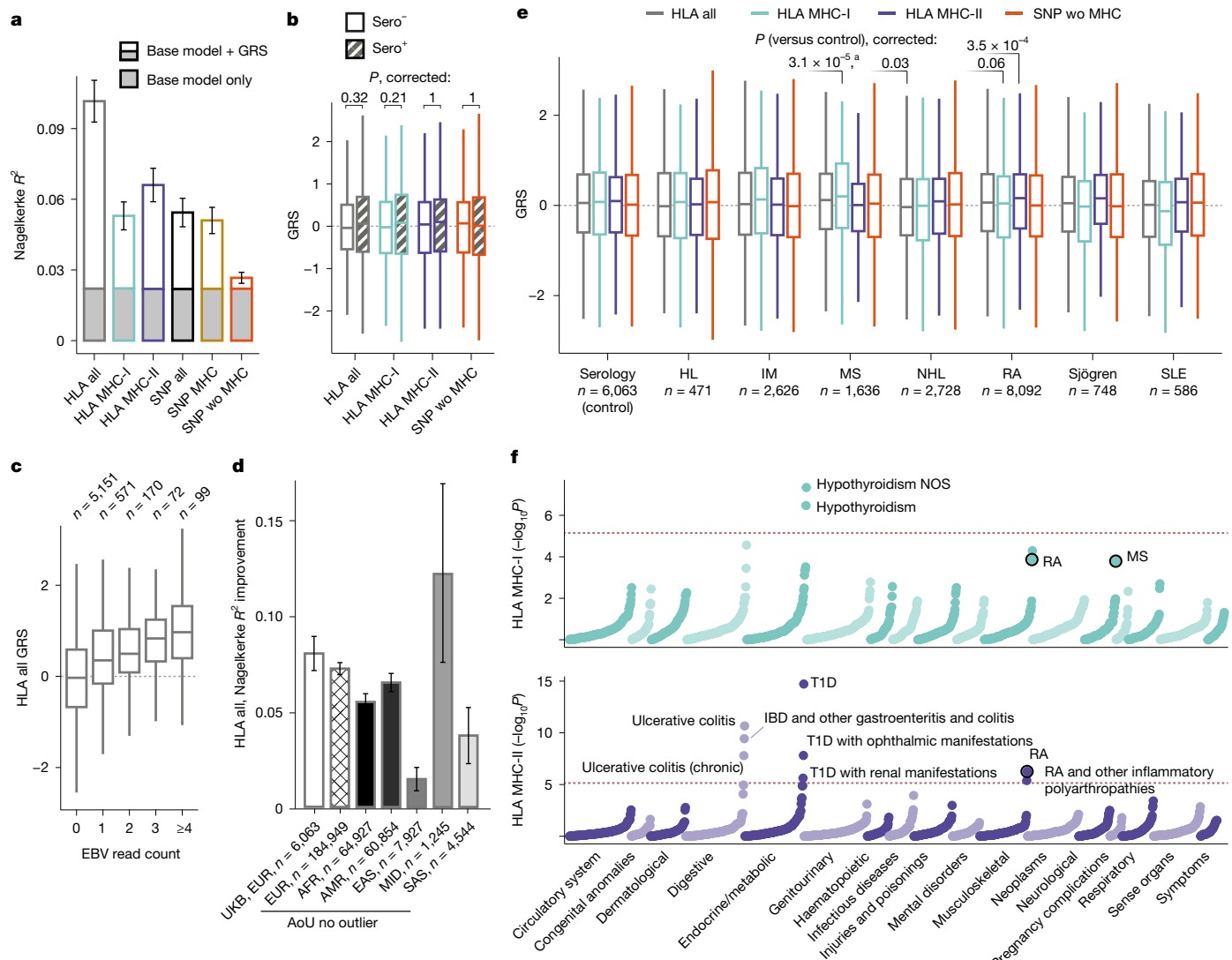

**Fig. 4 | GRS analyses in UKB and AoU. a**, In the UKB serology target cohort ($n = 6,063$, unrelated, EUR), EBVread[+] status was predicted using a baseline model with or without one of six GRSs: imputed HLA alleles (HLA all); HLA alleles of MHC-I (HLA MHC-I); HLA alleles of MHC-II (HLA MHC-II); genotyped SNPs (SNP all); SNPs within MHC (SNP MHC); or all non-MHC SNPs (SNP wo MHC). Nagelkerke $R^2$ values are plotted (error bars denote standard deviations; bootstrapped, $n = 1,000$). **b**, GRSs were compared between sero[−] ($n = 348$) and sero[+] ($n = 5,715$) individuals and were non-significant ($P$ values Bonferroni adjusted for four tests; statistical test: likelihood ratio test applied to logistic regression models, adjusted for covariates; see Methods). **c**, HLA all was positively correlated with EBV read counts. Sample sizes are indicated above the boxplots. **d**, Improvements in Nagelkerke $R^2$ for different AoU ancestry groups (HLA all GRS; abbreviations as in Extended Data Fig. 3), compared with the baseline model within UKB (from panel **a**). Error bars represent standard deviations (bootstrapped, $n = 1,000$). See the $x$ axis for sample sizes. AFR, African; AMR, admixed American; EAS, East Asian; MID, Middle Eastern; SAS, South Asian. **e**, GRS distributions in individuals of the

UKB serology target cohort and with EBV-associated diseases (UKB disease target cohort; see the $x$ axis for sample sizes). Individuals with multiple diseases were included in each respective group. The statistical test is as in panel **b**. $P$ values (Bonferroni adjusted) are provided if $P < 0.1$. [a]For multiple sclerosis (MS), the signal was driven by HLA-A*02:01. HL, Hodgkin lymphoma; IM, infectious mononucleosis; NHL, non-Hodgkin lymphoma; RA, rheumatoid arthritis; SLE, systemic lupus erythematosus. **f**, $-\log_{10}(P)$ of 1,751 PheCodes, grouped by organ systems or disease groups, for GRS HLA MHC-I and HLA MHC-II (statistical test as in panel **b**; dashed line: Bonferroni-corrected significance threshold) from the AoU QC EUR subset ($n = 189,658$). Phenotype terms are provided for test-wide significant results and for associations identified in panel **e** (encircled). IBD, inflammatory bowel disease; NOS, not otherwise specified; T1D, type 1 diabetes. The boxplots show the median (thick line), 25th and 75th percentiles (box) and the largest–smallest values no further from the box than 1.5 times the interquartile range (whiskers; **b,c,e**). Dashed lines in **b,c,e** correspond to values of 0.

although the underlying mechanisms remain largely unknown. Current smoking affects both adaptive and innate immunity, with the latter normalizing upon smoking cessation[56]. This suggests an interaction of the innate immune system with current smoking status in EBV host control. The increased prevalence of EBVread[+] in male sex encourages investigations into sex-specific factors, especially in the light of the contrary female predisposition of autoimmune diseases, including multiple sclerosis[57].

We found that EBVread[+] is polygenic and characterized by a major (and largely equal) contribution of alleles at MHC-I and MHC-II, which supports previous observations that CD8[+] cytotoxic T and NK cells[49] (MHC-I) as well as CD4[+] helper T cells[49,58] (MHC-II) are important in EBV control. Some genes implicated by common variants underly monogenic IEIs with increased susceptibility to severe EBV infections, often associated with a pronounced risk of EBV-associated diseases including lymphoma (for example, *CD70*)[59,60]. Our results thus probably harbour

novel candidate genes for IEIs, such as *CD226*, which is a member of the immunoglobulin superfamily that contributes to NK and CD8+ T cell regulation[61] and impairs CD8+ T cell response in chronic HIV when downregulated[62].

Using genetically predicted EBV viral load, we identified genetic overlap with multiple sclerosis and rheumatoid arthritis. Although EBV is a prerequisite for multiple sclerosis, HLA-A*02:01, which reduces multiple sclerosis risk, was among our most significant findings and was associated with better EBV control. By contrast, no consistent effect on EBVread+ was found for the major multiple sclerosis risk allele HLA-DRB1*15:01 (ref. 36), suggesting a pathomechanism distinct from EBV viral load control. This could include a stronger antibody response through preferential EBV peptide presentation[63,64], expansion of specific B cell subsets[65] or molecular mimicry. In support of this, detailed analysis of HLA-DRB1*15:01 (Extended Data Fig. 9) found that the strongest effect size was with antibody levels of IgG EBNA-1, in line with previous findings that antibodies to EBNA-1 cross-react with the central nervous system protein GlialCAM[8]. In rheumatoid arthritis, alleles at MHC-I and MHC-II were associated with lower and higher EBV viral load, respectively. This suggests a specific dysregulation of the immune response to EBV, rather than a generic loss of EBV immune control. Although further research is required to determine whether the effect of EBV viral load is causal, the 2SMR results support this hypothesis. Our analyses also revealed a genetic overlap between EBV control and type 1 diabetes, inflammatory bowel disease or ulcerative colitis, and hypothyroidism, suggesting that the pathophysiological relevance of EBV host control may be broader than currently assumed.

Our study had several limitations. First, owing to the standard depth of human GS, most individuals had an EBV read count of zero, and many had an EBV read count of exactly 1. For statistical analyses, we binarized the phenotype into low or high EBV viral load, based on absolute EBV read count numbers, and compared EBV read count 0 versus 1 and higher. Given the limited resolution, some individuals with presumed high viral load might actually have low viral load. However, this potential mis-classification is unlikely to have impacted the overall conclusions, which are supported by our sensitivity analyses and are similar to those of a recent study, which used a different definition for increased viral load[66]. If specific quantitative measures or deeper GS data become available, statistical power will probably increase. Second, unobserved factors may have confounded associations with EBVread+, although we mitigated this risk by replicating findings across biobanks. Third, despite the partial transferability of HLA-based GRS across ancestries, the discovery analyses mainly involved EUR individuals. This might have influenced the identity of associated HLA alleles, and limit the generalizability of the findings with respect to different EBV strains and EBV-associated diseases, which vary in terms of global distribution and prevalence. Thus, replication of the GWAS findings and downstream analyses in non-EUR ancestries are required. Finally, given the biological complexity of the MHC region and current challenges in HLA allele imputation[67], some HLA associations might have been missed or mimicked by extended regions of LD.

This work has established EBV viral sequence traces from blood-based human GS data as the basis for future investigations into functional, mechanistic and epidemiological aspects of persistent EBV infection. Quantification of viral load using host GS data could be extended to other human pathogens, and facilitate investigation of interactions between chronic infections and the host immune system in health and disease.

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

**Japan COVID-19 Task Force**

Genta Nagao[24], Hiromu Tanaka[24], Shuhei Azekawa[24], Ko Lee[24], Naoki Fukunaga[24], Junko Hamamoto[24], Hiroki Kabata[24], Katsunori Masaki[24], Hirofumi Kamata[24], Shinnosuke Ikemura[24], Shotaro Chubachi[24], Satoshi Okamori[24], Hideki Terai[24], Atsuho Morita[24], Takanori Asakura[24], Makoto Ishii[24], Koichi Fukunaga[24], Yoshifumi Uwamino[25], Sho Uchida[20], Shunsuke Uno[20], Tomoyasu Nishimura[20,26], Ho Namkoong[20], Naoki Hasegawa[20], Emmy Yanagita[27], Hiroshi Nishihara[27], Junichi Sasaki[28], Hiroshi Morisaki[29], Toshiro Sato[30], Yuko Kitagawa[31], Yuta Matsubara[32], Yohei Mikami[32], Kosaku Nanki[32], Takanori Kanai[32], Ryuya Edahiro[5,21,33], Yuya Shirai[5,21,33], Kyuto Sonehara[3,5,21], Daisuke Okuzaki[34], Daisuke Motooka[35], Masahiro Kanai[36], Tatsuhiko Naito[3,5,21], Kenichi Yamamoto[21,37], Qingbo S. Wang[21], Yasuhiro Kato[33,38], Takayoshi Morita[33,38], Shinichi Namba[3,5,21], Ken Suzuki[21], Yoko Naito[35], Yu-Chen Liu[34], Ayako Takuwa[34], Fuminori Sugihara[39], James B. Wing[40], Shuhei Sakakibara[41], Nobuyuki Hizawa[42], Takayuki Shiroyama[33], Satoru Miyawaki[43], Yusuke Kawamura[44], Akiyoshi Nakayama[44], Hirotaka Matsuo[44], Yuichi Maeda[33], Takuro Nii[33], Yoshimi Noda[33], Takayuki Niitsu[33], Yuichi Adachi[33], Takatoshi Enomoto[33], Saori Amiya[33], Reina Hara[33], Yuta Yamaguchi[33,38], Teruaki Murakami[33,38], Tomoki Kuge[33], Kinnosuke Matsumoto[33], Yuji Yamamoto[33], Makoto Yamamoto[33], Midori Yoneda[33], Toshihiro Kishikawa[21,45,46], Shuhei Yamada[47], Shuhei Kawabata[47], Noriyuki Kijima[47], Masatoshi Takagaki[47], Noah Sasa[3,5,21,45], Yuya Ueno[45], Motoyuki Suzuki[45], Norihiko Takemoto[45], Hirotaka Eguchi[45], Takahito Fukusumi[45], Takao Imai[45], Munehisa Fukushima[45,48], Haruhiko Kishima[47], Hidenori Inohara[45], Kazunori Tomono[49], Kazuto Kato[50], Meiko Takahashi[51], Fumihiko Matsuda[51], Haruhiko Hirata[33], Yoshito Takeda[33], Atsushi Kumanogoh[33,38,52,53], Yukinori Okada[3,5,21,22,23], Takanori Hasegawa[54], Kunihiko Takahashi[54], Tatsuhiko Anzai[54], Satoshi Ito[54], Yuji Uchimura[55], Akifumi Endo[56], Yasunari Miyazaki[57], Takayuki Honda[57], Tomoya Tateishi[57], Shuji Tohda[58], Naoya Ichimura[58], Kazunari Sonobe[58], Chihiro Tani Sassa[58], Jun Nakajima[58], Masumi Ai[59], Ryuji Koike[60], Akinori Kimura[61], Satoru Miyano[54], Tomomi Takano[62], Kazuhiko Katayama[63], Koji Okudela[64], Ryunosuke Saiki[65], Yasuhito Nannya[65], Seishi Ogawa[65,66], Takayoshi Hyugaji[67], Eigo Shimizu[67], Kotoe Katayama[67], Seiya Imoto[67], Yosuke Omae[68], Katsushi Tokunaga[68], Takafumi Ueno[69], Yoshinori Fukui[70], Hiroyuki Hayashi[71], Yukihiro Yoshimura[72], Natsuo Tachikawa[72], Kazuhisa Takahashi[73], Norihiro Harada[73], Yuki Tanabe[73], Toshio Naito[74], Makoto Hiki[75,76], Yasushi Matsushita[77], Haruhi Takagi[73], Ryousuke Aoki[78], Ai Nakamura[73], Sonoko Harada[73,79], Hitoshi Sakurai[80], Takashi Ishiguro[80], Taisuke Isono[80], Shun Shibata[80], Yuma Matsui[80], Chiaki Hosoda[80], Kenji Takano[80], Takashi Nishida[80], Yoichi Kobayashi[80], Yotaro Takaku[80], Noboru Takayanagi[80], Soichiro Ueda[81], Natsumi Yazaki[81], Ai Tada[81], Masayoshi Miyawaki[81], Masaomi Yamamoto[81], Eriko Yoshida[81], Reina Hayashi[81], Tomoki Nagasaka[81], Sawako Arai[81], Yutaro Kaneko[81], Kana Sasaki[81], Etsuko Tagaya[82], Masatoshi Kawana[83], Ken Arimura[82], Yasushi Nakano[84], Yukiko Nakajima[84], Ryusuke Anan[84], Ryosuke Arai[84], Yuko Kurihara[84], Yuko Harada[84], Kazumi Nishio[84], Tetsuya Ueda[85], Masanori Azuma[85], Ryuichi Saito[85], Toshikatsu Sado[85], Yoshimune Miyazaki[85], Ryuichi Sato[85], Yuki Haruta[85], Tadao Nagasaki[85], Yoshinori Yasui[86], Yoshinori Hasegawa[85], Akihiro Noda[85], Yusei Fukushima[85], Reina Kitagawa[85], Yoshikazu Mutoh[87], Tomoki Kimura[88], Tomonori Sato[88], Reoto Takei[88], Satoshi Hagimoto[88], Yoichiro Noguchi[88], Yasuhiko Yamano[88], Hajime Sasano[88], Sho Ota[88], Yasushi Nakamori[89], Kazuhisa Yoshiya[89], Fukuki Saito[89], Tomoyuki Yoshihara[89], Daiki Wada[89], Hiromu Iwamura[89], Syuji Kanayama[89], Shuhei Maruyama[89], Takashi Yoshiyama[90], Ken Ohta[90], Hiroyuki Kokuto[90], Hideo Ogata[90], Yoshiaki Tanaka[90], Kenichi Arakawa[90], Masafumi Shimoda[90], Takeshi Osawa[90], Hiroki Tateno[91], Isano Hase[91], Shuichi Yoshida[91], Shoji Suzuki[91], Miki Kawada[92], Hirohisa Horinouchi[93], Fumitake Saito[94], Keiko Mitamura[95], Masao Hagihara[96], Junichi Ochi[94], Tomoyuki Uchida[96], Rie Baba[97], Daisuke Arai[97], Takayuki Ogura[97], Hidenori Takahashi[97], Shigehiro Hagiwara[97], Shunichiro Konishi[97], Ichiro Nakachi[97], Koji Murakami[98], Mitsuhiro Yamada[98], Hisatoshi Sugiura[98], Hirohito Sano[98], Shuichiro Matsumoto[98], Nozomu Kimura[98], Yoshinao Ono[98], Hiroaki Baba[99], Yusuke Suzuki[100], Sohei Nakayama[100], Keita Masuzawa[101], Hidefumi Koh[101], Tadashi Manabe[101], Yohei Funatsu[101], Fumimaro Ito[101], Takahiro Fukui[101], Keisuke Shinozuka[101], Sumiko Kohashi[101], Masatoshi Miyazaki[101], Tomohisa Shoko[102], Takashi Inoue[103], Takahiro Asami[103], Toshiyuki Hirano[103], Keigo Kobayashi[103], Hatsuyo Takaoka[103], Kazuyoshi Watanabe[104], Naoki Miyazawa[104], Yasuhiro Kimura[105], Reiko Sado[105], Hideyasu Sugimoto[105], Akane Kamiya[106], Naota Kuwahara[107], Akiko Fujiwara[107], Tomohiro Matsunaga[107], Yoko Sato[107], Takenori Okada[107], Yoshihiro Hirai[108], Hidetoshi Kawashima[108], Atsuya Narita[108], Kazuki Niwa[109], Yoshiyuki Sekikawa[110], Koichi Nishi[111], Masaru Nishitsuji[111], Mayuko Tani[111], Junya Suzuki[111], Hiroki Nakatsumi[111], Takashi Ogura[112], Hideya Kitamura[112], Eri Hagiwara[112], Kota Murohashi[112], Hiroko Okabayashi[112], Takao Mochimaru[113,114], Shigenari Nukaga[113], Ryosuke Satomi[113], Yoshitaka Oyamada[113,114], Nobuaki Mori[115], Tomoya Baba[116], Yasutaka Fukui[116], Mitsuru Odate[116], Shuko Mashimo[116], Yasushi Makino[116], Kazuma Yagi[117], Mizuha Hashiguchi[117], Junko Kagyo[117], Tetsuya Shiomi[117], Satoshi Fuke[118], Hiroshi Saito[118], Tomoya Tsuchida[119], Shigeki Fujitani[120], Mumon Takita[120], Daiki Morikawa[120], Toru Yoshida[120], Takehiro Izumo[121], Minoru Inomata[121], Naoyuki Kuse[121], Nobuyasu Awano[121], Mari Tone[121], Akihiro Ito[122], Yoshihiko Nakamura[123], Kota Hoshino[123], Junichi Maruyama[123], Hiroyasu Ishikura[123], Tohru Takata[124], Toshio Odani[125], Masaru Amishima[126], Takeshi Hattori[126], Yasuo Shichinohe[127], Takashi Kagaya[128], Toshiyuki Kita[128], Kazuhide Ohta[128], Satoru Sakagami[128], Kiyoshi Koshida[128], Kentaro Hayashi[129], Tetsuo Shimizu[129], Yutaka Kozu[129], Hisato Hiranuma[129], Yasuhiro Gon[129], Namiki Izumi[130], Kaoru Nagata[130], Ken Ueda[130], Reiko Taki[130], Satoko Hanada[130], Kodai Kawamura[131], Kazuya Ichikado[131], Kenta Nishiyama[131], Hiroyuki Muranaka[131], Kazunori Nakamura[131], Naozumi Hashimoto[132], Keiko Wakahara[132], Sakamoto Koji[132], Norihito Omote[132], Akira Ando[132], Nobuhiro Kodama[133], Yasunari Kaneyama[133], Shunsuke Maeda[133], Takashige Kuraki[134], Takemasa Matsumoto[134], Koutaro Yokote[135], Taka-Aki Nakada[136], Ryuzo Abe[136], Taku Oshima[136], Tadanaga Shimada[136], Masahiro Harada[137], Takeshi Takahashi[137], Hiroshi Ono[137], Toshihiro Sakurai[137], Takayuki Shibusawa[137], Yoshifumi Kimizuka[138], Akihiko Kawana[138], Tomoya Sano[138], Chie Watanabe[138], Ryohei Suematsu[138], Hisako Sageshima[139], Ayumi Yoshifuji[140], Kazuto Ito[140], Saeko Takahashi[141], Kota Ishioka[141], Morio Nakamura[142], Makoto Masuda[143], Aya Wakabayashi[143], Hiroki Watanabe[143], Suguru Ueda[143], Masanori Nishikawa[143], Yusuke Chihara[144], Mayumi Takeuchi[144], Keisuke Onoi[144], Jun Shinozuka[144], Atsushi Sueyoshi[144,145], Yoji Nagasaki[146], Masaki Okamoto[147,148], Sayoko Ishihara[146], Masatoshi Shimo[146], Yoshihisa Tokunaga[147,148], Yu Kusaka[149], Takehiko Ohba[149], Susumu Isogai[149], Satoru Fukuyama[150], Yoshihiro Eriguchi[151], Akiko Yonekawa[151], Keiko Kan-o[150], Koichiro Matsumoto[150], Kensuke Kanaoka[152], Shoichi Ihara[152], Kiyoshi Komuta[152], Yoshiaki Inoue[153], Shigeru Chiba[154], Kunihiro Yamagata[155], Yuji Hiramatsu[156], Hirayasu Kai[155], Koichiro Asano[157], Tsuyoshi Oguma[157], Yoko Ito[157], Satoru Hashimoto[158], Masaki Yamasaki[158], Yu Kasamatsu[159], Yuko Komase[160], Naoya Hida[160], Takahiro Tsuburai[160], Baku Oyama[160], Minoru Takada[161], Hidenori Kanda[161], Yuichiro Kitagawa[162], Tetsuya Fukuta[162], Takahito Miyake[162], Shozo Yoshida[162], Shinji Ogura[163], Shinji Abe[164], Yuta Kono[164], Yuki Togashi[164], Hiroyuki Takoi[164], Ryota Kikuchi[164], Shinichi Ogawa[165], Tomouki Ogata[165], Shoichiro Ishihara[165], Arihiko Kanehiro[166,167], Shinji Ozaki[166], Yasuko Fuchimoto[166], Sae Wada[166], Nobukazu Fujimoto[166], Kei Nishiyama[168], Mariko Terashima[169], Satoru Beppu[169], Kosuke Yoshida[169], Osamu Narumoto[170], Hideaki Nagai[170], Nobuharu Ooshima[170], Mitsuru Motegi[171], Akira Umeda[172], Kazuya Miyagawa[173], Hisato Shimada[174], Mayu Endo[175], Yoshiyuki Ohira[176], Masafumi Watanabe[177], Sumito Inoue[177], Akira Igarashi[177], Masamichi Sato[177], Hironori Sagara[178], Akihiko Tanaka[178], Shin Ohta[178], Tomoyuki Kimura[178], Yoko Shibata[179], Yoshinori Tanino[179], Takefumi Nikaido[179], Hiroyuki Minemura[179], Yuki Sato[179], Yuichiro Yamada[180], Takuya Hashino[180], Masato Shinoki[180], Hajime Iwagoe[181], Hiroshi Takahashi[182], Kazuhiko Fujii[182], Hiroto Kishi[182], Masayuki Kanai[183], Tomonori Imamura[183], Tatsuya Yamashita[183], Masakiyo Yatomi[184], Toshitaka Maeno[184], Shinichi Hayashi[185], Mai Takahashi[185], Mizuki Kuramochi[185], Isamu Kamimaki[185], Yoshiteru Tominaga[185], Tomoo Ishii[186], Mitsuyoshi Utsugi[187], Akihiro Ono[187], Toru Tanaka[188], Takeru Kashiwada[188], Kazue Fujita[188], Yoshinobu Saito[188], Masahiro Seike[188], Hiroko Watanabe[189], Hiroto Matsuse[190], Norio Kodaka[190], Chihiro Nakano[190], Takeshi Oshio[190], Takatomo Hirouchi[190], Shohei Makino[191], Moritoki Egi[191] & The Biobank Japan Project[192]

[24]Division of Pulmonary Medicine, Department of Medicine, Keio University School of Medicine, Tokyo, Japan. [25]Department of Laboratory Medicine, Keio University School of Medicine, Tokyo, Japan. [26]Keio University Health Center, Tokyo, Japan. [27]Genomics Unit, Keio Cancer Center, Keio University Hospital, Tokyo, Japan. [28]Department of Emergency and Critical Care Medicine, Keio University School of Medicine, Tokyo, Japan. [29]Department of Anesthesiology, Keio University School of Medicine, Tokyo, Japan. [30]Department of Organoid Medicine, Keio University School of Medicine, Tokyo, Japan. [31]Department of Surgery, Keio University School of Medicine, Tokyo, Japan. [32]Division of Gastroenterology and Hepatology, Department of Medicine, Keio University School of Medicine, Tokyo, Japan. [33]Department of Respiratory Medicine and Clinical Immunology, Graduate School of Medicine, The University of Osaka, Suita, Japan. [34]Single Cell Genomics, Human Immunology, WPI Immunology Frontier Research Center, The University of Osaka, Suita, Japan. [35]Genome Information Research Center, Research Institute for Microbial Diseases, The University of Osaka, Suita, Japan. [36]Department of Biomedical Informatics, Harvard Medical School, Boston, MA, USA. [37]Laboratory of Children's Health and Genetics, Division of Health Science, Graduate School of Medicine, The University of Osaka, Suita, Japan. [38]Department of Immunopathology, Immunology Frontier Research Center (WPI-IFReC), The University of Osaka, Suita, Japan. [39]Core Instrumentation Facility, Immunology Frontier Research Center and Research Institute for Microbial Diseases, The University of Osaka, Suita, Japan. [40]Laboratory of Human Immunology (Single Cell Immunology), Immunology Frontier Research Center, The University of Osaka, Suita, Japan. [41]Laboratory of Immune Regulation, Immunology Frontier Research Center, The University of Osaka, Suita, Japan. [42]Department of Pulmonary Medicine, Faculty of Medicine, University of Tsukuba, Tsukuba, Japan. [43]Department of Neurosurgery, Faculty of Medicine, The University of Tokyo, Tokyo, Japan. [44]Department of Integrative Physiology and Bio-Nano Medicine, National Defense Medical College, Tokorozawa, Japan. [45]Department of Otorhinolaryngology-Head and Neck Surgery, Graduate School of Medicine, The University of Osaka, Suita, Japan. [46]Department of Head and Neck Surgery, Aichi Cancer Center Hospital, Nagoya, Japan. [47]Department of Neurosurgery, Graduate School of Medicine, The University of Osaka, Suita, Japan. [48]Department of Otolaryngology and Head and Neck Surgery, Kansai Rosai Hospital, Hyogo, Japan. [49]Division of Infection Control and Prevention, The University of Osaka Hospital, Suita, Japan. [50]Department of Biomedical Ethics and Public Policy, Graduate School of Medicine, The University of Osaka, Suita, Japan. [51]Center for Genomic Medicine, Kyoto University Graduate School of Medicine, Kyoto, Japan. [52]Integrated Frontier Research for Medical Science Division, Institute for Open and Transdisciplinary Research Initiatives, The University of Osaka, Suita, Japan. [53]Center for Infectious Disease Education and Research (CiDER), The University of Osaka, Suita, Japan. [54]M&D Data Science Center, Institute of Integrated Research, Institute of Science Tokyo, Tokyo, Japan. [55]Department of Medical Informatics, Institute of Science Tokyo Hospital, Tokyo, Japan. [56]Clinical Research Center, Institute of Science Tokyo Hospital, Tokyo, Japan. [57]Respiratory Medicine, Institute of Science Tokyo Hospital, Tokyo, Japan. [58]Clinical Laboratory, Institute of Science Tokyo Hospital, Tokyo, Japan. [59]Department of Insured Medical Care Management, Institute of Science Tokyo Hospital, Tokyo, Japan. [60]Health Science Research and Development Center (HeRD), Institute of Science Tokyo, Tokyo, Japan. [61]Institute of Science Tokyo, Tokyo, Japan. [62]Laboratory of Veterinary Infectious Disease, School of Veterinary Medicine, Kitasato University, Aomori, Japan. [63]Laboratory of Viral Infection, Department of Infection Control and Immunology, Omura Satoshi Memorial Institute and Graduate School of Infection Control Sciences, Kitasato University, Tokyo, Japan. [64]Department of Pathology Saitama Medical University, Saitama, Japan. [65]Department of Pathology and Tumor Biology, Kyoto University, Kyoto, Japan. [66]Institute for the Advanced Study of Human Biology (WPI-ASHBi), Kyoto University, Kyoto, Japan. [67]Division of Health Medical Intelligence, Human Genome Center, Institute of Medical Science, The University of Tokyo, Tokyo, Japan. [68]Genome Medical Science Project (Toyama), National Center for Global Health and Medicine, Tokyo, Japan. [69]Department of Biomolecular Engineering, Graduate School of Tokyo Institute of Technology, Tokyo, Japan. [70]Division of Immunogenetics, Department of Immunobiology and Neuroscience, Medical Institute of Bioregulation, Kyushu University, Fukuoka, Japan. [71]Division of Pathology, Yokohama Municipal Citizen's Hospital, Yokohama, Japan. [72]Division of Infectious Disease, Yokohama Municipal Citizen's Hospital, Yokohama, Japan. [73]Department of Respiratory Medicine, Juntendo University Faculty of Medicine and Graduate School of Medicine, Tokyo, Japan. [74]Department of General Medicine, Juntendo University Faculty of Medicine and Graduate School of Medicine, Tokyo, Japan. [75]Department of Emergency and Disaster Medicine,

Juntendo University Faculty of Medicine and Graduate School of Medicine, Tokyo, Japan. [76]Department of Cardiovascular Biology and Medicine, Juntendo University Faculty of Medicine and Graduate School of Medicine, Tokyo, Japan. [77]Department of Internal Medicine and Rheumatology, Juntendo University Faculty of Medicine and Graduate School of Medicine, Tokyo, Japan. [78]Department of Nephrology, Juntendo University Faculty of Medicine and Graduate School of Medicine, Tokyo, Japan. [79]Atopy (Allergy) Research Center, Juntendo University Graduate School of Medicine, Tokyo, Japan. [80]Department of Respiratory Medicine, Saitama Cardiovascular and Respiratory Center, Kumagaya, Japan. [81]Internal Medicine, Japan Community Healthcare Organization Saitama Medical Center, Saitama, Japan. [82]Department of Respiratory Medicine, Tokyo Women's Medical University, Tokyo, Japan. [83]Department of General Medicine, Tokyo Women's Medical University, Tokyo, Japan. [84]Kawasaki Municipal Ida Hospital, Department of Internal Medicine, Kawasaki, Japan. [85]Department of Respiratory Medicine, Osaka Saiseikai Nakatsu Hospital, Osaka, Japan. [86]Department of Infection Control, Osaka Saiseikai Nakatsu Hospital, Osaka, Japan. [87]Department of Infectious Diseases, Tosei General Hospital, Seto, Japan. [88]Department of Respiratory, Allergic Diseases Internal Medicine, Tosei General Hospital, Seto, Japan. [89]Department of Emergency and Critical Care Medicine, Kansai Medical University General Medical Center, Moriguchi, Japan. [90]Fukujuji hospital, Kiyose, Japan. [91]Department of Pulmonary Medicine, Saitama City Hospital, Saitama, Japan. [92]Department of Infectious Diseases, Saitama City Hospital, Saitama, Japan. [93]Department of General Thoracic Surgery, Saitama City Hospital, Saitama, Japan. [94]Department of Pulmonary Medicine, Eiju General Hospital, Tokyo, Japan. [95]Division of Infection Control, Eiju General Hospital, Tokyo, Japan. [96]Department of Hematology, Eiju General Hospital, Tokyo, Japan. [97]Saiseikai Utsunomiya Hospital, Utsunomiya, Japan. [98]Department of Respiratory Medicine, Tohoku University Graduate School of Medicine, Sendai, Japan. [99]Department of Infectious Diseases, Tohoku University Graduate School of Medicine, Sendai, Japan. [100]Department of Respiratory Medicine, Kitasato University Kitasato Institute Hospital, Tokyo, Japan. [101]Tachikawa Hospital, Tachikawa, Japan. [102]Department of Emergency and Critical Care Medicine, Tokyo Women's Medical University Adachi Medical Center, Tokyo, Japan. [103]Internal Medicine, Sano Kosei General Hospital, Sano, Japan. [104]Japan Community Healthcare Organization Kanazawa Hospital, Kanazawa, Japan. [105]Department of Respiratory Medicine, Saiseikai Yokohamashi Nanbu Hospital, Yokohama, Japan. [106]Department of Clinical Laboratory, Saiseikai Yokohamashi Nanbu Hospital, Yokohama, Japan. [107]Internal Medicine, Internal Medicine Center, Showa University Koto Toyosu Hospital, Tokyo, Japan. [108]Department of Respiratory Medicine, Japan Organization of Occupational Health and Safety, Kanto Rosai Hospital, Kawasaki, Japan. [109]Department of General Internal Medicine, Japan Organization of Occupational Health and Safety, Kanto Rosai Hospital, Kawasaki, Japan. [110]Division of Infectious Diseases, Japanese Red Cross Musahino Hospital, Tokyo, Japan. [111]Ishikawa Prefectural Central Hospital, Kanazawa, Japan. [112]Kanagawa Cardiovascular and Respiratory Center, Yokohama, Japan. [113]Department of Respiratory Medicine, National Hospital Organization Tokyo Medical Center, Tokyo, Japan. [114]Department of Allergy, National Hospital Organization Tokyo Medical Center, Tokyo, Japan. [115]Division of Clinical Infectious Diseases, Department of Medicine, Showa University School of Medicine, Tokyo, Japan. [116]Department of Respiratory Medicine, Toyohashi Municipal Hospital, Toyohashi, Japan. [117]Keiyu Hospital, Yokohama, Japan. [118]Department of Respiratory Medicine, KKR Sapporo Medical Center, Sapporo, Japan. [119]Division of General Internal Medicine, Department of Internal Medicine, St. Marianna University School of Medicine, Kawasaki, Japan. [120]Department of Emergency and Critical Care Medicine, St. Marianna University School of Medicine, Kawasaki, Japan. [121]Japanese Red Cross Medical Center, Tokyo, Japan. [122]Matsumoto City Hospital, Matsumoto, Japan. [123]Department of Emergency and Critical Care Medicine, Faculty of Medicine, Kyushu University, Fukuoka, Japan. [124]Department of Infection Control, Fukuoka University Hospital, Fukuoka, Japan. [125]Department of Rheumatology, National Hospital Organization Hokkaido Medical Center, Sapporo, Japan. [126]Department of Respiratory Medicine, National Hospital Organization Hokkaido Medical Center, Sapporo, Japan. [127]Department of Emergency and Critical Care Medicine, National Hospital Organization Hokkaido Medical Center, Sapporo, Japan. [128]NHO Kanazawa Medical Center, Kanazawa, Japan. [129]Department of Internal Medicine, Division of Respiratory Medicine, School of Medicine, Nihon University, Tokyo, Japan. [130]Musashino Red Cross Hospital, Musashino, Japan. [131]Division of Respiratory Medicine, Social Welfare Organization Saiseikai Imperial Gift Foundation, Inc., Saiseikai Kumamoto Hospital, Kumamoto, Japan. [132]Department of Respiratory Medicine, Nagoya University Graduate School of Medicine, Nagoya, Japan. [133]Department of Internal Medicine, Fukuoka Tokushukai Hospital, Kasuga, Japan. [134]Respiratory Medicine, Fukuoka Tokushukai Hospital, Kasuga, Japan. [135]Department of Endocrinology, Hematology and Gerontology, Chiba University Graduate School of Medicine, Chiba, Japan. [136]Department of Emergency and Critical Care Medicine, Chiba University Graduate School of Medicine, Chiba, Japan. [137]National Hospital Organization Kumamoto Medical Center, Kumamoto, Japan. [138]Division of Infectious Diseases and Respiratory Medicine, Department of Internal Medicine, National Defense Medical College, Tokorozawa, Japan. [139]Sapporo City General Hospital, Sapporo, Japan. [140]Department of Internal Medicine, Tokyo Saiseikai Central Hospital, Tokyo, Japan. [141]Department of Pulmonary Medicine, Tokyo Saiseikai Central Hospital, Tokyo, Japan. [142]National Hospital Organization Kanagawa Hospital, Hadano, Japan. [143]Department of Respiratory Medicine, Fujisawa City Hospital, Fujisawa, Japan. [144]Uji-Tokushukai Medical Center, Uji, Japan. [145]Fukuoka Tokushukai Hospital, Kasuga, Japan. [146]Department of Infectious Disease, NHO Kyushu Medical Center, Fukuoka, Japan. [147]Department of Respirology, NHO Kyushu Medical Center, Fukuoka, Japan. [148]Division of Respirology, Rheumatology, and Neurology, Department of Internal Medicine, Kurume University School of Medicine, Kurume, Japan. [149]Ome Medical Center, Ome, Japan. [150]Research Institute for Diseases of the Chest, Graduate School of Medical Sciences, Kyushu University, Fukuoka, Japan. [151]Department of Medicine and Biosystemic Science, Kyushu University Graduate School of Medical Sciences, Fukuoka, Japan. [152]Daini Osaka Police Hospital, Osaka, Japan. [153]Department of Emergency and Critical Care Medicine, Faculty of Medicine, University of Tsukuba, Tsukuba, Japan. [154]Department of Hematology, Faculty of Medicine, University of Tsukuba, Tsukuba, Japan. [155]Department of Nephrology, Faculty of Medicine, University of Tsukuba, Tsukuba, Japan. [156]Department of Cardiovascular Surgery, Faculty of Medicine, University of Tsukuba, Tsukuba, Japan. [157]Division of Pulmonary Medicine, Department of Medicine, Tokai University School of Medicine, Isehara, Japan. [158]Department of Anesthesiology and Intensive Care Medicine, Kyoto Prefectural University of Medicine, Kyoto, Japan. [159]Department of Infection Control and Laboratory Medicine, Kyoto Prefectural University of Medicine, Kyoto, Japan. [160]Department of Respiratory Internal Medicine, St Marianna University School of Medicine, Yokohama-City Seibu Hospital, Yokohama, Japan. [161]KINSHUKAI Hanwa The Second Hospital, Osaka, Japan. [162]Emergency and Disaster Medicine, Gifu University School of Medicine Graduate School of Medicine, Gifu, Japan. [163]School of Health Sciences, Asahi University, Gifu, Japan. [164]Department of Respiratory Medicine, Tokyo Medical University Hospital, Tokyo, Japan. [165]JA Toride Medical Hospital, Toride, Japan. [166]Okayama Rosai Hospital, Okayama, Japan. [167]Himeji St. Mary's Hospital, Himeji, Japan. [168]Emergency and Critical Care, Niigata University, Niigata, Japan. [169]Emergency and Critical Care Center, National Hospital Organization Kyoto Medical Center, Kyoto, Japan. [170]National Hospital Organization Tokyo Hospital Hospital, Kiyose, Japan. [171]Fujioka General Hospital, Fujioka, Japan. [172]Department of General Medicine, School of Medicine, International University of Health and Welfare Shioya Hospital, Yaita, Japan. [173]Department of Pharmacology, School of Pharmacy, International University of Health and Welfare Shioya Hospital, Ohtawara, Japan. [174]Department of Respiratory Medicine, International University of Health and Welfare Shioya Hospital, Ohtawara, Japan. [175]Department of Clinical Laboratory, International University of Health and Welfare Shioya Hospital, Ohtawara, Japan. [176]Department of General Medicine, School of Medicine, International University of Health and Welfare Shioya Hospital, Ohtawara, Japan. [177]Department of Cardiology, Pulmonology, and Nephrology, Yamagata University Faculty of Medicine, Yamagata, Japan. [178]Division of Respiratory Medicine and Allergology, Department of Medicine, School of Medicine, Showa University, Tokyo, Japan. [179]Department of Pulmonary Medicine, Fukushima Medical University, Fukushima, Japan. [180]Kansai Electric Power Hospital, Osaka, Japan. [181]Division of Infectious Diseases, Kumamoto City Hospital, Kumamoto, Japan. [182]Department of Respiratory Medicine, Kumamoto City Hospital, Kumamoto, Japan. [183]Department of Emergency and Critical Care Medicine, Tokyo Metropolitan Police Hospital, Tokyo, Japan. [184]Department of Respiratory Medicine, Gunma University Graduate School of Medicine, Maebashi, Japan. [185]National Hospital Organization Saitama Hospital, Wako, Japan. [186]Tokyo Medical University Ibaraki Medical Center, Inashiki, Japan. [187]Department of Internal Medicine, Kiryu Kosei General Hospital, Kiryu, Japan. [188]Department of Pulmonary Medicine and Oncology, Graduate School of Medicine, Nippon Medical School, Tokyo, Japan. [189]Division of Respiratory Medicine, Tsukuba Kinen General Hospital, Tsukuba, Japan. [190]Division of Respiratory Medicine, Department of Internal Medicine, Toho University Ohashi Medical Center, Tokyo, Japan. [191]Division of Anesthesiology, Department of Surgery Related, Kobe University Graduate School of Medicine, Kobe, Japan. [192]Institute of Medical Science, The University of Tokyo, Tokyo, Japan.

## Methods

### Analysis of UKB data

UKB data, accessed based on application ID 135122, were used as the primary discovery cohort, unless stated otherwise (Supplementary Note 1). Individual-level data analyses were conducted within the UKB Research Analysis Platform (RAP).

**Extraction of high-quality EBV reads.** All individuals with available GS data ($n = 490,293$)[24] were included in the initial stage of analysis (UKB cohort). During the process of the project, 208 individuals (0.04%) withdrew their consent from UKB, explaining slightly lower sample counts in some follow-up analyses ($n = 490,085$). DNA extraction, library preparation, sequencing and alignment have been described elsewhere[68,69] and are summarized in Supplementary Note 2. Reads mapping to the EBV genome (NC_007605.1) were accessed in CRAM files (field 24048), which had been previously generated by aligning fastq data to a GRCh38 graph genome (including the contig chrEBV) and were extracted using samtools (v1.20). Only read pairs where both forwards and reverse reads, respectively, mapped to NC_007605.1, were retained. Within pairs, reads were removed if they had more than 20 soft-clip bases, less than 120 bases matching the reference or were duplicates (see Supplementary Note 2). Finally, if at least one read of a read pair remained, this was counted as one EBV read. We also generated a similar dataset for HHV7 for the purpose of comparison, as described in Supplementary Note 6.

**Quality control.** We calculated the fraction of individuals with EBV reads per library preparation plate (field 32056). Fifty-one plates had excessively high proportions of EBVread+ individuals, probably due to contamination, and were excluded (Extended Data Fig. 1 and Supplementary Note 1). We also excluded individuals with low GS data quality (field 32064), sex chromosome aneuploidies (array-based genotyping data, field 22019) or discrepancies between reported and genetic sex (fields 31, 22001), resulting in the UKB QC cohort. For analyses limited to EUR ancestry, individuals were selected based on UKB field 22006. Applying a high-quality set of common genotyped variants for principal component analysis and for regenie step 1 (Supplementary Note 9) led to the exclusion of an additional 180 individuals (Supplementary Note 2), leaving $n = 403,014$ individuals for analyses (UKB EUR cohort).

We also generated a subcohort of the UKB QC cohort, comprising individuals for whom serology measurements were available (UKB serology cohort; $n = 9,281$, based on data field 23053). In this cohort, EBV seropositivity was defined based on the detection of at least two out of four EBV-related IgG antibodies (EA-D, ZEBRA, EBNA-1 and VCA-p18), as previously suggested[44,70].

**Processing of covariates.** For individuals of the UKB EUR cohort, potentially important confounders of EBV read detection were retrieved based on ref. 71, including information on sequencing, technical aspects, blood composition and demographics. On the basis of the SNOMED associations identified in the AoU cohort, we additionally considered smoking status, pack years of smoking, number of cigarettes smoked per day (or previously smoked in cigar and/or pipe smokers) and number of weekly alcoholic drinks. Extracted values were processed to finally obtain transformed values for each covariate (Supplementary Note 4). Correlated covariates were identified by calculating Pearson correlations (one of each pair removed if correlation > 0.7; $n = 4$). Together with covariates age × sex and age × age, this resulted in 28 potential covariates, which were further reduced to a final set of 18 covariates by forwards and backwards selection with Bayesian information criterion (Supplementary Table 4 and Supplementary Note 4).

**Immunosuppressive and EBV-associated conditions.** Immunosuppressed individuals were identified as those reported with (1) taking immunosuppressive drugs (including glucocorticoids) at the time of visiting the UKB assessment centre (verbal interview, field 20003; $n = 9,681$), or (2) HIV infection (UKB fields 130204, 130206, 130208, 130210 and 130212; $n = 230$). Individuals affected by EBV-associated diseases were identified based on self-reporting in the assessment centre, *International Statistical Classification of Diseases and Related Health Problems, 10th revision* codes or codes for operative procedures (OPCS4). Full lists are given in Supplementary Table 23.

**Association analyses.** For common variants and HLA alleles, the main GWAS on EBVread+ was conducted with two-step regenie (v3.2.4)[72], on related individuals of the No immune supp. cohort. Common variants have been previously imputed using the Haplotype Reference Consortium and UK10K haplotype resource[22] (UKB field 22828; 29,865,259 variants with info-score > 0.8; 481 individuals lacked imputation data). Individual HLA alleles were obtained from field 22182, based on previous imputation with HLA*IMP:02 (ref. 73). Variants were included if they had a predicted minor allele count of ≥ 25. Non-classical HLA alleles were not included due to the lack of established standards for imputing these alleles. For compatibility with regenie step 2, the provided dosages were converted to plink2 pgen-files. In the statistical analysis, the 18 selected covariates and 20 principal components were used as covariates, and saddle point approximation was applied to account for case–control imbalance (see Supplementary Fig. 7 and Supplementary Notes 9 and 10).

For conditional analysis of HLA alleles, we applied a forwards-stepwise regression approach to identify HLA alleles that independently associate with the trait, based on the following procedure: (1) initial single-variant test for all HLA alleles as described in common variants and HLA alleles. (2) Iterative conditioning: repeat the following process: (i) Identify the allele with the lowest $P$ value from the previous step; (ii) add this allele to the alleles to condition on; and (iii) run the conditioned association analysis (regenie v3.2.4). Step 2 was repeated until the most significant allele in the current iteration had a $P$ value greater than the commonly used genome-wide significance threshold of $5 \times 10^{-8}$.

For epistatic analyses, the lead variants of the three top non-MHC loci for EBVread+ were tested for interaction with conditionally independent HLA alleles, based on data from non-related individuals of the UKB no immune supp. cohort ($n = 304,523$ with complete data). Likelihood-ratio tests (LRTs; 1 d.f.) were used, comparing an additive logistic regression model with a model that additionally included an interaction term between the non-MHC SNP and the HLA allele (see Supplementary Note 11). LRT $P$ values were Bonferroni corrected for multiple testing.

For rare variants, RVAS_gene was performed as described for common variants (identical phenotypes, and covariates, same procedure for regenie step 1), but based on exome variants and annotations as provided by the UKB[74] (field 23158; Supplementary Note 9). This resulted in a slight reduction of the overall sample number (based on no immune supp. cohort; $n = 54,259$ EBVread+ cases and $n = 293,834$ EBVread− controls). For regenie step 2, SKAT-O was used as a test (parameter: '--vc-tests skato') and we restricted the analysis to rare variants with an alternative allele frequency below 1% (parameter '--vc-maxAAF 0.01'). The following definitions of variant pathogenicity (masks) were used: (1) M1: predicted loss-of-function variants; (2) strong coding: variants from (1) and likely deleterious missense variants; (3) medium coding: variants from (2) plus possibly deleterious missense variants; and (4) all coding variants from (3) plus likely benign missense variants (Supplementary Note 9). Overall, this analysis comprised rare variants in 18,796 protein-coding genes.

**Additional case–control definitions and subcohorts.** In addition to the main analysis of EBVread+, in which we compared individuals with EBV reads (1–18) to those without any EBV reads (0), we generated modified case–control definitions. These included GWAS analyses

of 0 versus 1 read counts, 0 versus 2–18 read counts, and a 'within EBVread+' analysis comparing individuals with 1 read count versus 2–18 read counts. We also performed sex-restricted analyses, that is, on male or female participants only. Sample numbers are provided in Supplementary Table 12.

## Analysis in the AoU cohort

We used release 8 (C2024Q3R3) of the AoU Research Program, which included array and GS data from blood-based DNA samples of 365,931 individuals (AoU cohort). The AoU resource, including data generation, processing and quality control of genomic data, is described in ref. 23 and accompanying documents.

**Generation of EBV read data and cohort from GS data.** First, EBV reads were extracted from CRAM files as described for UKB participants. At the individual level, we restricted our analyses to unrelated individuals with plausible time points of DNA sampling (between 11:00 and 23:59), without mismatch between reported and genetic sex and who were not flagged as population outliers ('flagged samples') in accompanying documents (AoU QC cohort, $n = 336,123$). For population-specific analyses, precomputed genetically predicted population backgrounds were used, which assigned each individual to one of six continental populations (Extended Data Fig. 3; see 'Genomic research data quality report').

**Phenome-wide association analysis of EBVread+.** We retrieved individuals from the AoU QC cohort who had electronic health record data available. For SNOMED concept IDs annotated in 250 or more individuals ($n = 11,111$), associations with the presence of EBV reads was tested as follows: we first applied logistic regression models with the presence of EBV reads as outcome, the presence of a SNOMED ID as predictor, and included age, sex, age × sex and 16 precomputed principal components as covariates. In a second step, we also included HIV and smoking status as covariates (see Supplementary Note 3). P values were calculated using LRT.

**Replication of associated loci and HLA alleles.** Detailed information of variant sets, generation of principal components, imputation and quality control of HLA alleles are described in Supplementary Fig. 8 and Supplementary Notes 3 and 12. Association analyses were performed in the EUR subcohort of AoU using regenie (v2.0.2), but without using step 1. We selected similar covariates as in the analysis within UKB, that is, sex, age, age × sex, mean sequencing coverage, hour as well as the week and time of biosample collection, nicotine usage, sequencing site and 20 principal components. However, certain covariates (including blood count traits) are not directly available in AoU and therefore could not be included (see Supplementary Table 4), which prevented a meta-analysis between UKB and AoU (Supplementary Note 4).

## Validation cohorts

Two non-UKB/non-AoU cohorts were used for validation (Supplementary Note 13). For each of them, EBV reads were extracted from short-read GS data, in analogy to the analysis in UKB:

(1) Validation 1, qPCR. This cohort was recruited to study ACE inhibitor-induced angiooedema and consisted of 110 participants for whom GS data and DNA samples were available (blood or saliva derived[25]). To quantify EBV viral load, qPCR was performed on 72 individuals, including all EBVread+ and a random subset of EBVread− individuals, using the clinically validated GeneProof EBV PCR Kit (TaqPath Menu, Applied Biosystems; four technical replicates per sample), with the target gene *EBNA1*.

(2) Validation 2, qPCR and RNA-seq. Partially overlapping subsets of JCTF participants with SARS-CoV-2 infection[26,75] were used for qPCR for EBV viral load and reanalysis of RNA-seq data ($n = 1,010$), respectively. GS was obtained from whole-blood-derived DNA. For qPCR ($n = 262$

individuals, 3 technical replicates each), an in-house developed qPCRs assay was run, targeting *EBNA1* (Supplementary Note 13; sequences available on request). Full-length RNA-seq data were reanalysed for the expression of 94 EBV genes. In short, reads were aligned against the GRCh38 reference genome, which included the EBV sequence NC_007605.1, and EBV transcripts were quantified using RSEM (v1.3.0). Given the high prevalence of EBVread+ in the JCTF subcohort, we investigated whether a more severe COVID-19 disease course drives EBVread+, but did not observe a strong effect (Supplementary Table 5 and Supplementary Note 5).

## Genetic risk loci associated with EBVread+

**Annotation of non-MHC risk loci.** Regional association plots were generated with LocusZoom[76] (see Supplementary Fig. 9 and Supplementary Note 14). Genome-wide significance was defined as $P < 5 \times 10^{-8}$, and independent risk loci were defined in FUMA (v1.6.3)[48], based on 1000Gv3 (EUR population; $r^2$ threshold of 0.6) lead SNPs (merging distance of 250 kb). For each locus, we reported (1) closest gene (based on distance of lead SNP to the transcription starting site); (2) linkage disequilibrium genes (that is, genes located within associated region, defined through variants with $r^2 > 0.2$ to lead variant); (3) eGenes from GTEx (based on Adult GTEx v10, with genome-wide significant single-tissue eQTL effects ($P < 5 \times 10^{-8}$) in any tissue); and (4) V2G scores from Open Targets (v22.10; based on a cut-off > 0.1). To identify pleiotropic effects of lead variants, we retrieved from OpenTargets (v22.10) all traits at $P < 0.005$ that were reported in either GWAS Catalog, UKB or FinnGen. To identify potential targets for drug repurposing, approved drugs (clinical phase IV) targeting the identified genes were retrieved from OpenTargets (v25.3).

To investigate for potential regulatory effects on transcription in specific blood cell types, lead variants (or proxies thereof; $r^2 > 0.7$ based on 1000Gv3, EUR subset) were retrieved from the OneK1K dataset[52], and reported eQTLs with FDR < 0.05 in the original dataset.

**Generation of credible SNP sets.** Fine mapping for each non-MHC locus was performed with SuSie (sum of single effects regression)[33], using 1-Mb window size (except for 12q24.12_SH2B3 (3 Mb) and 5q31.1_SLC22A5 (5 Mb) due to extended local linkage disequilibrium). Linkage disequilibrium matrices were generated from the imputed genotype data of the unrelated UKB EUR cohort (see above; $n = 339,539$; without principal component filter) using plink2 (v2.0.0-a.6). Coding variants within the credible SNP sets (cumulative PIP: 0.95) were annotated using Ensembl Variant Effect Predictor[77] (VEP; release 113), ClinVar (version June, 2023)[78] and AlphaMissense prediction scores[79].

**Correlation of effect sizes.** We retrieved association statistics for lead variants at the 27 genome-wide significant non-MHC risk loci as well as for 54 conditionally independent HLA alleles, from additional GWAS. These included four different case–control definitions based on EBV read counts and female-only or male-only GWAS (see above), as well as from three external datasets: memory B cell absolute counts (GCST90001407 (ref. 42); no MHC data available) and EBV antibody titres[44]. We additionally calculated effect sizes at these loci for HHV7read+ (Supplementary Note 6) and recalculated effect sizes for main EBVread+ GWAS (0 versus 1–18) using different sets of covariates (Supplementary Note 4). Variants in linkage disequilibrium with the lead variant were used if they increased the overlap between datasets. We then calculated the correlation of effect sizes (betas) using Spearman's correlation for non-MHC risk variants as well as HLA alleles. For HHV7, we investigated loci with potentially shared causal variants using coloc (v5.2.3)[80] in R (v4.4.2).

## Gene-level analyses

Gene-based association testing as well as enrichment analyses were conducted using MAGMA (v1.08)[45], using default settings unless stated otherwise. Variants were assigned to 19,736 genes using the MAGMA

gene boundaries Ensembl v102 file (excluding the extended MHC region as previously suggested[71] (25–36 Mb)), and a window of 10 kb upstream and 1.5 kb downstream. Gene sets for IEIs were defined based on literature[15] (*n* = 14 genes) or the IEI classification (available at https://iuis.org/committees/iei/, accessed 6 May 2025; *n* = 456 genes available in our data). Gene ontology biological processes (*n* = 7,743 terms) and tissue types (*n* = 54, GTEx v8) were provided by FUMA (v1.6.3).

Cell-type identification was performed using scDRS[51] (v1.0.3) and single-cell RNA-seq data from the 1M-scBloodNL project, published by the sc-eQTLGen consortium[50] (samples processed with Genomics (v3); broader level of cell-type annotations with 10 cell types; see Supplementary Note 15). Following data processing using the Seurat package (v5.2.1) in R (v4.3.2), 37,033 cells annotated to 8 cell types remained for the scDRS analysis. The top 1,000 EBVread[+] MAGMA genes and their *z*-scores were used as weights in the scDRS analysis, with otherwise default parameters. Subsequent group analyses (that is, cell-type association and heterogeneity) were conducted with default parameters. Multiple testing correction of *P* values for the number of cell types was performed using Benjamini–Hochberg procedure.

### GRS analyses in UKB

**Analysis of polygenic contribution.** To study the joint contribution of common variants associated with EBVread[+] to EBV-associated diseases within EUR individuals of the UKB no outlier cohort, we generated an independent base cohort plus additional target cohorts, which encompassed (1) individuals with EBV-associated diseases (see results, data fields given in Supplementary Table 23), or (2) individuals for whom serology data were available. We additionally removed all individuals that were related to any other individual of the target or the base cohort. Individuals of the target cohorts were required to pass all filters applied to the UKB no outlier cohort, except that individuals within the top 1% EBV read counts were kept.

For SNP-based GRS, common variant association analysis was performed within the base cohort as described above, except that only autosomal and genotyped variants with a minor allele count > 25 were used for regenie step 2 (*n* = 739,066). Polygenic risk scoring was performed on non-ambiguous SNPs using PRS-CS (v1.0.0) in combination with a linkage disequilibrium matrix derived from EUR individuals of the 1000 genomes project[81]. To generate separate GRS for the MHC regions and the non-MHC regions, the summary statistics generated using the base dataset were split to only contain the required regions (that is, MHC region and non-MHC regions). Scores of individuals of the target cohort were obtained using the score function of plink (v1.90b6.21). For HLA allele-based GRS, GRS based on imputed HLA alleles were calculated by fitting a multivariable logistic regression model to the base dataset, where EBVread[+] was the outcome. As predictors, we used the 18 covariates and the 20 principal component (see above), plus 178 HLA alleles that had a minor allele frequency > 0.1%. After model fit, the coefficient estimates of the 178 HLA alleles were retrieved. Scores for individuals in target cohorts were generated by multiplying HLA allele dosages with coefficient estimates and summing-up these values. To obtain MHC-I-specific or MHC-II-specific scores, we calculated the GRS using the same coefficient estimates, but considered only the HLA alleles belonging to the respective MHC class. All risk scores were normalized to means of 0 and standard deviations of 1 within the combined target cohorts.

**Evaluation of GRS performance.** To evaluate GRS performance, we used logistic regression models, where the group membership was the outcome (for example, EBVread[+] versus EBVread[−] or serology cohort versus a cohort of individuals with EBV-associated disease). As predictors, we used age, sex, age × sex, 20 principal components (base model) or the predictors of the base model as well as the respective GRS (GRS model). We then calculated Nagelkerke $R^2$ for the base and the GRS models, where variability of Nagelkerke $R^2$ estimates was evaluated

using bootstrapping (*n* = 1,000). To test for statistical significance, we compared base models and GRS models using LRT.

### GRS analyses in AoU

**Transferability of GRS.** For GRS analyses across biobanks and populations, we used our EBVread[+] summary statistics from the UKB no immune supp. cohort, as the base dataset. SNP-based GRS were calculated based on 1,509,024 genotyped variants within AoU, which had a Hardy–Weinberg equilibrium $P \geq 1 \times 10^{-10}$ and a variant-level missingness < 0.05 in each continental ancestry. Polygenic risk scoring was performed using PRS-CS CS and plink (v1.9.0-b.7.7) as described above. To obtain HLA-based GRS, we used coefficient estimates of the 178 HLA alleles from the UKB (see above) and multiplied them with the estimated dosage for each HLA allele in AoU. Mapping of HLA allele names between HLA*IMP:02 and HLA-TAPAS was performed manually, which resulted in a successful mapping for 166 of 178 alleles (no clear mapping for one MHC-I allele and 11 MHC-II alleles).

**Phenome-wide association of EBVread[+] GRS with PheCodes.** We used the software package PheTK (v0.1.47)[82] to assign PheCodes (v1.2) to individuals of the AoU QC cohort (EUR subset, *n* = 189,658). Individuals were considered as having a certain PheCode if the PheCode was annotated at least twice to the individual, as suggested by PheTK. We used logistic regression models with the presence of a PheCode as outcome and GRS as predictors, respectively. Age, sex, age × sex and 20 population-specific principal components were used as covariates. *P* values were calculated using LRT comparing models with and without the respective GRS. To comply with AoU publishing guidelines, we have only reported PheCodes annotated to more than 20 individuals, that do not supply count-related data and only give proportions when all underlying groups contain more than 20 individuals. PheCodes (*n* = 1,751) were compliant with these parameters.

### 2SMR analysis

**Selection of outcome traits.** We retrieved publicly available summary statistics for EUR ancestry cohorts for (1) known EBV-associated diseases: multiple sclerosis case–control[83], multiple sclerosis severity[84], Hodgkin disease[85], non-Hodgkin lymphoma[85], systemic lupus erythematosus[86] and rheumatoid arthritis[87]; and (2) candidate diseases based on significant PheWAS results: hypothyroidism[85], type 1 diabetes[88] and inflammatory bowel disease[89]. Of note, none of the nine outcome GWAS included samples from UKB. We also used 'red hair colour'[90] (including UKB) as a negative control outcome. Summary statistics were retrieved from the GWAS Catalog except for multiple sclerosis severity, where summary statistics were shared by the authors. Further details are provided in Supplementary Table 20. Analyses were performed in R (v4.5.0) using the packages ieugwasr (v1.0.3) and TwoSampleMR (v0.6.15)[91,92].

**Selection of instrumental variables.** First, we applied quality control on exposure and outcome GWAS summary statistics, retaining autosomal and non-duplicate variants with minor allele frequency > 0.01 and info-score > 0.8. Linkage disequilibrium-independent genome-wide significant variants from the GWAS on EBVread[+] (the exposure) were identified using linkage disequilibrium clumping (ld_clump function of the ieugwasr package, standard parameters) and harmonized with the outcome GWAS summary statistics. The remaining genome-wide significant variants of the exposure were not suspicious of weak-instrument bias ($I^2$ statistic = 0.99). We further used Steiger filtering[92] to exclude potentially invalid instruments (that is, variants showing stronger associations with the outcome versus with the exposure). Together, this resulted in a reduction of the number of variants available for 2SMR.

**2SMR.** 2SMR was performed using four methods[93]: the inverse variance-weighted estimator, MR-Egger, weighted median and weighted mode. For traits in which the exposure–outcome associations were

nominally significant in all four estimators and reached test-wide significance ($P < 0.05/9$) in two out of four (Supplementary Table 20), we applied the outlier-robust and pleiotropy-robust estimators MR-RAPS[94] and MR-PRESSO[95]. Outcomes that were also significant in these two additional tests were then subjected to further sensitivity analyses, specifically heterogeneity (Cochran's $Q$ statistic[96]), pleiotropy tests (for example, MR-Egger intercept[97]) and leave-one-out analyses. For these outcomes, we also performed 2SMR after excluding variants in the MHC region.

## Ethics declaration

This study used de-identified data from the UKB and AoU, which were accessed through the respective computing platforms. UKB has approval from the North West Multi-centre Research Ethics Committee (MREC) as a Research Tissue Bank. This approval means that researchers do not require separate ethical clearance and can operate under the Research Tissue Bank approval. The data collection of the AoU Research Program was conducted under centralized Institutional Review Board (IRB) approval, with informed consent being obtained from the participants. Further ethical approvals were obtained from the Ethics Committee of the Medical Faculty Bonn (no. 101/16; for analysis of validation cohort 1) and by the ethical committees of the affiliated institutes (Keio IRB approval 20200061, Osaka University IRB approval 734-14 and University of Tsukuba IRB approval H29-294) for the JCTF.

## Reporting summary

Further information on research design is available in the Nature Portfolio Reporting Summary linked to this article.

## Data availability

All genetic and phenotype data from the biobanks are available upon application and approved data access from the UKB study and AoU projects. All interested readers will be able to access the data in the same manner that the authors did, including usage of the UKB Research Analysis Platform and AoU workbench environments for the analysis of de-identified individual-level data. GWAS summary statistics are available through the GWAS Catalog (main EBVread⁺ GWAS: GCST90809298; GWAS for additional case–control definitions: GCST90809299–GCST90809306). The main EBVread⁺ GWAS is also available at Locus-Zoom (https://my.locuszoom.org/gwas/968885/?token=b74ac20f6a d94a88a5ea27b6ac214645). All additional data are either provided in Supplementary Tables or through Zenodo[98]. Data access for the two validation cohorts is described in their respective original articles[25,26]. Complementary data used for secondary analyses were obtained from: OneK1K (https://onek1k.org/), eQTLgen 1M-scBloodNL (https://www.eqtlgen.org/sc/datasets/1m-scbloodnl-dataset.html), GTEx (https://www.gtexportal.org/home/), OpenTargets (https://platform.open-targets.org/), IUIS (https://iuis.org/committees/iei/), GWAS Catalog (https://www.ebi.ac.uk/gwas/) and the International Multiple Sclerosis Genomics consortium (https://imsgc.net/).

## Code availability

Code to extract and quantify EBV reads is archived in the repository EBVread-extraction (https://github.com/Ax-Sch/EBVread-extraction), and the analysis code can be found within the repository EBVread_data_analysis (https://github.com/Ax-Sch/EBVread-data-analysis). Archived versions of the repositories are also available via Zenodo[98]. The analyses relied on several publicly available tools, which can be accessed as follows and which are referenced within the respective Methods section, including versions: R (https://cran.r-project.org/), tidyverse (https://github.com/tidyverse), Python (https://www.python.org/), snakemake (https://snakemake.github.io/), nextflow (https://www.nextflow.io), bwa-mem2 (https://github.com/bwa-mem2/bwa-mem2),

samtools/bcftools (https://www.htslib.org), IGV (https://igv.org), plink/plink2 (https://www.cog-genomics.org/plink/2.0/), FlashPCA (https://github.com/gabraham/flashpca), regenie (https://github.com/rgcgithub/regenie), HLA-TAPAS (https://github.com/immunogenomics/HLA-TAPAS), LocusZoom (http://locuszoom.sph.umich.edu/), FUMA (https://cncr.nl/research/fuma_gwas/), MAGMA (https://cncr.nl/research/magma/), SuSie (https://github.com/stephenslab/susieR), Seurat (https://satijalab.org/seurat/), SeuratDisk (https://github.com/mojaveazure/seurat-disk), Ensembl VEP (https://www.ensembl.org/info/docs/tools/vep/), coloc (https://github.com/cran/coloc), scDRS (https://github.com/martinjzhang/scDRS), PheTK (https://github.com/nhgritctran/PheTK), PRScs (https://github.com/getian107/PRScs), ieugwasr (https://github.com/MRCIEU/ieugwasr), TwoSampleMR (https://github.com/MRCIEU/TwoSampleMR), MR-RAPS (https://github.com/qingyuanzhao/mr.raps), MR-PRESSO (https://github.com/rondolab/MR-PRESSO), linkage disequilibrium score regression (https://github.com/bulik/ldsc), STAR (https://github.com/alexdobin/STAR) and RSEM (https://github.com/deweylab/RSEM).

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

**Acknowledgements** We thank A. Vyvers and S. Heilmann-Heimbach for critical discussions; H. Schrage for laboratory support; C. Schmäl for manuscript editing; the AoU participants for their contributions, without whom this research would not have been possible; the US National Institutes of Health's AoU Research Program for making available the participant data examined in this study; the International Multiple Sclerosis Genetics Consortium for providing summary statistics on multiple sclerosis; and the granted access to the Bonna and Marvin HPC clusters hosted by the University of Bonn. K.B., A.-K.P., M.M.N. and K.U.L. are members of the Excellence Cluster ImmunoSensation[3] (EXC2151), which is funded by the German Research Foundation (DFG) under 390873048. A.S. was supported by the BONFOR program of the Medical Faculty of the University of Bonn (O-149.0134). Y. Okada was supported by JSPS KAKENHI (25H01057); AMED (JP24km0405217, JP24ek0109594, JP24ek0410113, JP24kk0305022, JP223fa627001, JP223fa627002, JP223fa627010, JP223fa627011, JP22zf0127008, JP24tm0524002, JP24wm0625504 and JP24gm1810011); JST Moonshot R&D (JPMJMS2021 and JPMJMS2024); Takeda Science Foundation; Ono Pharmaceutical Foundation for Oncology, Immunology, and Neurology; Bioinformatics Initiative of Osaka University Graduate School of Medicine; Institute for Open and Transdisciplinary Research Initiatives; Center for Infectious Disease Education and Research (CiDER); and Center for Advanced Modality and DDS, Osaka University, and RIKEN TRIP initiative (AGIS). H.N. was supported by AMED (JP24tm0524008, JP22fk0108510 and JP22fk0108537), JST PRESTO(JPMJPR21R7) and Takeda Science Foundation. UKB analyses were performed under application 135122. This work uses data provided by patients and collected by the NHS as part of their care and support, and we thank the participants and coordinators of the UKB study. This publication was supported by the Open Access Publication Fund of the University of Bonn.

**Author contributions** A.S., M.M.N. and K.U.L. conceptualized the study. A.S., T.M.A., F.S.D., L.F., S.K.H. and A.T.D. provided the methodology. A.S., T.M.A., F.S.D., L.F., S.R., M.S. and Y. Ogawa performed the formal analysis. C.M.M., Japan COVID-19 Task Force, A.J.F., H.N. and Y. Okada provided resources. Y. Ogawa, H.N. and E.C.B. conducted the investigation. A.S., F.S.D., Y. Ogawa, L.F., H.N., E.C.B. and K.U.L. wrote the original draft of the manuscript. T.M.A., S.R., M.S., C.M.M., S.K.H., A.J.F., A.T.D., A.-K.P., K.B., Y. Okada and M.M.N. reviewed and edited the manuscript. A.S., T.M.A., F.S.D., Y. Ogawa, L.F., S.R., M.S. and E.C.B. performed the visualization. A.S., M.M.N., Y. Okada and K.U.L. provided supervision. A.S., Y. Okada and K.U.L. acquired funding.

**Competing interests** K.U.L. is a co-founder of LAMPseq Diagnostics. A.T.D. is a co-founder of Peptide Groove, a company that commercializes statistical HLA-typing approaches. A.-K.P. (institution) has received speaker honoraria from Biogen, Novartis, Roche and UCB. M.M.N. has received fees for membership in the advisory board from HMG Systems Engineering, for membership in the Medical-Scientific Editorial Office of the Deutsches Ärzteblatt, for review activities from the European Research Council, and for serving as a consultant for EVERIS Belgique SPRL in a project of the European Commission (REFORM/SC2020/029); and receives salary payments from Life & Brain and holds shares in Life & Brain. The remaining authors declare no competing interests.

**Additional information**
**Correspondence and requests for materials** should be addressed to Axel Schmidt or Kerstin U. Ludwig.

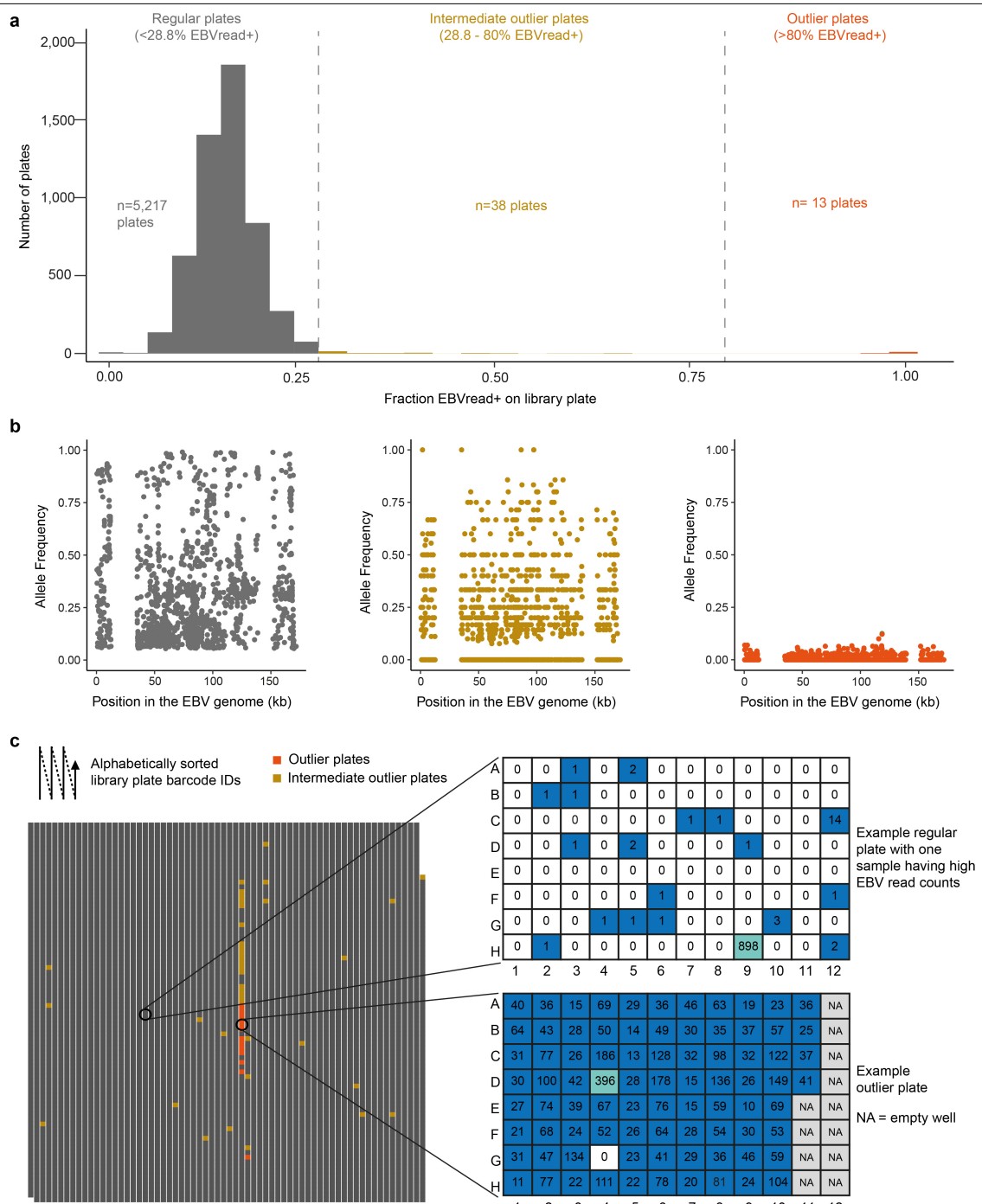

**Extended Data Fig. 1 | Identification of library plate outliers in UK Biobank.**
**a**) The distribution of EBVread+ individuals per 96-well library plate was used to identify 51 library plates with high rates of EBVread+ individuals. Different colors are assigned to regular plates (grey), intermediate plates (above 28.8%; i.e., 2 standard deviations of mean, up to 80%; yellow), and outlier plates (orange). **b**) Allele frequencies of common EBV variants were determined in each group, based on aggregated reads (see Supplementary Note 2). **c**) When library plate barcode IDs were sorted alphabetically, the outlier plates with very high rates of EBVread+ were clustered, along with some of the intermediate plates, further supporting potential batch effects. Two representative library plates (one regular plate containing one sample with high EBV read count, and one outlier plate with high number of EBVread+ individuals) are highlighted by circles (left) and shown as examples (right), with plate-positions colored as follows: white: no EBV read; blue: at least one EBV read, light blue: highest EBV read count on plate. NA = empty positions.

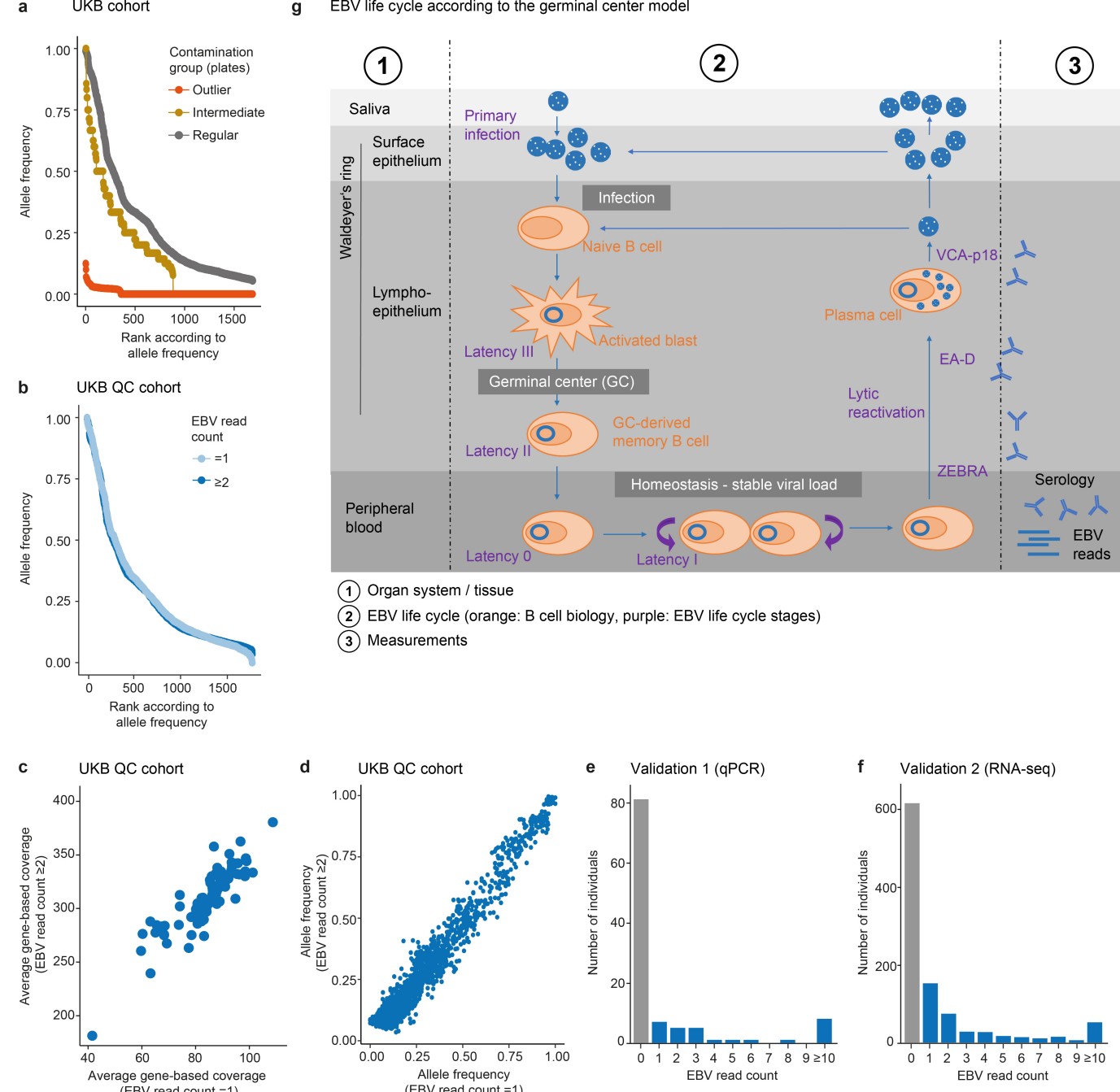

**Extended Data Fig. 2 | Technical validation of GS-based EBV-reads.** Multiple lines of evidence support that individuals with EBV read count =1 are true positives. **a**) Plotted rank distribution of the allele frequency data (AF; described in Extended Data Fig. 1) illustrate separate trajectories for contaminated outlier (orange) or regular plates (grey). **b-d**) Comparison of individuals with EBV read counts =1 and ≥2, regarding rank distribution of allele frequencies (b), average coverage values across 94 EBV genes (c) and allele frequencies of common EBV

variants (d). Note that rank distribution plotted in (b) is different from outlier plates in (a). **e, f**) Distribution of EBV read counts in individuals of the two validation cohorts, i.e. validation 1 ((e); GS of 110 European individuals; 26.3% EBVread+) and validation 2 ((f), JCTF, GS of 1,010 East Asian individuals; 39.2% EBVread+). Samples from these cohorts were used for qPCR analyses as shown in Fig. 1. **g**) Proposed model of EBV life cycle and the correlation with EBVread+ as determined in our study. Figure adapted from ref. 7, Springer Natute Ltd.

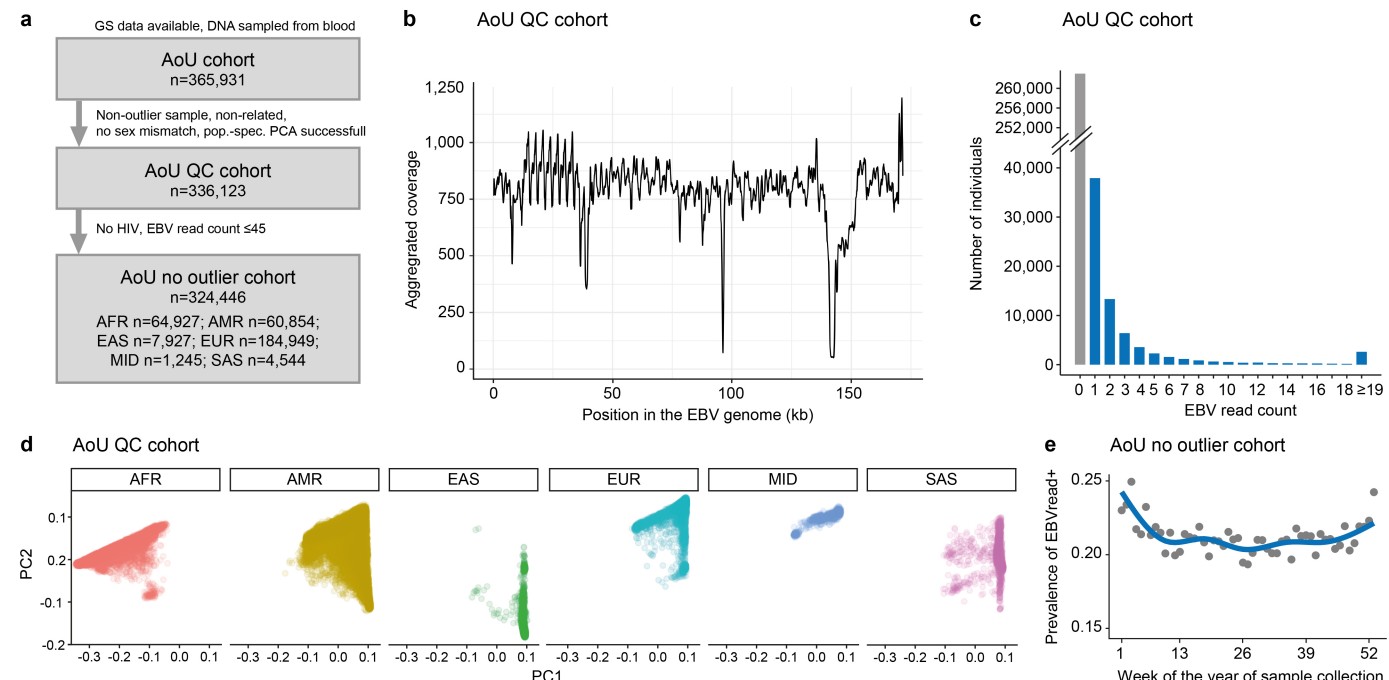

**Extended Data Fig. 3 | Analysis of EBVread+ in All of Us cohort. a)** Flow chart showing the generation of different All of Us (AoU) cohorts that were used for subsequent steps of the analysis. Details are provided in the Methods section. The number of individuals in each of the six population backgrounds are given for the AoU no outlier cohort. **b)** Cumulative read coverage across the EBV genome (line smoothed, 500 bp rolling window), for all individuals of the AoU QC cohort. **c)** Number of individuals within EBV read count groups. **d)** The first two principal components (PCs) of common genotypes as provided by AoU are displayed for each of the six population backgrounds. **e)** EBVread+ in relation to the week of the year in which blood samples were collected. Abbreviations: AFR: African, AMR: Admixed American, EAS: East Asian, EUR: European, MID: Middle Eastern, SAS: South Asian.

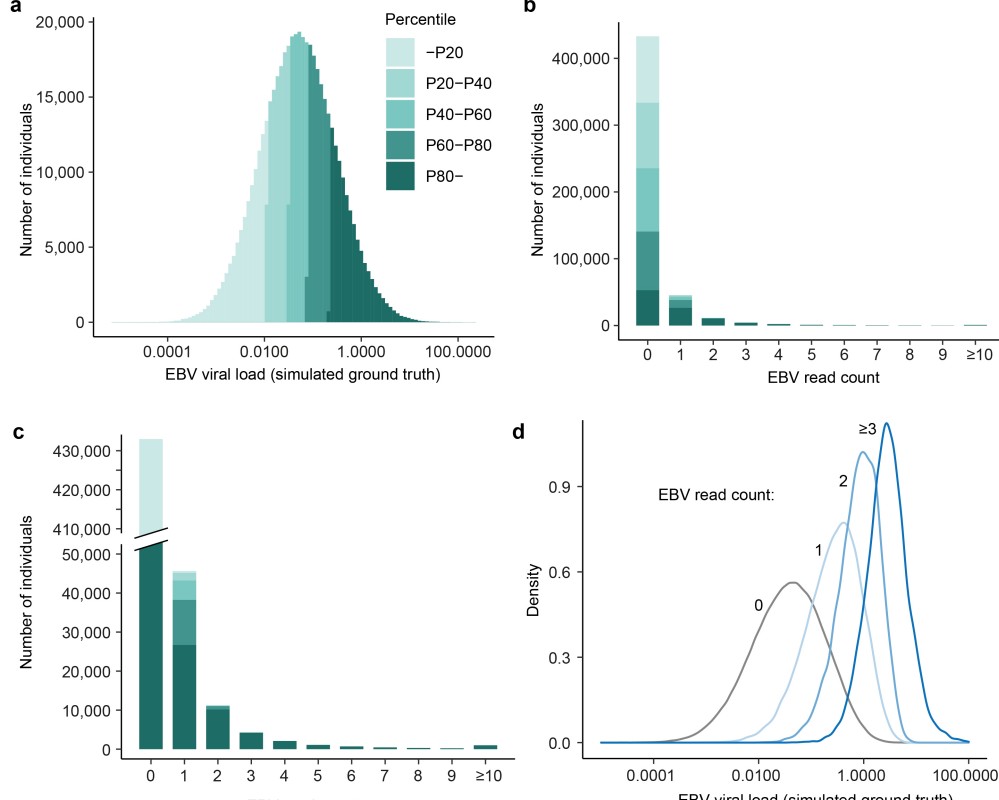

**Extended Data Fig. 4 | Simulation of EBV viral load and the generation of EBV-reads from genome sequencing (GS). a)** We modeled EBV viral load in 500,000 individuals using a log-normal distribution ("ground truth"). This distribution was informed by prior observations on measured viral load in HIV[27]. The x-axis reflects theoretical units, which could be transferred to biological units if quantified standards were available. The numbers of individuals per unit are plotted on the y-axis. Individuals were assigned to 20% percentile groups (color coded). **b)** From the simulated viral loads, we sampled "reads" for each individual, using a binomial distribution, with 400 million trials (approximately the average number of sequencing reads available per individual in our study). The probability values for successfully drawing EBV reads were proportional to the viral load of the respective individual. The success rate of the binomial distribution as well as the parameters of the log-normal distribution shown in **a)**, were manually fitted to match the observed read count distribution in our data (cf. Fig. 1c). **c)** is a zoom in on panel b. **d)** Within our simulation, EBV viral load increased with increasing numbers of observed EBV reads (reads of 3 or above are aggregated).

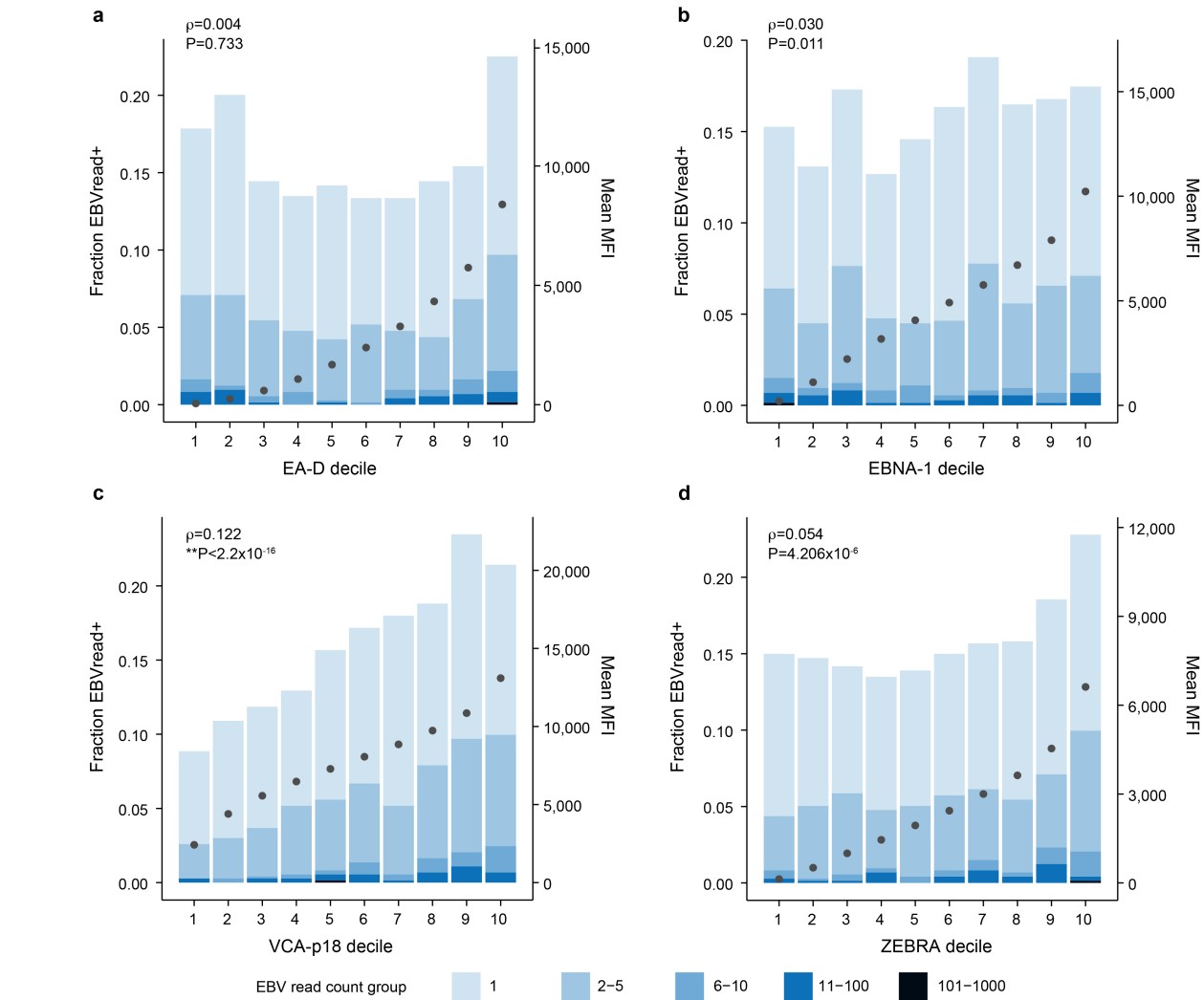

**Extended Data Fig. 5 | Correlation of GS-based EBV-reads and individual measurements of four EBV-related antibodies.** 7,338 individuals of the UKB EUR cohort were seropositive for EBV, based on the detection of at least 2 out of 4 EBV-related antibodies. For IgG antibodies against **a)** EA-D, **b)** EBNA-1, **c)** VCA-p18 and **d)** ZEBRA, individuals were assigned to deciles based on median fluorescence intensity (MFI), and the deciles were tested for significant correlation with the 0/1-encoded EBVread+ status using Spearman correlation coefficients (ρ) and two-sided P-values (P). **Exact *P* value not available due to computational limits. Bar sizes indicate overall fractions of EBVread+ individuals within the respective deciles (left y-axis), with colors representing different EBV read count groups (legend on bottom). Dots represent average MFI values per decile (right y-axis labels). Analysis was performed on raw measurement data, without adjustment for covariates.

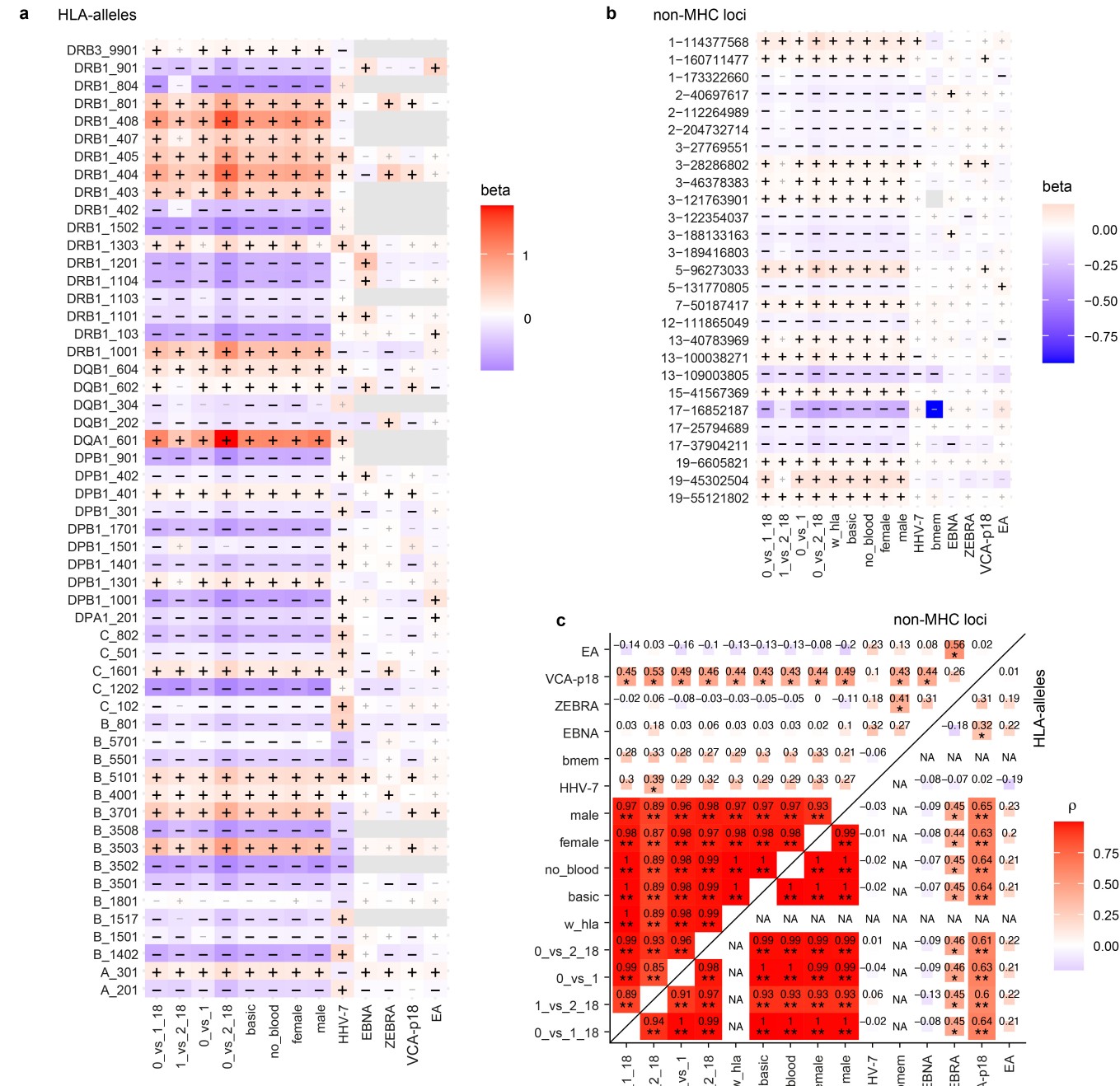

**Extended Data Fig. 6 | Correlation of effect sizes of EBVread+ GWAS lead variants.** We compared the results of the main analysis (EBVread+: controls: 0 EBV reads vs. cases: 1–18 EBV reads) to (i) three different case-control definitions based on EBV read counts in UKB (1 read vs. 2–18 reads, 0 reads vs. 1 read, 0 reads vs. 2–18 reads), (ii) different sets of covariates in UKB ("basic", "no blood", "w_hla" see Supplementary Table 4). (iii) male- and female-specific analyses in UKB, (iv) HHV7read+ in UKB and (v) external GWAS: memory B cell absolute counts (from GCST90001407[30], no MHC-data), and EBV-related serology data for four antibodies (Methods). Point estimates of effect sizes (beta) are color-coded for **(a)** 54 conditionally independent HLA-alleles and

**(b)** the lead variants at 27 non-MHC loci. +/− illustrates the direction of effect and +/− font is faded grey if the individual association was not nominally significant. Grey boxes indicate missing data. In **(c)**, Spearman's correlations and respective *P* values (two-sided) were calculated between all pairs of traits, based on effect sizes and alleles. Correlation coefficients (ρ) are shown for HLA-alleles (bottom triangle) and non-MHC loci (upper triangle). * P < 0.05; ** P < 0.001; NA, not available. Numbers of individuals as well as association statistics used to calculate correlation of effect sizes are given for each trait in Supplementary Table 12.

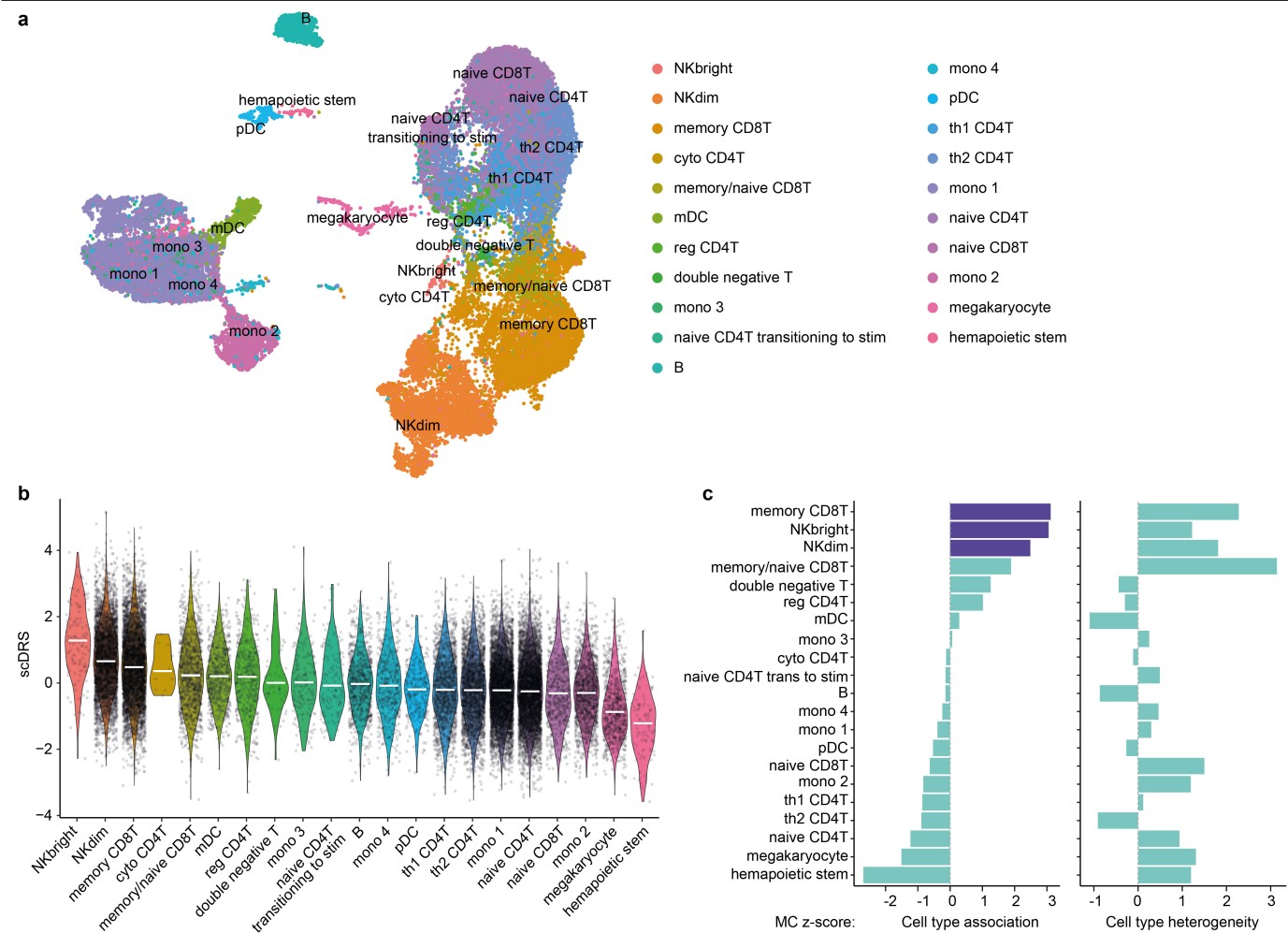

**Extended Data Fig. 7 | scDRS analysis as in Fig. 3, using more fine-grained cell annotation. a**) UMAP representation plot of the 1M-scBloodNL data (v3) colored according to cluster labels of cell type annotation level 2. **b**) Distribution of normalized single-cell disease relevance scores (scDRS) across cell types of annotation level 2, sorted according to the largest average score. White bars indicate the median scDRS. **c**) Results of the Monte Carlo (MC)-based statistical inference of cell type association (left) and within-cell type heterogeneity (right) with scDRS based on EBVread+. Bar colors represent significance, with purple color indicating a multiple comparison-adjusted false discovery rate (FDR) < 0.05. Further information is provided in Methods and Fig. 3.

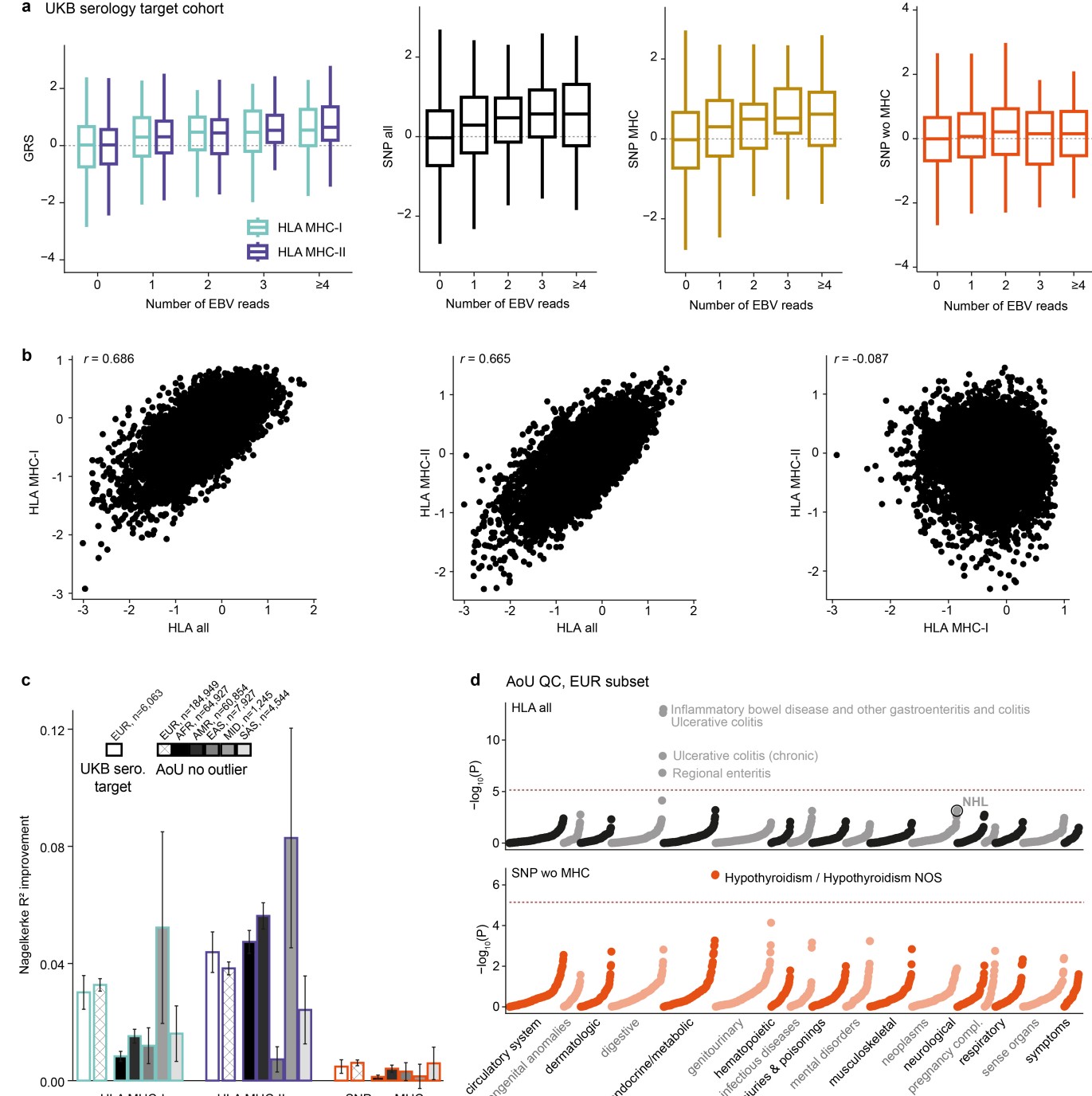

**Extended Data Fig. 8 | Prediction of EBVread+ using Genetic Risk Scores and Phenome-wide association studies (PheWAS). a)** Individuals from the UKB serology target cohort (n = 6,063, unrelated) were stratified according to EBV read counts in the GS data, and the distributions of specific GRSs within these groups are shown as boxplots (median (thick line), 25th and 75th percentile (box) and largest/smallest value no further from the box than 1.5 times the interquartile range (whiskers)). **b)** Scatter plots of individual GRSs (indicated by the axis labels) illustrate the correlation structures between HLA all, HLA MHC-I, and HLA MHC-II. Only weak correlation was observed between the GRS encompassing HLA-alleles from MHC class I vs those from MHC class II

(Pearson correlation). **c)** In analogy to Fig. 4e, improvements in Nagelkerke's $R^2$ relative to base models within the UKB serology target cohort (extreme left bar of each GRS category), and the six continental ancestries in AoU for the indicated GRSs are given (abbreviations as in Extended Data Fig. 3). Sample sizes are provided within the panel and error bars correspond to standard deviation derived from n = 1,000 bootstrap iterations. **d)** PheWAS using HLA all and SNP wo MHC in analogy to Fig. 4f. In addition to annotating all significant PheWAS associations, the association identified in UKB with NHL (HLA all) is encircled. P values were calculated using logistic regression, with adjustment for covariates and likelihood ratio tests (Methods).

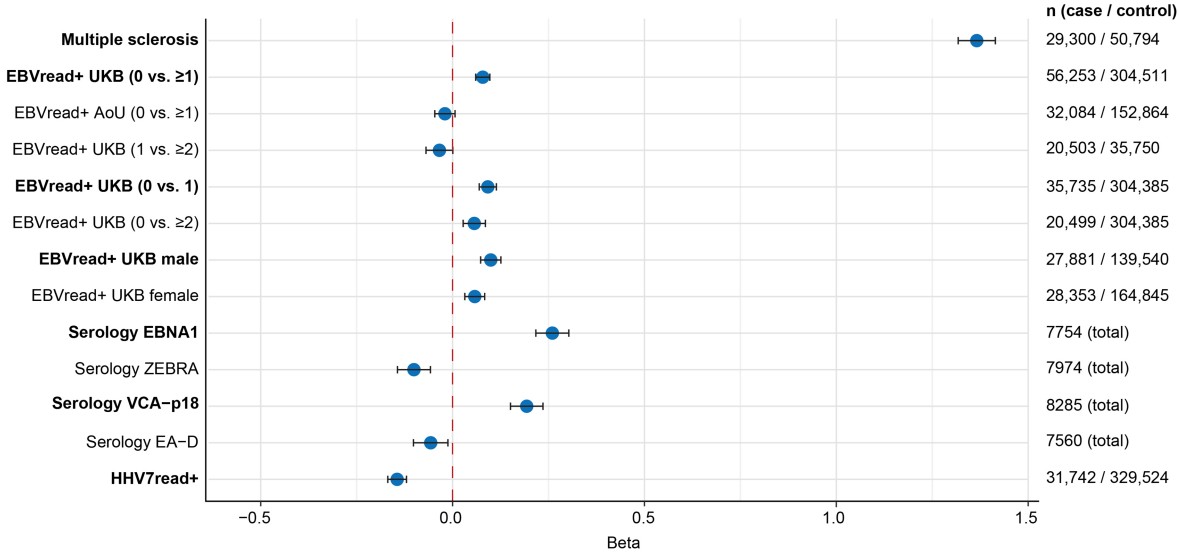

**Extended Data Fig. 9 | Analysis of HLA-DRB1\*15:01 associations across datasets.** Point estimates of effect sizes (beta) and 95% confidence intervals (unadjusted) for the major multiple sclerosis (MS) risk allele HLA-DRB1\*15:01, across different EBV-associated traits and multiple case-control definitions in the present study. For comparisons, we also extracted these values from a recent MS GWAS in which HLA-alleles were present[36]. Highlighted in bold are analyses in which the association of HLA-DRB1\*15:01 reached genome-wide significance. Sample sizes are given within the panel with case-control numbers for binary traits and total numbers for continuous traits.

| | |
|---|---|

# Reporting Summary

## Statistics

For all statistical analyses, confirm that the following items are present in the figure legend, table legend, main text, or Methods section.

| n/a | Confirmed | |
|---|---|---|
| ☐ | ☒ | The exact sample size (*n*) for each experimental group/condition, given as a discrete number and unit of measurement |
| ☐ | ☒ | A statement on whether measurements were taken from distinct samples or whether the same sample was measured repeatedly |
| ☐ | ☒ | The statistical test(s) used AND whether they are one- or two-sided<br>*Only common tests should be described solely by name; describe more complex techniques in the Methods section.* |
| ☐ | ☒ | A description of all covariates tested |
| ☐ | ☒ | A description of any assumptions or corrections, such as tests of normality and adjustment for multiple comparisons |
| ☐ | ☒ | A full description of the statistical parameters including central tendency (e.g. means) or other basic estimates (e.g. regression coefficient) AND variation (e.g. standard deviation) or associated estimates of uncertainty (e.g. confidence intervals) |
| ☐ | ☒ | For null hypothesis testing, the test statistic (e.g. *F*, *t*, *r*) with confidence intervals, effect sizes, degrees of freedom and *P* value noted<br>*Give P values as exact values whenever suitable.* |
| ☒ | ☐ | For Bayesian analysis, information on the choice of priors and Markov chain Monte Carlo settings |
| ☒ | ☐ | For hierarchical and complex designs, identification of the appropriate level for tests and full reporting of outcomes |
| ☐ | ☒ | Estimates of effect sizes (e.g. Cohen's *d*, Pearson's *r*), indicating how they were calculated |

*Our web collection on statistics for biologists contains articles on many of the points above.*

## Software and code

Policy information about availability of computer code

| Data collection | We analysed existing genome sequencing data and phenotypic information from the two large biobanks UKBiobank (UKB) and All of Us (AoU). Reads mapping to the EBV or HHV7 genome were extracted from GS-derived CRAM files within the frameworks snakemake (v.7.32.4; UKB) or nextflow (v25.04; AoU). In particular, read extraction and filtering was performed using samtools (UKB: v1.20, AoU: v1.22), alignment of reads to the HHV7 genome with bwa-mem2 (v2.2.1). Viral reads were visualized with IGV (v2.12.3). |
|---|---|
| | Common variants for association analyses were retrieved from data field 22828 (imputed genotypes, bgen format) in UKB, and from GS-based variant call plink2 files for AoU. Rare variants (UKB, exome sequencing data) were retrieved from data field 23158. HLA-alleles were retrieved from field 22182 in UKB, or imputed based on genotype data (plink1 file format) using HLA-TAPAS for AoU (https://github.com/immunogenomics/HLA-TAPAS). Phenome-wide analyses were conducted in AoU based on SNOMED-IDs as provided in the AoU database. |
| | Code to extract and quantify EBV-reads is archived in the repository EBVread-extraction (https://github.com/Ax-Sch/EBVread-extraction), and analysis code can be found within the repository EBVread_data_analysis (https://github.com/Ax-Sch/EBVread-data-analysis). Archived versions of the repositories are also available via zenodo (10.5281/zenodo.18417294 ). |

| Data analysis | Data analysis was performed within the frameworks R (v4.3.2 and higher, i.e. UKB: v4.4.0, AoU: v4.5.0; tidyverse v2.0.0) and python (UKB: v3.9.16, AoU: v3.10.16). Variant level genetic data was analyzed and handled with plink (UKB: v1.90b7.4; and v1.90b6.21 in GRS-scoring; AoU: v1.90b6.22 and v1.9.0-b.7.7 in GRS-scoring), plink2 (v2.0.0-a.6), bcftools (UKB: v1.20, AoU: v1.12) and FlashPCA (v2.0). Association analysis was performed using Regenie v3.24 (UKB) or v2.0.2 (AoU). Covariates were analyzed using R libraries 'splines' (v4.4) and 'MASS' (v7.3-6). Typing of HLA-alleles was performed using kourami (v0.9.6). Downstream analyses and visualizations were performed using FUMA (v1.6.3), MAGMA (v1.08), OpenTargets (v22.10, v25.3), Ensembl VEP (v113.0), coloc (v.5.2.3), scDRS (v1.0.3), seurat (v5.2.1), PRS-CS (v1.0.0), PheTK (v0.1.47), PheCodes (v1.2), ieugwasr (v1.0.3), TwoSampleMR (v0.6.15), MR-PRESSO (https://github.com/rondolab/MR-PRESSO), MR_RAPS (arXiv:1801.09652), LDSR (v1.0.1), SuSie (v0.15.4), Seurat Disk (v0.0.0.9021). Processing of RNAseq-data: STAR (v2.5.3a) and RSEM (v.1.3.0). See "Data collection" for custom code availability. |
|---|---|

For manuscripts utilizing custom algorithms or software that are central to the research but not yet described in published literature, software must be made available to editors and reviewers. We strongly encourage code deposition in a community repository (e.g. GitHub). See the Nature Portfolio guidelines for submitting code & software for further information.

# Data

Policy information about availability of data

All manuscripts must include a data availability statement. This statement should provide the following information, where applicable:
- Accession codes, unique identifiers, or web links for publicly available datasets
- A description of any restrictions on data availability
- For clinical datasets or third party data, please ensure that the statement adheres to our policy

All genetic and phenotype data from the biobanks are available upon application and approved data access from the UK Biobank study and AllofUs projects. All interested readers will be able to access the data in the same manner that the authors did, including usage of the UKB Research Analysis Platform and AoU workbench environments for the analysis of de-identified individual-level data. GWAS summary statistics are available through the GWAS catalog (GCST GCST90809298–GCST90809306). All additional data are either provided in Supplementary Tables or through Zenodo (10.5281/zenodo.18417294), including the custom code repository as .zip files (EBVread-data-analysis-main_1.0.zip; EBVread-extraction-main_1.0.zip). Complementary data used for secondary analyses were obtained from: OneK1K (https://onek1k.org/), eQTLgen 1M-scBloodNL ( https://www.eqtlgen.org/sc/datasets/1m-scbloodnl-dataset.html), GTEx (https://www.gtexportal.org/home/), OpenTargets (https://platform.opentargets.org/), IUIS ( https://iuis.org/committees/iei/), GWAS Catalogue (https://www.ebi.ac.uk/gwas/), the International Multiple Sclerosis Genomics consortium (https://imsgc.net/). Data access for the two validation cohorts is described in their respective original articles (references PMIDs: 35923707 (validation cohort-1), 39317738 (validation cohort-2)).

# Research involving human participants, their data, or biological material

Policy information about studies with human participants or human data. See also policy information about sex, gender (identity/presentation), and sexual orientation and race, ethnicity and racism.

| Reporting on sex and gender | Analyses were performed including males and females. Sex was used as covariate in many statistical analyses and was determined based on information reported in UKB or AoU via genetic inference. |
|---|---|
| Reporting on race, ethnicity, or other socially relevant groupings | Within the UKB dataset, individuals of European population were selected based on information given in UK Biobank field 22006, i.e. self reported 'White British' ethnicity and very similar genetic ancestry based on a principal components analysis (PCA) of the genotypes. In AoU, we used precomputed genetically predicted population backgrounds, which assigned each individual to one of six continental populations (African, Admixed American, East Asian, European, Middle Eastern, South Asian; see AoU Genomic Research Data Quality Report). |
| Population characteristics | We used all individuals of the UKB and AoU projects for whom blood-based genome sequencing data were available following quality control. Discovery analyses were performed in the UKB-QC-cohort (mixed ancestry, mean age of 56.5 years, 54.2% being female). In the AoU-QC cohort used for replication analyses, ancestry groups were as follows: Africans (mean age 49.3 years, 57.1% female), Admixed Americans (44.5 years, 65.0% female), East Asian (43.45 years, 63.0% female), European (55.5 years, 59.1% female), Middle Eastern (44.4 years, 52.5% female), South Asian (40.8 years, 52.8% female). Validation cohorts comprised European (validation-1: 67.3 years, 48.6% female) or EastAsian (validation-2: 60.8 years, 48.1% female) individuals. |
| Recruitment | No recruitment of participants was performed in this study as we used existing cohorts and datasets. More information on the recruitment for UKB and AoU can be found in previously published work. UKB: Sudlow et al., 2015 PLOS Medicine; Bycroft et al., 2018, Nature; Halldorsson et al., 2022 Nature; UKB WGS consortium, 2025, Nature; AoU: All of Us Research Program Investigators, 2019 NEJM; AoU Genomic Program, 2024, Nature. Recruitment of the two validation cohorts has been described previously (references PMIDs: 35923707 (validation cohort-1), 39317738 (validation cohort-2)). |
| Ethics oversight | This study used de-identified data available from UKB and AoU, which were accessed through their respective platforms. UKB has approval from the North West Multi-centre Research Ethics Committee (MREC) as a Research Tissue Bank (RTB). This approval means that researchers do not require separate ethical clearance and can operate under the RTB approval. The data collection of the AoU Research Program was conducted under centralized Institutional Review Board (IRB) approval, with informed consent obtained from participants. UKB Tier-3 data was accessed based on application-ID 135122. For the two validation cohorts, ethical approvals were obtained from the Ethics Committee of the Medical Faculty Bonn (no. 101/16; for analysis of validation cohort 1) and by the ethical committees of the affiliated institutes (Keio IRB approval 20200061, Osaka University IRB approval 734-14, University of Tsukuba IRB approval H29-294) for the JCTF. |

Note that full information on the approval of the study protocol must also be provided in the manuscript.

# Field-specific reporting

Please select the one below that is the best fit for your research. If you are not sure, read the appropriate sections before making your selection.

☒ Life sciences ☐ Behavioural & social sciences ☐ Ecological, evolutionary & environmental sciences

For a reference copy of the document with all sections, see nature.com/documents/nr-reporting-summary-flat.pdf

# Life sciences study design

All studies must disclose on these points even when the disclosure is negative.

| | |
|---|---|
| Sample size | We included all individuals of UKB and AoU for whom genome sequencing data from blood were available. For each analysis we maximized the number of used samples based on quality control measures, without a priori sample size calculation. The total sample size for EBV-read extraction after quality control was 486,315 individuals for UKB and 336,123 for AoU. For the validation cohorts, sample sizes were determined by the number of individuals in each study. |
| Data exclusions | Data were excluded during the study procedure for quality control reasons, in particular: low-quality genome sequencing data, outliers during library preparation, implausible covariates, missing data. All details are provided in the Methods section. |
| Replication | Whenever possible, we used one of the two biobanks for discovery, and the other one for replication. This is illustrated as Supplementary Note 1. Specifically, for the genetic data, we aimed to replicate the results of the main EBVread+ GWAS from UKB in 184,948 individuals of European ancestry from AoU. We observed nominal significance and consistent effect direction for 100 of 106 HLA alleles that could be matched across both datasets and for lead variants at 25 of the 27 non-MHC loci. Similar analyses were performed for a GRS generated in UKB, and for the non-genetic factors which were determined in AoU first and replicated in UKB. For non-genetic factors, all replication efforts between UKB and AoU were successful for phenotypes that were available and comparably assessed in both biobanks. |
| Randomization | This population-based study is observational, therefore randomization was not relevant for our work. We extensively test and correct for potential confounders using existing phenotype and metadata in UKB and AoU, which is described in detail in the Methods section. |
| Blinding | Blinding was not relevant for this study as experimental group assignment was not performed. |

# Reporting for specific materials, systems and methods

We require information from authors about some types of materials, experimental systems and methods used in many studies. Here, indicate whether each material, system or method listed is relevant to your study. If you are not sure if a list item applies to your research, read the appropriate section before selecting a response.

## Materials & experimental systems

| n/a | Involved in the study |
|---|---|
| ☒ | ☐ Antibodies |
| ☒ | ☐ Eukaryotic cell lines |
| ☒ | ☐ Palaeontology and archaeology |
| ☒ | ☐ Animals and other organisms |
| ☒ | ☐ Clinical data |
| ☒ | ☐ Dual use research of concern |
| ☒ | ☐ Plants |

## Methods

| n/a | Involved in the study |
|---|---|
| ☒ | ☐ ChIP-seq |
| ☒ | ☐ Flow cytometry |
| ☒ | ☐ MRI-based neuroimaging |

# Plants

| | |
|---|---|
| Seed stocks | NA |
| Novel plant genotypes | NA |
| Authentication | NA |

