## [Peer Review File · Nature]

Host control of persistent Epstein-Barr virus infection

Corresponding Author: Professor Kerstin Ludwig

Version 1:

Reviewer comments:

Referee #1

(Remarks to the Author)

Summary: The authors describe a study of non-genetic and genetic factors contributing to EBV DNA positivity in genome sequence data from two large biobanks (UKBB and All of Us). The authors demonstrate that EBV reads can be successfully recovered from a large percentage of biobank contributors. Non genetic factors including immune compromise, smoking, age and sex were correlated to EBV positivity. A GWAS approach demonstrated a strong genetic component to EBV positivity implicating a major role for HLA alleles and non-HLA immune processes in determining EBV positivity. Importantly, these signals were consistent across biobanks and polygenic risk scores were transferable across biobanks and ancestries. PheWAS and Mendelian randomization analysis showed overlap between the genetic contributors to EBV positivity and several autoimmune diseases with 2SMR analysis suggesting a causal effect of EBV positivity on type 1 diabetes. Overall this is an interesting and well executed study that improves our understanding of control of a common chronic viral infection and the overlap of these processes with additional immune disorders.

Comments:

1. The authors report consistent overlap of MHC and non-MHC associations between the UKBB and AoU cohorts but don't report on results of a genome-wide meta-analysis across both samples. Can the authors comment on if this analysis was performed and if any additional signals were uncovered?
2. Interestingly, in addition to identifying several strongly associated HLA alleles, the authors identify associations in key proteins involved in antigen processing. Did the authors perform any interaction or epistatic analysis between HLA alleles and these other proteins? I'm specifically thinking about interaction between HLA and ERAP alleles as was observed for ankylosing spondylitis [PMID 21743469]. This might provide additional information as to the function of various alleles and mechanism of immune evasion.
3. Perhaps I'm mis-interpreting but the analysis presented in figure 2b seems to suggest that individuals with only one detectable EBV read may be mis-classified in the beta_main analysis. If the effect size of the associations are roughly the same when using the EBV_One group as controls as with the full, non-EBV set of individuals, wouldn't this motivate either excluding the EBV_one group or including them with the full control set? Can the authors comment on if they are thinking the EBV_One set are the result of some mis-alignment or true positives with very low EBV levels?
4. Given the incomplete overlap in clinical data between UKBB and AoU, the methods for covariate selection were necessarily different. Can the authors comment on how these different sets of covariates may influence the results of the association analyses and the accuracy of the polygenic risk score predictions?

(Remarks on code availability)

I was able to download and run the provided code and obtain the expected output. I did not have an external set of genome sequences aligned with the EBV contig readily available so can't comment on non-test cases.

Referee #2

(Remarks to the Author)

Schmidt et al. used whole genome sequencing data from 486,315 individuals from the UK Biobank (UKB) and 336,123 individuals from All of Us (AOU) to identify genetic and non-genetic factors involved in controlling latent EBV infection. Short reads mapping to the EBV genome were identified in 16.2% of UKB samples and 21.8% of AOU samples. The authors assert that the detection of EBV short reads (as few as one EBV short read) represent a robust surrogate measure for high EBV viral loads. GWAS comparing individuals who were EBV read-positive vs -negative identified strong associations at the MHC loci as well as 27 loci external to the MHC. The presence of EBV reads was associated with immunosuppression, winter sample acquisition, and inflammatory diseases including rheumatoid arthritis, inflammatory bowel disease, and type 1 diabetes. In addition, gene-set enrichment analysis implicated NKT and CD8+T cells in the control of EBV infection. Status as an active smoker, an environmental risk factor for EBV-associated cancers and autoimmunity, was identified as a modifiable factor that influences EBV read detection and, the authors argue, by extension, EBV viral loads. The strength of this study lies in the impressive size of the two cohorts, the identification of novel non-HLA genes associated with control of EBV infection, and correlates of EBV DNA detection/load with various disease indications.

Comments:

1. The claim that the detection of short reads mapping to the EBV genome is a robust surrogate for higher viral loads should be validated with an additional measurement (e.g. EBV DNA detection in PBMC, serum/plasma, or saliva)—although this may not be available for UKB or AOU, it could be performed with a smaller cohort. The simulation of EBV viral loads (Extended data figure 1) is interesting, but should be validated experimentally, rather than rely on comparisons with HIV viral loads, especially because there is not a clear relationship between EBV reads and seropositivity. Moreover, the viral lifecycles of retroviruses and herpesviruses are quite different and may not be directly comparable.
2. EBV reads from 50 preparation plates were discarded because they had an unexpectedly high number of individuals who were EBV positive, and it was determined that these plates might be contaminated. What was the threshold for determining that plates had too many positive individuals (what percent +)? Whole genome sequence analysis should be able to identify whether the EBV in those plates had the identical EBV sequence—which would support contamination. The lower overall % positivity (16.2% vs 21.8%) in the UKB vs AOU cohort suggest that the threshold used for the plates in the UKB cohort likely missed/underestimated many individuals with poor control of EBV infection. How might the inclusion of these high copy number of EBV alter the correlation studies and overall conclusions?
3. 61.9% of EBVread+ individuals had only one read—with a maximum 27,649 reads detected. The authors state that there was uniform coverage of the EBV genome, but the distribution of EBV genes detected should be shown to see if there was a technical bias in terms of which EBV gene was detected in the short read analysis. The effect of size for lead variants is shown based on antibody abundance against read association analysis (Extended data figure 2), but this does not indicate the relative abundance of each read.
4. The authors discuss non-HLA risk loci associated with high EBV viral load during latency. However, without determining EBV gene expression, it is unclear if the reads detected are associated with EBV latency or lytic reactivation.
5. EBV detection was increased in males vs females, but autoimmune diseases associated with EBV are more frequently observed in females. Are there genes associated with males who are EBV read positive, not immunocompromised, and without inflammatory disease? Perhaps these genes could be associated with a protection from EBV mediated-autoimmune disorders despite poor EBV control.
6. No association was found for EBV reads and the major MS risk allele HLA-DRB1*15:01, leading the authors to conclude this risk allele does not alter EBV viral load. What are the implications of this relating to the known correlation of EBV seropositivity and MS, and viral copy number and seropositivity, if any?

Minor points.

1. Figures are difficult to read and not very easy to understand. They could be higher quality. Especially Fig 4g. Font colors are hard to read, some figure fonts are fuzzy/low quality.
2. Line 443. Interesting that MS did not score as most significant in the Mendelian Randomization and predictive causal relationship, while several other autoimmune diseases RA, CI, and T1D.
3. Line 455. "our study is the first to .." This may not be completely correct, so best to avoid "first".
4. The conclusion that genes affecting T-cell and NK-cells are most significant regulating EBV load is already well established, so this is not a very novel finding, although it does support the methodological approach.
5. Extended Data Figure 2, panel e. "not available". Maybe a PDF issue or else this panel be replaced/ removed?

(Remarks on code availability)

Referee #3

(Remarks to the Author)

This manuscript by Schmidt, et al. aims to uncover the extent of Epstein Barr Virus (EBV) viral load in human subjects, and determine genetic and non-genetic host factors that control EBV infection. Their approach exploits the fact that EBV reads will be present as a by-product of genome sequencing, owing to the integration of EBV DNA during latent infection. Using the UK Biobank sequencing data as a discovery cohort, the authors map these reads to the EBV genome, thus acquiring measure of viral load, and proceed to analyze these data with respect to host genotype and phenotypes, uncovering a number of robust associations. These are mostly replicated by the authors using the All of Us cohort. Overall, this is a meticulously preformed and well-written study that yields a number of novel insights. The authors are exceedingly careful at each step in their analysis, and provide substantial support for each of their conclusion. This paper will be of interest as it greatly improves our understanding of factors underlying EBV control, with important implications for

autoimmunity for example; as well as serving as an important jumping off point for any number of new research avenues. I do have a few issues that I think need to be addressed however:

-My major concern involves the characterization of samples with a single read mapping to EBV as "EBV+," which corresponds to 61.9% of all EBV+ samples. This strikes me as a concerningly low threshold, despite the fact that the authors do make an effort to eliminate read mis-mapping (e.g., requiring forward and reverse reads, requiring 120 matching bases). Despite these precautions, the percentage of EBV+ samples (16.2%) is higher than cited for immunocompetent individuals detected using qPCR (11%). If these single-read samples were not included, by my calculation the EBV+ samples would come in at 6.2%, which seems equivalently plausible. It would be good to have more understanding of what we are looking at with these samples with only a single EBV read. For example, is there a specific region of the EBV genome where we are seeing the preponderance of these reads mapped? Or is their distribution similar to that for samples with higher numbers of reads? Is it possible to validate any of these samples with qPCR?

-Likewise, in the All of Us cohort, EBV+ is given as 21.8%; how many of those samples were single-read only?

-The above concerns also extend to the analysis performed examining samples with read counts of 2 or more vs those with only a single read. Again, one wonders whether the clear differences here were due to viral load, as the authors assert, or the fact that some of the single read samples were in fact EBV-.

-The results for the MHC region are extremely interesting and important, especially given the role of this region in autoimmune disease. It's unclear to me whether any non-classical HLA variants within the MHC were examined after accounting for the classical HLA associations, could the authors please clarify?

-Regarding the imputation of HLA for the All of Us cohort, it would be helpful to see quality metrics, particularly as some of the associated alleles are rather rare. It's especially surprising that many of these are given at 4-field resolution. Was this reduced for analysis (e.g., to protein-coding resolution only)?

- As currently written, it is not clear how the conditional analysis of HLA alleles was conducted. Please clarify the writing or add the relevant description to the Methods section.

- Lines 215-220: Were all 54 independent HLA alleles that withstood conditional analysis testing identified in these comparisons?

- In the UKB GRS analysis, the conclusion for the HLA_MHC I-MS result described in the text does not seem to agree with the interpretation of Figure 4. Once HLA-A*02:01 is accounted for, the signal disappears. This is not surprising, since A*02:01 is a well-known protective marker for MS. However, Figure 4 only shows the significant association, without mention that it is entirely driven by A*02:01. Further, when DRB1*04:04 was accounted for in the RA model, there was no signal loss, suggesting that there may be additional/novel genetic factors driving the association. The authors might want to consider finding a way to distinguish between these two outcomes in Figure 4.

(Remarks on code availability)

- The authors should be congratulated for their extremely thorough and comprehensive documentation of code used in this manuscript. It is very accessible and easy to follow. It includes informative descriptions and workbooks for interfacing with UKB and AoU. The context and commentary provided for each script are very helpful. Example data is provided. This is one of the more reproducible and well-documented workflows this reviewer has ever seen.

- My one comment would be just to go back through and double check that the input data files for each script are all properly described and/or have the correct repository location, with respect to the structure that will be hosted on Github. Most scripts do have this, but I found a few examples of some that may have been missed, see below:

- EBVread_host_control-0.1/Finemapping_Coloc/finemapping_coloc_forestplot.R:

```
o sumstats <- read_tsv("C:/Users/srichter/Documents/Promotion/sumstats/zero_vs_1_18_sumstats.tsv.gz")
```

o Could not find this file. Please include this file or a description that demonstrates the formatting/data that it should contain

- EBVread_host_control-0.1/UKB-RAP_notebooks/2_2_coverage_plot/coverage_plot.R:

```
o per_b_coverage<-read_tsv(file="/mnt/project/2_1_count_ebv_reads/per_base_coverage.tsv", col_names=c("CHR", "POS", "COV"))
```

o "mnt/project/" does not match the repository structure as provided

Referee #4

(Remarks to the Author)

I co-reviewed this manuscript with one of the reviewers who provided the listed reports.

(Remarks on code availability)

Referee #5

(Remarks to the Author)

Schmidt et al describe a novel study investigating genetic and non-genetic factors that contribute to latent EBV infection control, by undertaking large-scale analyses of the UK Biobank and All of US. The authors provide their rationale for this study, namely that inefficient control of latent EBV infection is linked to the development of EBV-associated diseases, including EBV-associated malignancies, however there is a lack of data that can be used to study latent infection, and BioBanks often miss the viral load data. Some studies include serological data, but this only helps to identify previous infections or current lytic infections; it is difficult to use for latency. Instead, the authors use short-reads mapping to the EBV genome as surrogate measurements for increased viral load in latent infections. Overall, the study is novel and robust, and

the paper is well-written, providing context both for the rationale of analyses and for the results. However there are still some major points that need to be addressed, please see below:

Detection of EBV-reads using GS data from UKB and AoU

1) The authors use the presence of EBVread+ as a proxy of latent EBV infection. The question is how they can distinguish between latent infection and a primary or lytic infection in the cohort. I would assume this was addressed when the authors excluded immunocompromised individuals (who are more likely to have a lytic infection) and excluded outliers with a higher number of EBV read+. However, the authors should explain this more explicitly.

2) Line 107 & Figure 1: The authors use serology data to corroborate EBV sequencing read detection. The EBV antibodies used, if these are the ones shown in extended Figure 2 (EA-D and ZEBRA are both acute phase and VCA depends on whether it is IgM or IgG, which is not specified), would not show non-controlled latent EBV infection, rather lytic/acute infection. How do the authors explain the use of all antibodies and what their analysis means. Maybe a sub-analysis would help to tease this out.

Non-genetic factors contributing to EBVread+

3) Line 150: "In contrast, increasing age, GS yield..." It is expected that the higher the genome sequencing yield is, the higher the chance of detecting EBV reads. The authors should not really use read counts for EBV positivity, but rather reads per million (RPM) to normalise for the differing sequencing depths in the cohorts.

4) Line 151-153: I don't follow the seasonality effect on EBVread+. Are the authors suggesting that viral load (in latent infections) is higher during winter? If so, is this an effect of other seasonal diseases (i.e. flu)? If it were the effect of other infections reactivating the virus, it would again point to lytic infections rather than latent. Could the authors please comment.

Identification of common genetic variants associated with EBVread+

5) Line 189: "To ensure the robustness of our results..." The authors here show that irrespective of whether the "EBV positive case" is defined by just one read or more reads, the effect sizes of the EBVread+ GWAS lead variants correlate well. Could the authors repeat this (ie EBVread+=1 compared to EBVread+>=2) for other analyses as well, to show that by including the singleton reads, they are not interpreting low level contamination as EBV positivity?

6) Line 197-199: "In contrast, only weak evidence was generated for correlations of effect sizes to EBV sero-positivity (Fig 2c)" In the legend of figure 2 the authors state that seropositivity was defined as having 2 /4 antibodies exceeding thresholds, however as above it would be best to stratify depending on whether the antibodies are indicative of lytic or previous (latent) infections.

7) Lines 215-217: In the genetic study, the discovery cohort was UKB, and the AoU was used to replicate the results. In the previous section (non-genetic factors), the authors start with analysis of the AoU cohort and then UKB. I find this a bit confusing – is there a rationale for this?

Annotation and fine-mapping of associated non-MHM loci

8) Line 268-270: Some specific techniques need a bit more context, particularly for the large audience of a broad-theme journal. Just as an example: "Fine-mapping", "credible sets of variants" might be common in statistical genetics, but they are not well known in other fields (i.e. EBV virology). I find this section very difficult to follow: in Supplementary Table S4 I can see 28 variants, not sets of variants?

Gene-based analyses suggest novel genes for involvement in host-EBV interaction

9) Line 290 : "... we observed 83 genes classified as IEI" I think this is quite an interesting finding and I am wondering whether the authors can check from the Biobank and All of us data, whether these immune deficiencies are also associated with impaired control of other herpesviruse (CMV?) or in general, increased susceptibility to infections (ie other pathogens). In short, is this an EBV-specific phenomenon, or are the individuals that are more likely to not control well the EBV latent infection, overall more susceptible to other infections?

10) Also are these IEI genes involved in EBV-related cancers?

Polygenic architecture of EBVread+ and EBV-associated diseases

11) Line 364: "to IM, HL, MS, RA, non-Hodgkin-Lymphoma, SLE, and Sjogren disease" How were these EBV-associated diseases chosen (aside IM, lymphomas and MS). Also is it not the case that these diseases are associated, for the most part, with an active EBV infection?

Two-Sample Mendelian Randomization (2SMR)

12) Line 445-446: "significant heterogeneity of effects was observed across 445 SNPs, and the causal effects for these outcomes were driven by SNPs in the MHC region" .

I think it is problematic that the observed effects of EBVread+ on RA and T1D were driven by MHC variants. This violates core MR assumptions since MHC/HLA variants directly affect autoimmune disease risk independent of viral control mechanisms, suggesting shared genetic susceptibility rather than causal relations. Also, the evidence suggesting an association between EBV and T1D is weak.

Discussion

13) The authors should also include amongst the limitations that the cohorts used in this study are heavily skewed towards European ancestry populations. This is relevant 1) because of the association with HLA, which varies dramatically across populations; 2) EBV-associated diseases prevalence varies a lot worldwide; 3) EBV viral strains are different across different populations.

Minor comments

Methods

14) Lines 749&772: why did the authors use different criteris for mapping the EBV and HHV7 reads ie soft clipped bases and length of alignment?

Line 7480749: "Only reads where both, forward and reverse, reads mapped to the EBV genome were retained." How can there be EBVread+=1, if both mates in a pair must map to EBV to be retained? Do the authors mean one EBV fragment rather than read?

(Remarks on code availability)

I reviewed the attachments provided with the code. I did not attempt to install and run the analyses. The code seems well organised, with different directories for different analyses and some documentation available.

Referee #6

(Remarks to the Author)

I co-reviewed this manuscript with one of the reviewers who provided the listed reports.

(Remarks on code availability)

Version 2:

Reviewer comments:

Referee #1

(Remarks to the Author)

I thank the authors for their detailed responses to my previous review. The authors have done substantial additional work to address my major concerns and, on the whole, am satisfied with their responses.

My one additional concern surrounds the validation studies of the EBV reads = 1 phenotype. Although the majority of their new analyses support the conclusion that these are true positives, the qPCR validation studies show surprisingly high rates of positivity in the EBV=0 samples (figure 1 panels e & f). In particular, the validation-2 cohort shows essentially equal proportions of positive replicates in the EBV=0, 1 and 2 read-count groups. At this stage, I don't know how much more work can be done to address any potential mis-classification, and I certainly don't want to delay publication further, but I do feel the authors need to address this issue further in discussion and describe how this may influence their results.

Otherwise, I am very impressed with the quality of this manuscript and the additional experiments/analysis.

(Remarks on code availability)

I did not review and revised code

Referee #2

(Remarks to the Author)

The authors have done an excellent job in responding to previous reviewer comments and in revising the manuscript. The revised study is a technically strong and innovative correlation analysis of large health research databases with detection of EBV DNA in genome sequencing from blood samples. The authors demonstrate the feasibility, validity and utility of this type of study, providing a path forward for analysis of multiple other viral and genetic biomarkers.

While the technical aspects of the study are well justified, one could argue that the ultimate conclusion is that EBV viral loads are elevated in individuals that are either immunosuppressed (e.g. HIV or steroids), or have some type of immune-inflammatory or autoimmune disorder. These conclusions are somewhat expected given what is already known about EBV biology. While the methodological approach to large data analysis is technically innovative and exemplary, the biomedical insights may not be very impactful since it is already known that EBV control is affected by other immune disrupting events, including immunosuppressive agents, other viral infection and autoimmune-inflammatory disease.

Nevertheless, the study is among the first to demonstrate the analysis of GWAS data sets can be explored for EBV and other biomarkers to correlate with previously unappreciated disease connections.

Very minor.

Line 369-370. Seems to be a fragment sentence “..54 tissues available in GTEx V8,.”

(Remarks on code availability)

Referee #3

(Remarks to the Author)

The authors have done an admirable job in addressing reviewer concerns, including substantial new analyses. All critiques have been adequately addressed and I have no additional concerns.

(Remarks on code availability)

All code concerns have now been addressed.

Referee #4

(Remarks to the Author)

I co-reviewed this manuscript with one of the reviewers who provided the listed reports.

(Remarks on code availability)

All comments were satisfactorily addressed

Referee #6

(Remarks to the Author)

I co-reviewed this manuscript with one of the reviewers who provided the listed reports.

(Remarks on code availability)

I reviewed the code, but did not attempt to re-run the analysis. The code looks clear, and the folder includes README files and instructions.

Rebuttal Letter

Schmidt et al.: Host control of persistent Epstein-Barr virus infection

We are very grateful for the constructive feedback on the first version of the manuscript, and we provide a detailed response on each of the points below. However, we have identified two major concerns that were flagged by several of the reviewers, which we would like to address in aggregate in an initial paragraph, to avoid redundancy.

1) What does the measure “EBV-read count=1” reflect? Is it really true positives, or false positives due to quality issues (e.g., contaminations/misalignments)?

We had performed our study under the hypothesis that there is a correlation between the amount of EBV-genomes present in an individuals' blood cells, and the EBV-read counts as determined based on genome sequencing (GS) from this sample. Specifically, we expected the individuals in the EBV-read count=1 group to harbor, on average, more EBV viral genomes than the EBV-read count=0 group, but less than the EBV-read count =2 group (and so on; assuming identical GS coverage). Though we are not able to demonstrate the exact quantitative distribution due to the lack of appropriate data in UKB/AoU, we provide multiple lines of new evidence (analyses 1 to 5 below) which, together with the previously reported data, robustly support our claim that EBV-read count=1 are true positives and that higher EBV-read counts represent increased viral load:

Analysis 1: We first addressed the possibility of technical confounding in the EBV-read count=1 group, through the analysis of potential systematic differences in the distribution of EBV-reads between samples with EBV-read counts of 1 and those with EBV-read counts of ≥ 2 :

We compared the coverage along the EBV-genome, for aggregated reads stemming from individuals with EBV-read counts=1 and those with EBV-read counts ≥ 2 . As seen in the left panel below, the distribution of reads along the EBV-genome was highly similar between both groups. We also calculated the average coverage of aggregated reads per EBV gene and, again, observed a strong correlation between the groups EBV-read counts=1 and EBV-read counts ≥ 2 (right panel below; correlation coefficient of 0.895).

► In conclusion, we did not observe systematic differences in the coverage of the EBV-genome or across EBV genes, for individuals with EBV-read counts=1 and those with EBV-read counts ≥ 2 . These plots are now included in the manuscript (**Figure 1b** and **Extended Data Figure 2**), and coverage values for individual genes are provided as **Supplementary Table 1**.

Analysis 2: We next addressed the reviewers' concern that EBV-read count=1 might represent contamination:

For the QC-analysis of UKB library plates, we had established a pipeline for systematic detection of contamination (see **Extended Data Figure 1**, and reviewer 2 - comment 2): Briefly, we had called allelic variants of EBV based on EBV-reads, calculated their allele frequencies (AF) and compared the distribution of common variants across plates. We observed that common variants were absent in EBV-reads from plates with a particularly high number of EBV-read+ individuals (so-called "outlier plates"), supportive of contamination. Specifically, the outlier plates carried EBV with a sequence similar to the EBV reference sequence (NC_007605.1). Notably, this EBV reference is largely derived from the EBV-positive B95-8 cell line, which is commonly used as a source of EBV for generating Lymphoblastoid Cell Lines (PMID 3017841). In contrast, EBV-reads from "non-contaminated" plates showed substantial genetic variability.

When we applied the same approach to reads stemming from individuals with EBV-read count=1 and EBV-read count \geq 2, AF were strongly correlated between both groups (n=1,768 common variants, left panel below). Additionally, a rank-based analysis of these variants (based on AF) was almost identical between both EBV-read count groups (middle panel). When we plotted the same AF distribution of the contaminated and non-contaminated plates, it was observed that the distribution of EBV-read count=1 is clearly different from the one observed on contaminated plates (right panel).

► This data suggests that reads from the EBV-read count=1 group have identical characteristics as those from non-contaminated individuals with higher EBV-read counts. In the manuscript we report those data in the Results section and as part of **new Extended Data Figure 2**.

Analysis 3: We applied quantitative PCR (qPCR) as an orthogonal method to provide support for our hypothesis that GS-based EBV-reads are a measure of elevated EBV viral load.

As no biomaterial for validation could be obtained from UKB or AoU, we identified a small in-house cohort for which standard GS data as well as aliquots of the same DNA samples for

qPCR were available. First, we extracted EBV-reads from GS using our pipeline (n=110 individuals; generated from either whole blood, (n=95) or saliva (n=15)). Similarly to UKB, we did not observe EBV-reads in the majority of samples, but detected 29 samples being EBVread+ (left panel below). Next, we analyzed 72 DNA samples by qPCR, including all samples that were EBVread+ (n=28; one individual did not have DNA available), plus 44 randomly selected EBVread- samples. qPCR was performed using a clinically validated assay targeting *EBNA1* (GeneProof™ EBV PCR Kit (TaqPath™ Menu, Applied Biosystems)).

The result is shown in the right panel below: With increasing EBV-read counts we observed (i) an increase in the number of qPCR-positive replicates, and (ii) an increase in the amount of viral copies, as indicated by a decrease in Cp-values among positive replicates (**new panel e in Figure 1**). Looking at the EBV-read count=1 group in particular, we observed more positive replicates compared to the EBVread=0 group, but fewer positive replicates compared to groups with EBV-read counts of two and higher, as expected.

We also analyzed a second cohort (validation-2), comprising 262 individuals with GS data & DNA samples, from the Japan COVID-19 Task Force (JCTF). In this cohort, qPCR was performed using a self-developed in-house assay on the target gene *EBNA1*. Despite the less sensitive assay and different cohort characteristics (e.g. non-European ethnicity, individuals being coinfecting with SARS-CoV-2), we observed similar results, though we note that the resolution at the lower viral copy range (i.e., in lower EBV-read count groups) was limited (**new panel f in Figure 1**).

► Together, the orthogonal validation by qPCR in two independent cohorts supports our hypothesis that more EBV DNA copies are present in samples with increasing EBV-read counts. We have integrated this analysis in the Results section (panels e and f in **Figure 1**) and **Extended Data Figure 2**.

Analysis 4: We analyzed the correlation of GS-based EBV-read counts and EBV-transcripts in blood-based RNA-seq, to further validate that EBV-read counts=1 are true positives.

We had access to paired GS/RNA-seq data available from 1,010 individuals of the JCTF, (PMID 39317738; n=63 samples overlapped with analysis 3 above). GS data were analyzed using our pipeline, and EBV-transcript counts were obtained as described in PMID 39317738 (modified to contain the EBV-genome instead of the human genome).

Again, the distribution of EBV-read counts was zero-inflated, similar to AoU and UKB, but we observed a higher fraction of EBV-read+ individuals (39.2%; left panel below). This might be due to the co-infection with SARS-CoV-2 at the time of sample collection, or additional cohort characteristics, such as increased average age, larger fraction of males and different ancestry (see **Supplementary Table 2**). Analysing the EBV-transcript counts, we found that the fraction of individuals with any expressed EBV-transcript is lowest in the group of EBV-read count=0, and then increases with higher EBV-read counts (right panel).

► These data provide evidence that EBV-reads are derived from blood cells that are infected with EBV, even in individuals with EBV-read count=1. Further, if we consider transcript abundance as an approximator of the amount of infected blood cells, the correlation of EBV-read counts and EBV-transcript counts indicates that we are measuring more EBV-infected cells, i.e., more viral load, with higher EBV-read counts. We provide this information as new panel in **Figure 1** and **Extended Data Figure 2**.

Analysis 5: Finally, we followed up on the reviewer’s suggestion to include individuals with EBV-read count=1 as controls (i.e., among the group with EBV-read count=0), based on their presumed very low viral load.

We ran GWAS on additional case-control definitions based on different EBV-read counts, using identical sets of covariates as in the main analysis. The different analyses are listed below; with the analysis name indicating EBV-read count groups:

Analysis name	Def cases (EBV-read counts)	Def controls (EBV-read counts)	comment
0 vs 1-18	1 and higher	0	main analysis
1 vs. 2-18	2 and higher	1	already in first version (“within positives”)
0 vs 1	1	0	new
0 vs 2-18	2 and higher	0	new

We then extracted and plotted effect size point estimates for each of the conditionally independent HLA-alleles (left panel) and lead variants at the 27 non-MHC loci (right upper panel). We also calculated inter-analysis-correlations, separately for HLA-alleles and non-MHC (right lower panel).

The main observations are:

- 1) If individuals with EBV-read=1 were genetically similar to individuals with EBVread count=0, a GWAS comparing 0 vs. 1 reads would not show any significant associations beyond by-chance findings (false-positives). This is clearly not the case.
- 2) All risk loci showed the same direction of effects across the four different case-control definitions, and the Spearman's correlations of effect sizes were all >0.85.
- 3) In average, effect sizes were highest in the analysis of 0_vs_2-18 reads, and then decreased in the following order: 0_vs_1-18 reads, 0_vs_1 reads and 1_vs_2-18 reads. This is in line with more risk alleles being present in individuals with higher EBV-read

counts and, thereby, a higher average AF difference to controls (i.e. 0 EBV-reads). This accumulation of risk alleles with higher EBV-read counts is also supported by the increase of the genetic risk score with increasing EBV-read counts (see **Figure 4c**).

► Together, these data show that individuals with EBV-read count=1 are genetically distinct from individuals with EBV-read count=0, supporting our decision that including EBV-read=1 individuals among the cases reflects a phenotype definition of “elevated viral load in blood cells”. In the manuscript, this data is now presented as part of **Extended Figure 6**.

In addition to these novel analyses 1-5, the simulation analysis (now **Extended Data Figure 4**) had already provided robust support for the viral load hypothesis: The simulation shows that a log-normal distribution of EBV reads would yield the exact data pattern as we observe in both UKB (now **Figure 1c**) and AoU (now **Extended Data Fig. 3**), respectively. This includes the observation that the group of EBV-read count=1 have higher simulated EBV genomes than individuals with EBV-read counts=0.

Together, these data provide multiple lines of evidence that (i) the individuals with EBV-read count=1 are indeed true positive samples with low viral copy number, and not misclassified/contaminated, and (ii) the GS-based EBV-read measure reflects increased EBV viral load in blood cells.

We have now integrated all the different lines of evidence in the manuscript, together with the Figures and Extended Data Figures, as indicated above. Also, we now dedicate an entire new paragraph of the Results section to those analyses.

2) Which part of the EBV viral cycle (active/primary infection, latency or lytic reactivation) do we capture?

Before we address this question in detail, we would like to clarify what we had meant by “EBV latency”, which (as we realized) can have two different meanings: It can either refer to (i) the general state following primary EBV infection, with EBV remaining in cells with or without virus production, or (ii) the specific latent state of the viral program, i.e. latency programs O/I-III (which would be distinct from the state of “lytic reactivation”). In our initial manuscript, we had used the term “latent infection” in accordance with definition (i), thereby including potential individuals that might have ongoing low-level or systemic reactivation. We have realized that this has caused confusion, and we apologize for this lack of clarity.

We have now changed the wording of the paper to “persistence” or “persistent infection”, to avoid the term latency when describing the phenotype we are analysing. This is also reflected in the change of the title of the manuscript.

Apart from this terminology aspect, the underlying question remains of key importance: What aspect of the EBV infection do we measure? And which stage of the EBV viral cycle does this measurement reflect?

To answer this, we first summarize the current hypothesis regarding the EBV life cycle in immunocompetent individuals, with a focus on the stage of EBV persistent infection (see scheme below). The currently best established model is the Germinal center (GC) model in

which EBV infects naive B-cells in the lymphoepithelium, leading to the generation of activated blasts which, upon migration to the GC, release GC-derived memory B cells. These memory B cells are characterized by a silent transcriptional program (latency 0 or I) and participate in circulation. Upon cues (e.g. stress, or antigens), EBV-infected memory B cells enter the lytic program, characterized by the subsequent expression of immediate early (e.g. ZEBRA), early (e.g. EA-D) and late (e.g. VCA-p18) antigens (PMIDs 36113467, 7685403, 3018282). This triggers memory B cells to both differentiate into plasma cells and to release new virions into the system. Lytic reactivations are important for viral transmission, but also to replenish the pool of latently infected memory B cells (PMID 19740997). We have summarized this cycle [Redacted text] in **Extended Data Figure 2** (panel g), modified from PMID 26424647:

[Redacted text and figure]

We next aimed to integrate the results from analyses 1-5 above, which demonstrated that we measure the abundance of EBV genomes in blood cells and that increasing EBV-read counts represent increased viral load, and additional analyses (described below) with this model:

i) According to the GC model, EBV persists in memory B cells, mainly in a dormant stage (latency 0), though some cells are undergoing homeostatic cell division (latency I, PMID 36113467, 41223250). Both latency 0 and I are characterized by low transcriptional activity that is limited to the expression of non-coding BARTs and EBERs (PMIDs 10449527, 14688409), with *EBNA1* being additionally expressed in latency stage I.

Using the GS/RNAseq data from 1,010 individuals of the JCTF (see above), we first identified those genes whose expression correlates best with increased EBV-read counts. Using Spearman's rank correlation, we observed the strongest correlation for the EBV-genes *A73* ($\rho=0.47$), *BARF0* ($\rho=0.42$) and *RPMS1* ($\rho=0.43$), all of which belong to the BART gene cluster (see Figure below; now provided as new panel **g** in **Figure 1**). We did not observe any detectable expression for *EBNA1*, despite different approaches to exclude any technical bias introduced by repetitive sequences in that region. The absence of *EBNA1* expression would support that the majority of data has been generated from cells at latency stage 0.

However, we note that standard RNAseq (such as used for the JCTF) might be limited in resolution and therefore prevent proof of latency type 0.

ii) While the correlations were highest for BART genes indicative of latency state 0, we also observed some positive correlations with transcript counts of lytic EBV genes. Among those, the highest correlations were observed for genes located close to the BART gene cluster, which might reflect a technical bias of transcript annotation due to largely overlapping sequences. The observed correlations with lytic genes might suggest some degree of lytic reactivation occurring in peripheral blood cells, a phenomenon that has been recently observed in individuals with Systemic Lupus Erythematosus (see below; PMID 41223250).

iii) We next used IgG levels of four EBV-related antibodies (IgG ZEBRA, EA-D, EBNA-1 and VCA-p18), which were available in a subcohort of UKB individuals (PMID 35383168). We retrieved all individuals within the *EUR*-cohort who were sero-positive for EBV (i.e., 2 out of 4 above Median Fluorescence Intensity (MFI) threshold, $n=7,338$). For each individual and antibody, we compared the GS-based EBV-read count with the respective MFI value. Therefore, MFI values were grouped into deciles, and fractions of EBVread+ individuals were visualized (colored by EBV-read count groups). Correlations between MFI decile and the EBVread+ were also calculated and the results are shown below and in the new

Extended Data Figure 5:

We observe a strong and highly significant correlation of EBVread+ with antibodies (IgG) against VCA-p18, and to a lesser extent to ZEBRA and EBNA-1. While EBNA-1 is expressed in latency-I, ZEBRA and VCA-p18 are expressed in different phases of lytic infection of EBV (see scheme above): ZEBRA in the immediate early phase, and VCA-p18 in the late phase of lytic infection. As VCA-p18 IgG-antibodies are markers of past infections (24175209), and as the age distribution of UKB-participants (>40 years at time of recruitment) renders acute infection unlikely, the correlation of EBVread+ with high IgG antibody levels against VCA-p18 might reflect ongoing lytic activity during persistent infection. Significantly increased IgG titers against VCA-p18 have also been observed in individuals with increased EBV-DNA in blood in independent, though small, studies (PMIDs 11953465, 35921373).

Based on this data, we propose a model in which EBV-reads mainly stem from memory B cells that are latently infected with EBV, and a small pool of reactivating cells. The *in vivo* processes that underlie the presence of reactivating cells in peripheral blood, and their extent, are currently unknown, but the recent observation of B cells in latency II-III as well as lytic stages in peripheral blood of individuals with SLE (PMID 41223250) provides support that lytic reactivation in peripheral blood cells indeed exists. Additionally, EBVread+ is correlated with serological patterns suggesting increased rates of lytic reactivation, which would be in line with models of the EBV life cycle that predict more reactivations when the pool of latently infected B cells increases (GC model). Together, these results now provide novel impulses that should prompt respective follow-up studies.

Referee #1 (Remarks to the Author):

Summary: The authors describe a study of non-genetic and genetic factors contributing to EBV DNA positivity in genome sequence data from two large biobanks (UKBB and All of Us). The authors demonstrate that EBV reads can be successfully recovered from a large percentage of biobank contributors. Non genetic factors including immune compromise, smoking, age and sex were correlated to EBV positivity. A GWAS approach demonstrated a strong genetic component to EBV positivity implicating a major role for HLA alleles and non-HLA immune processes in determining EBV positivity. Importantly, these signals were consistent across biobanks and polygenic risk scores were transferable across biobanks and ancestries. PheWAS and Mendelian randomization analysis showed overlap between the genetic contributors to EBV positivity and several autoimmune diseases with 2SMR analysis suggesting a causal effect of EBV positivity on type 1 diabetes. Overall this is an interesting and well executed study that improves our understanding of control of a common chronic viral infection and the overlap of these processes with additional immune disorders.

Comments:

1. The authors report consistent overlap of MHC and non-MHC associations between the UKBB and AoU cohorts but don't report on results of a genome-wide meta-analysis across both samples. Can the authors comment on if this analysis was performed and if any additional signals were uncovered?

We thank the reviewer for this comment. No, we have not performed a genome-wide meta-analysis between AoU and UKB, for the following reason: Given the strong expected contribution of blood count traits to the probability of detecting EBV-reads in GS data, we considered the lack of information regarding blood count traits in AoU a substantial risk of

confounding. We therefore limited the analysis in AoU to replication of genome-wide significant findings, and herewith followed a recent UKB/AoU-paper published in *Nature* (Gupta et al), which had faced the same challenge. We have added a sentence on this in the Results section:

“No meta-analysis of the two studies was performed, largely due to the lack of blood count data in AoU (Supplementary Note 4).”

Based on the reviewer’s comments 1 and 4 (below), we investigated this “risk” in more detail. Therefore, we re-ran the UKB analyses using different sets of covariates, in-/excluding blood count traits (now added in **Supplementary Table S3**). To our surprise, when inspecting genome-wide significant variants, the effect of blood count traits as covariates was smaller than we expected (see new **Extended Data Figure 6**). However, we cannot rule out any residual confounding beyond the genome-wide significant findings. In order to follow a conservative approach for reporting our result and in analogy to Gupta et al, we therefore kept the genetic analysis in AoU to a replication-only strategy, with the additional aspect that this also allowed us to use the genome-wide results in AoU as an independent resource for the additional analyses in the manuscript. Still, a meta-analysis would likely identify more genetic contributors to EBV viral load control, in particular when performed in different ancestries where it could also be used for more specific finemapping. As this is important, we have now also added a perspective statement on these kinds of analysis in the Discussion section and in the referred Supplementary Note 4.

2. Interestingly, in addition to identifying several strongly associated HLA alleles, the authors identify associations in key proteins involved in antigen processing. Did the authors perform any interaction or epistatic analysis between HLA alleles and these other proteins? I’m specifically thinking about interaction between HLA and ERAP alleles as was observed for ankylosing spondylitis [PMID 21743469]. This might provide additional information as to the function of various alleles and mechanism of immune evasion.

We thank the reviewer for this excellent suggestion. We performed an interaction analysis based on the approach described in the cited paper (PMID 21743469). To reduce the burden of multiple testing, we restricted our analysis to the 54 conditionally independent HLA-alleles and the three top-associated non-MHC loci, which included the *ERAP2* locus. We would like to stress that none of the genes located at the other non-MHC loci had evidence for a functional role in antigen processing.

We identified 13 nominally significant epistatic effects, three of which withstood Bonferroni correction for multiple testing (for 162 tests). All three Bonferroni-significant interactions were between the *ERAP2* locus and HLA-alleles of MHC class I, which is biologically plausible as the endoplasmic reticulum aminopeptidase 2 encoded by *ERAP2* is involved specifically in the processing of antigens for MHC class I presentation (PMID 37925533).

We have now added this information in the Results section and illustrate the three significant findings in new plots of **Figure 2**. Further, the analysis is described in the Methods and the full results of the interaction analysis can be found in the new **Supplementary Table S10**.

3. Perhaps I’m mis-interpreting but the analysis presented in figure 2b seems to suggest that individuals with only one detectable EBV read may be mis-classified in the beta_main

analysis. If the effect size of the associations are roughly the same when using the EBV_One group as controls as with the full, non-EBV set of individuals, wouldn't this motivate either excluding the EBV_one group or including them with the full control set?

As presented in the initial paragraph, there is robust evidence that individuals with EBV-read counts =1 aren't misclassified, but rather true positives with low viral load. We comment on the specific implications for this reviewer point below:

Though we saw a high correlation of effect sizes between the main analysis (i.e., comparing 0 reads to ≥ 1 reads) and the "within-positives" analysis (i.e., read=1 vs ≥ 2 reads (former Figure 2b), the point estimates of effect sizes at genome-wide significant loci were, on average, larger in the main analysis. This suggests that there is more statistical power when the controls are EBV-read counts=0. This information had not been well presented in the first version of the manuscript. We now made this more explicit in **Extended Data Figure 6**.

We also performed an analysis comparing 0 vs. 1 reads. If individuals with EBV-read=1 are similar to individuals with EBVread counts=0, such a GWAS would not show any significant associations beyond by-chance findings (false-positives). However, we observed a striking correlation of effect sizes for genome-wide significant lead variants between this analysis and the main analysis, as illustrated in **Extended Data Figure 6**.

Notably, this finding is in line with a quantitative genetic model in which the polygenic predisposition is increasing when the group comparisons in a GWAS get "more extreme": In groups with higher EBV-read counts, risk-increasing alleles accumulate, while in the controls, the protective alleles aggregate. This leads to an increase in absolute differences of allele frequencies between groups, and therefore to increased absolute point estimates.

We tested if this was true, using different additional case-control definitions (see initial paragraph). As expected, the absolute effect sizes for the genome-wide significant loci further increased with "more extreme" group definitions. This reflects an accumulation of risk alleles with higher EBV-read counts, which is also supported by the increase of the polygenic risk score with increasing EBV-read counts (**Figure 4c**).

We now provide these analyses as **Supplementary Table 7** and also illustrate this graphically in **Extended Data Figure 6**.

Can the authors comment on if they are thinking the EBV_One set are the result of some mis-alignment or true positives with very low EBV levels?

We think that these are true positives with low EBV DNA levels, and we now provide additional experimental and computational evidence for this assumption. As this concern has been jointly raised by several reviewers, we address this comment in the introductory paragraph.

4. Given the incomplete overlap in clinical data between UKBB and AoU, the methods for covariate selection were necessarily different. Can the authors comment on how these different sets of covariates may influence the results of the association analyses and the accuracy of the polygenic risk score predictions?

We are grateful to the reviewer for bringing this aspect up. In particular the lack of blood count data in AoU was a substantial difference in the covariates between AoU and UKB. As part of this comment, we have realized that a comprehensive overview of the shared and different covariates between AoU/UKBB was missing, which we have now added into **Supplementary Table S4**. We have also added new columns to **Supplementary Table S4**, which describe the sets of variables that we used for a detailed analysis in response to this comment, as described below:

The effect of the covariate differences between UKB and AoU is difficult to quantify, in particular as even when correcting for some covariates, the underlying data structure could still be subject to different confounding. Therefore, to get an idea of their influence, we calculated within UKB two additional GWAS: (i) similar covariates as in the main analysis but without blood count traits (“no_blood”), and (ii) a basic covariate model only considering age, sex (“basic”; detailed in **Supplementary Table S4**; results in **Supplementary Table S7**). When we plotted the effect size estimates for the genome-wide significant loci from the main analysis in comparison to those two results (**Extended Data Figure 6**), we were surprised to not find large differences in effect sizes, at least not within UKB and for the genome-wide significant loci. However, there might still be residual confounding that is not reflected in the genome-wide significant loci. We describe this additional analysis in **Supplementary Note 4**.

To specifically answer the reviewer’s question on how this covariate selection influenced the genetic risk score (GRS) analyses, we would like to emphasize that we only generated the GRS in the UKB cohort, which is the “better controlled dataset”, and then applied it to AoU, in which it performed equally well for the same (European) ancestry. Importantly, in the UKB cohort, we calculated all weights for genetic variants while correcting for potential confounders - therefore, the GRS / the effects of the genetic variants should mainly capture EBVread+, though we cannot rule out that there might still be some residual confounding. We have not generated any GRS in AoU, largely due to the uncertainty associated with potential blood count trait and residual confounding, though this effect seems to be small (as described above).

Referee #1 (Remarks on code availability):

I was able to download and run the provided code and obtain the expected output. I did not have an external set of genome sequences aligned with the EBV contig readily available so can't comment on non-test cases.

Thank you.

Referee #2 (Remarks to the Author):

Schmidt et al. used whole genome sequencing data from 486,315 individuals from the UK Biobank (UKB) and 336,123 individuals from All of Us (AOU) to identify genetic and non-genetic factors involved in controlling latent EBV infection. Short reads mapping to the EBV genome were identified in 16.2% of UKB samples and 21.8% of AOU samples. The authors assert that the detection of EBV short reads (as few as one EBV short read) represent a

robust surrogate measure for high EBV viral loads. GWAS comparing individuals who were EBV read-positive vs -negative identified strong associations at the MHC loci as well as 27 loci external to the MHC. The presence of EBV reads was associated with immunosuppression, winter sample acquisition, and inflammatory diseases including rheumatoid arthritis, inflammatory bowel disease, and type I diabetes. In addition, gene-set enrichment analysis implicated NKT and CD8+T cells in the control of EBV infection. Status as an active smoker, an environmental risk factor for EBV-associated cancers and autoimmunity, was identified as a modifiable factor that influences EBV read detection and, the authors argue, by extension, EBV viral loads. The strength of this study lies in the impressive size of the two cohorts, the identification of novel non-HLA genes associated with control of EBV infection, and correlates of EBV DNA detection/load with various disease indications.

Comments:

1. The claim that the detection of short reads mapping to the EBV genome is a robust surrogate for higher viral loads should be validated with an additional measurement (e.g. EBV DNA detection in PBMC, serum/plasma, or saliva)—although this may not be available for UKB or AOU, it could be performed with a smaller cohort. The simulation of EBV viral loads (Extended data figure 1) is interesting, but should be validated experimentally, rather than rely on comparisons with HIV viral loads, especially because there is not a clear relationship between EBV reads and seropositivity. Moreover, the viral lifecycles of retroviruses and herpesviruses are quite different and may not be directly comparable.

We thank the reviewer for this suggestion. As this question relates to the more general concern of “what is the EBV-read count=1 group”, we would like to refer to our introductory paragraph, where we have aggregated all lines of evidence which show that we are indeed measuring variation in EBV viral load. Among the analyses conducted, analysis 3 specifically addresses the reviewer’s comment. We provide additional details here:

Unfortunately, it is not possible to get hold of biomaterial from a (sub-)sample of UKB or AoU for validation purposes. Also, we were not able to identify any other large cohort that had GS data plus PBMC/serum or plasma reported or directly available. We therefore performed a qPCR-based validation of samples from a smaller, independent cohort which had (i) standard, short-read GS data generated, (ii) additional DNA from the same collection time point available for laboratory analysis, and (iii) proper consent to be used as control/replication study. This in-house cohort (validation-1) comprised 110 European patients with ACE inhibitor-induced angioedema (ACE-cohort):

We first identified EBV-reads in the GS data by applying our EBV-read extraction pipeline. Based on the results, we selected a subset of 72 individuals for qPCR analysis, with the aim to represent both EBVread- and EBVread+ samples (with different EBV-read counts). qPCR was performed in four replicates on genomic DNA, using the clinically validated GeneProof™ EBV PCR Kit (TaqPath™ Menu, Applied Biosystems). We then correlated EBV-read counts with two measures, i.e. (i) the number of qPCR-positive replicates and (ii) the average Cp-value, which is a direct measure reflecting the input of viral material. As shown below (left panel), we observed an increase in percentage of positive replicates with increasing EBV-read counts:

- In EBV-read count=0, 50/176 (= 28.41%) were qPCR positive, reflecting the limited sensitivity of the GS-based EBVread+ and the presence of individuals with low viral load in this group.
- In the group of individuals with EBV-read counts =1, the fraction of qPCR positive replicates was 67.9%, and
- further increased to 100% in EBV-read counts of 10 or higher.

In parallel, we observed a negative correlation between Cp-values and EBV-read counts, reflecting the presence of more viral copies in samples for which more EBV-reads are detected. In the EBV-read count=1 group in particular, we found more positive replicates/lower Cp-values compared to the EBV-read count =0 group, but fewer positive replicates/higher Cp-values compared to groups with EBV-read counts of two and higher . These data support that samples from EBVread+ individuals contain higher viral loads than those with EBV-read count=0.

To investigate how generalizable these findings are, we performed a similar analysis in 262 individuals from the JCTF cohort (validation-2), which had different sample characteristics (non-Europeans, older, more male, co-infected with SARS-CoV-2). Here, samples were analyzed in triplicates, using a non-clinically validated in-house assay and, again, *EBNA1* as target gene. As illustrated below (right panel), we saw the same correlation pattern, i.e. a positive correlation with EBV-read counts and % positive replicates, and a negative correlation with Cp values. However, we note a lack of resolution in the low EBV viral count groups, probably due to a lower assay sensitivity.

Together with the data from the other analyses (see introductory paragraph), we provide robust evidence for GS-based EBV-read counts to reflect viral load during persistence.

2. EBV reads from 50 preparation plates were discarded because they had an unexpectedly high number of individuals who were EBV positive, and it was determined that these plates might be contaminated. What was the threshold for determining that plates had too many positive individuals (what percent +)? Whole genome sequence analysis should be able to identify whether the EBV in those plates had the identical EBV sequence—which would support contamination. The lower overall % positivity (16.2% vs 21.8%) in the UKB vs AOU cohort suggest that the threshold used for the plates in the UKB cohort likely missed/underestimated many individuals with poor control of EBV infection. How might the

inclusion of these high copy number of EBV alter the correlation studies and overall conclusions?

We thank the reviewer for this comment, which contains two important aspects that are related, but also to a certain degree distinct:

a) What is the basis for the decision to exclude plates with a high number of EBV-read positive samples (or: how do we ensure they were really contaminated?),

and

b) How do we explain the differences in EBVread+ between UKB and AoU?

We first address point a):

Though the number of EBV-reads per sample is limited, we assigned all library preparation plates in UKB to one of three groups, based on the percentage of EBVread+ individuals (see **Extended Data Figure 1**):

- Group 1 - regular plates: Reads from all other plates, i.e. those with EBVread+ fractions within 2 standard deviations (“non_contaminated.bam”; 5,217 plates)
- Group 2 - “intermediate”: Reads from 38 plates with less than 80%, but more than 28.8% (i.e., 2 standard deviations of mean) of EBVread+ individuals (“intermediate.bam”)
- Group 3 - “outlier”: Reads from 13 plates with >80% positive samples (“contaminated80.bam”)

We then aggregated the EBV-reads within these three groups and visualized them in IGV. Below we provide a representative snapshot:

As expected for an endemic circulating virus, we observe substantial variation within the EBV- genome across the regular plates (group 1). In contrast, the contaminated plates (group 3) almost perfectly match the EBV reference - we do not see any large strain variability here. Notably, the reference sequence of EBV, NC_007605.1, is derived from the EBV-positive B95-8 cell line (with the exception of a region between ~140,000 and ~150,000 bp), which is commonly used as a source of EBV for generating Lymphoblastoid Cell Lines (LCLs, PMID 3017841). Therefore, EBV stemming directly or indirectly (i.e. via LCLs) from the B95-8 cell line is the probable cause of the contamination with “reference EBV” in the contaminated plates. The intermediate plates (group 2) have a mixed pattern, indicating that this group might contain both contaminated and non-contaminated plates.

We additionally performed a more systematic analysis based on the AF of EBV variants in aggregated reads. Herefore, we first called EBV variants using a multi-sample bam file of the aggregated EBV-reads across all groups as described above (commands: mpileup (parameters: -a QS -r chrEBV -d 9999), call (parameters: -G - -mv) within bcftools (version 1.20)). To obtain relatively high quality and common variants, only sites with an aggregated coverage above 300 and variants with an AF above 5% were kept (n=2,240 variants). Samples were then reattributed to the three groups, and those variants that were not well represented in any of the three groups were removed from all three groups. This was achieved by identifying variants with low allele numbers (AN), i.e. variants with AN below $\text{mean}(\text{AN}) - 2 \cdot \text{SD}(\text{AN})$ in each group, which resulted in n=1,687 variants. Finally, AFs were plotted against AF ranks, for each of the three groups (gray: regular plates; orange: intermediate plates; red: outlier plates):

For further illustration, we aggregated all three groups into one panel, using rank distributions of the variants. Again, we observe different rank distributions across the three groups:

Together, these analyses indicate that there is almost no genetic variation in the plates of group 3, as opposed to the two others (with the intermediate group ranging in the middle, reflecting a likely mixture of contaminated and non-contaminated plates). Therefore, following a conservative approach in order to minimize false-positive individuals in our

analysis, we removed both the intermediate group (defined as more than 2 standard deviations of the mean of EBVread+ individuals per plate), as well as the contaminated group, from further analyses.

We now explain this analysis in the **Supplementary Note 2** and provide the plots above (and additional ones) in **Extended Data Figures 1 and 2**.

We next address point b):

The reviewer suggests that the lower % of EBVread+ individuals observed in UKB as compared to AoU might be due to the exclusion of true-positive samples from UKB as part of QC. While we agree that we theoretically might have removed some true positives in the “intermediate plates”, we present robust evidence that most of the individuals on the 51 plates are likely contaminations (see comment above).

Also, even if we hadn’t excluded any of the 51 plates at all (and had assumed that all individuals in those plates were true positives), the overall positivity rate in UKB would still be lower than in the overall AoU cohort (21.8%). We calculate this theoretical value here:

- overall UKB cohort size: 490,294 individuals
- UKB-QC cohort: 486,315 individuals, of which 16.2% are EBVread+ (n=78,783)
- excluded individuals on 51 plates: n=3,979
 - → including those would yield a maximum of $78,783+3,979=82,762$ EBVread+ individuals → $82,762/490,294 = 16.9\%$

These data render it rather unlikely that the exclusion of the 51 plates is the source of the lower % of EBVread+ in UKB as compared to AoU. Instead, we hypothesize that there are underlying differences between both datasets which could explain the observed difference: In line with our observation that smoking, male sex and older age increase the probability of being detected as EBVread+, we observe that in the AoU cohort the number of smokers is higher, and it has a larger fraction of men (new **Supplementary Table S2**). Further, the average GS coverage in AoU is substantially higher (ca 38.25x) than the average coverage of the UKB cohort (ca. 32.5x, based on PMID 40770095), again increasing the probability of detecting EBV-reads.

In addition, we observe a substantial variation in EBVread+ across the different genetic ancestries in AoU, with Europeans having the lowest number and Africans the highest fraction of EBVread+ individuals. Indeed, when specifically comparing the European subcohorts of both UKB and AoU, the numbers of EBV-read positive samples get much closer (16.2% to 17.6%), with the remaining difference likely being explained by the increased prevalence of smokers and men (with age being similar), and technical confounders such as higher GS coverage.

We describe these additional considerations in the Results section, and provide detailed cohort characteristics in **Supplementary Table S2**.

3. 61.9% of EBVread+ individuals had only one read—with a maximum 27,649 reads detected. The authors state that there was uniform coverage of the EBV genome, but the distribution of EBV genes detected should be shown to see if there was a technical bias in

terms of which EBV gene was detected in the short read analysis. The effect of size for lead variants is shown based on antibody abundance against read association analysis (Extended data figure 2), but this does not indicate the relative abundance of each read.

This comment was raised by all reviewers, indicating a major question: what does the EBV-read count=1 reflect (and have we ruled out any potential technical bias)? To address this, we have generated multiple lines of evidence which we summarize in the initial paragraph at the beginning of this rebuttal letter. However, we would also like to provide some specific comments here:

As shown in new **Figure 1c** (old Figure 1b), there is an even distribution of short reads across the entire EBV-genome when all EBV-reads are aggregated. To see whether individuals with EBV-read counts=1 do have a different distribution of reads (which might indicate some bias or technical error in this group), we plotted the cumulative coverage of reads separated by EBV-read count number (EBVreads=1; EBVreads>=2) in comparison to the overall distribution. As can be seen from the plot, the distribution of reads across the EBV- genome is highly similar.

Furthermore, we calculated the average coverage for each of the 94 EBV genes. This data, which we provide as new **Supplementary Table S1**, further reflect the even distribution of reads across the different genes, as each position is covered between 222 (min) and 489 (max) reads when all EBVread+ individuals are considered. Notably, 95% of all EBVread genes are within a coverage range from 336.5 to 445.3X. When we, again, split the EBVread+ individuals by those with EBV-read counts=1 and those with EBV-read counts=>2, we find a very strong correlation between both groups (new **Extended Data Figure 2**)

Together, this data provides further evidence that EBV-read count=1 individuals are true positives.

4. The authors discuss non-HLA risk loci associated with high EBV viral load during latency. However, without determining EBV gene expression, it is unclear if the reads detected are associated with EBV latency or lytic reactivation.

The question “what state of the EBV viral life cycle do our GS-based EBV-reads capture” is central to our paper. Accordingly, this question has been raised by several reviewers, and we summarize our answers to that question in the initial paragraph. As one main aspect, we have realized that we had intended a different meaning for the term “latency” in our first version of the manuscript. To avoid confusion, we now refer to the trait as “EBV persistence”. However, in addition to this aspect of terminology, we also investigated this question in detail:

We analyzed 1,010 individuals from a published cohort (PMID 39317738) for whom paired blood-based GS and bulk RNAseq data are available from the same biological sample. Following alignment to 94 genes of the EBV-genome, EBV-transcript counts were correlated with GS-based EBV-read counts. The highest correlations were observed for *A73*, *RPMS1* and *BARF0*, all of which are members of the BART gene cluster that is primarily expressed in latency. In healthy individuals, it is expected that the predominant stage of EBV in memory B cells in peripheral blood is latency stage 0, though latency 1 (during low-grade cell division) might also be present and would be characterized by the expression of *EBNA1*.

However, we did not find any *EBNA1* expression with the standard analyses, and also did not find any when we accounted for the technically challenging profile of this region (repetitive sequences). We therefore conclude that we measure EBV DNA derived from memory B cells which are representing latent EBV, likely latency 0, though we cannot rule out any low *EBNA1* expression in at least some samples.

Importantly, we also observed some correlation between EBVread+ and lytic transcripts, even when excluding those transcripts adjacent to the BART gene cluster (**Figure 1g**). Therefore, this data points towards a correlation between EBVread+ and lytic activity.

Indeed, additional correlation analyses of EBV-read counts and IgG-based EBV antibody titers (see introductory paragraph) suggest that EBVread+ is associated with more (prior) lytic reactivations during persistent EBV infection. This is further supported by literature, e.g. a correlation between (i) blood viral load and IgG of VCAp18 in immunocompetent individuals (PMIDs 11953465, 35921373). These findings are also consistent with the GC model of EBV infection, and in accordance with findings that the suppression of lytic reactivation leads to a decrease in EBV viral load (for references, see initial paragraph).

5. EBV detection was increased in males vs females, but autoimmune diseases associated with EBV are more frequently observed in females. Are there genes associated with males who are EBV read positive, not immunocompromised, and without inflammatory disease? Perhaps these genes could be associated with a protection from EBV mediated-autoimmune disorders despite poor EBV control.

While we agree that potential male-/female-specific effects are of high interest, we have not investigated this aspect specifically in the first version, and also cannot fully cover this in the current manuscript. However, in order to understand whether it is encouraged to pursue research into this direction, we calculated two sex-restricted GWAS in UKB, including only male cases/controls (“male”) or only female cases/controls (“female”), to see whether any of the genome-wide significant loci show different, or even opposite, effects between males and females. As shown in new **Extended Data Figure 6**, all conditionally independent HLA alleles (and all of the 27 non-MHC loci) had the same directions of effect and were similar in magnitude. Calculating Spearman’s correlation between both analyses revealed $\rho=0.99$ (HLA-alleles) and 0.93 (non-MHC), respectively. Therefore, at the genome-wide significant loci we cannot find evidence for a prominent sex-specific genetic effect on EBV viral load. However, this does not rule out that some protective mechanisms might be present in specific subgroups, or that gene-environment interactions between host control of EBV viral load and sex-specific factors could modify the risk. We now indicate this as a future research direction in the Discussion:

“Our data also encourage investigations of potential sex-specific effects of EBV host control, given the increased prevalence of EBVread+ in males and the reported increased prevalence in autoimmune diseases in females (PMID 39286970).”

6. No association was found for EBV reads and the major MS risk allele HLA-DRB1*15:01, leading the authors to conclude this risk allele does not alter EBV viral load. What are the implications of this relating to the known correlation of EBV seropositivity and MS, and viral copy number and seropositivity, if any?

This is a great question! It actually illustrates the potential of our study: to provide additional context for some correlations and associations that have been observed with EBV-related traits or diseases.

Our statement of “no associations on EBVreads were found for HLA-DRB1*15:01” referred to the observations that this HLA-allele had

- not been selected as an independent HLA-allele in the conditional analysis in UKB, and
- had not been associated at all in the AoU cohort.

However, we note that this allele has been among the 116 genome-wide significant HLA-alleles in the main analysis of UKB, prior to conditioning.

To better understand the contribution of HLA-DRB1*15:01 in the context of EBV and MS, we retrieved association statistics for HLA-DRB1*15:01 from all unconditioned analyses we performed (including different case-control definitions and serology data) as well as from a recent GWAS of MS (PMID 26343388), and plotted the effect sizes as Forest plot (genome-wide significant traits in bold):

In aggregate, the data illustrate that HLA-DRB1*15:01

- shows the strongest association with Multiple Sclerosis case/control status,
- has some strong associations at serology level, particularly against IgG EBNA-1 and VCA-p18
- is associated with EBVread+ in UKB, but with comparably small effect sizes
- is not significantly associated with EBVread+ in AoU
- has a genome-wide significant association with HHV7, though effect size is in the opposite direction.

Based on these data, we propose that HLA-DRB1*15:01 does not have a (strong) effect on the control of EBV viral load in peripheral blood, but mediates its effect at protein level through other mechanisms, which could include preferential presentation of EBV antigens, (Drosu et al. 2024, PNAS; Lanz et al., 2024, PNAS), an expansion of specific B cell subsets (Läderach et al., 2025, Nature), or molecular mimicry. In support of this, we would like to

highlight that the strongest association of HLA-DRB1*15:01 at the molecular level is with antibody level against EBNA-1, which has been shown to cross-react with GlialCAM due to molecular mimicry (Lanz et al. 2022).

We additionally note that imputation of HLA-alleles based on common variants is imperfect, and extended LD between different HLA-genes additionally complicates downstream analyses. In the UK population, HLA-DRB1*15:01 has been shown to be in strong LD with HLA-DQB1*06:02 (PMID: 26343388), which is among the conditionally independent alleles of our analyses, and has been selected at position 30. Thus, while HLA-DRB1*15:01 / HLA-DQB1*06:02 are not the most significant HLA-alleles associated with EBV-read counts, they could still contribute to this trait as part of its multifactorial etiology.

We now provide the Forest plot as **Extended Data Figure 9**, and have added the following statements to the Discussion section:

*“In contrast, no consistent effect on EBVread+ was found for the major MS risk allele HLA-DRB1*15:01 (ref 36), supporting a pathomechanism distinct from EBV viral load control. This could include a stronger antibody response through preferential EBV peptide presentation (ref 63,64), expansion of specific B cell subsets (ref 65), or molecular mimicry. Indeed, in a detailed analysis of HLA-DRB1*15:01 (Extended Data Fig. 9), the strongest effect size was found for a positive status for IgG EBNA-1, and EBNA-1 has been shown to cross-react with the central nervous system protein GlialCAM (ref 8).”*

Minor points.

1. Figures are difficult to read and not very easy to understand. They could be higher quality. Especially Fig 4g. Font colors are hard to read, some figure fonts are fuzzy/low quality.

We have extensively modified all Main Figures, including re-design with Adobe Illustrator. As far as we can see, this has resolved the “fuzzyness”-problem. Also, we have extended the information in the legend at many instances, to improve understanding of the Figures’ content.

2. Line 443. Interesting that MS did not score as most significant in the Mendelian Randomization and predictive causal relationship, while several other autoimmune diseases RA, CI, and T1D.

In our study we explored correlations between EBVread+ and EBV-associated traits by two different approaches: using genetic risk scores (GRS), and Mendelian Randomization (MR). In the GRS-analysis, we identified evidence for nine traits to be genetically correlated with EBV viral load, including MS. As GRS are not designed for causal inference, we applied MR on these nine traits and identified some suggestive evidence for causality for two traits, i.e. RA and T1D. However, these signals were largely driven by variants in the MHC region for which pleiotropy cannot be ruled out. Therefore, the MR results need to be interpreted with caution. We discuss this more specifically below (reviewer 5, comment 12).

While the lack of a causal effect of EBVread+ on MS seems surprising, there is also some reasonable explanations:

First, the basis for GRS/MR is the genetically-mediated control of EBV viral load (and, in the case of MR, its direct effect on downstream traits). Thus, if the effect of EBV on a trait is not

biologically mediated by host control of EBV viral load (but through other mechanisms), it might not be detectable in these blood–GS-based analyses. As an example, for MS, the effect of one specific allele (HLA-DRB1*15:01) might be driven by preferential presentation of EBV peptides, thus triggering a stronger immune response, and/or by a molecular mimicry effect in nervous tissue (see reviewer 2, main point 6), both of which do not largely impact the host control EBV viral load.

Second, some traits where the pathomechanisms might indeed be mediated by EBV viral load might have gone unnoticed till date, due to the lack of appropriate data in large cohorts. So far, most analyses had been performed using serology data or EBV seropositivity, which are only limitedly correlated with EBV viral load (see **Extended Data Figure 6**).

We therefore do not consider our results in contrast to other studies, but they rather give novel mechanistic context for known associations (e.g. HLA-DRB1*15:01 and MS) and suggest traits that should be explored for a potential contribution of EBV host control in future studies.

3. Line 455. “our study is the first to ..” This may not be completely correct, so best to avoid “first” .

We have removed this comment.

4. The conclusion that genes affecting T-cell and NK-cells are most significantly regulating EBV load is already well established, so this is not a very novel finding, although it does support the methodological approach.

This has been rephrased.

5. Extended Data Figure 2, panel e. “not available”. Maybe a PDF issue or else this panel be replaced/ removed?

We have entirely removed this panel as part of the revision process. The corresponding data is now shown in a Heatmap, in **Extended Data Figure 6**, and provided in **Supplementary Table S7**. Still, we would like to mention that the “not available” was not a technical error, but referred to the fact that no HLA-allele information or variants in the MHC-region were available for the memory B cell abundance GWAS.

Referee #3 (Remarks to the Author):

This manuscript by Schmidt, et al. aims to uncover the extent of Epstein Barr Virus (EBV) viral load in human subjects, and determine genetic and non-genetic host factors that control EBV infection. Their approach exploits the fact that EBV reads will be present as a by-product of genome sequencing, owing to the integration of EBV DNA during latent infection. Using the UK Biobank sequencing data as a discovery cohort, the authors map these reads to the EBV genome, thus acquiring measure of viral load, and proceed to analyze these data with respect to host genotype and phenotypes, uncovering a number of robust associations. These are mostly replicated by the authors using the All of Us cohort.

Overall, this is a meticulously performed and well-written study that yields a number of novel insights. The authors are exceedingly careful at each step in their analysis, and provide substantial support for each of their conclusions. This paper will be of interest as it greatly improves our understanding of factors underlying EBV control, with important implications for autoimmunity for example; as well as serving as an important jumping off point for any number of new research avenues. I do have a few issues that I think need to be addressed however:

1 -My major concern involves the characterization of samples with a single read mapping to EBV as “EBV+,” which corresponds to 61.9% of all EBV+ samples. This strikes me as a concerningly low threshold, despite the fact that the authors do make an effort to eliminate read mis-mapping (e.g., requiring forward and reverse reads, requiring 120 matching bases). Despite these precautions, the percentage of EBV+ samples (16.2%) is higher than cited for immunocompetent individuals detected using qPCR (11%). If these single-read samples were not included, by my calculation the EBV+ samples would come in at 6.2%, which seems equivalently plausible. It would be good to have more understanding of what we are looking at with these samples with only a single EBV read. For example, is there a specific region of the EBV genome where we are seeing the preponderance of these reads mapped? Or is their distribution similar to that for samples with higher numbers of reads? Is it possible to validate any of these samples with qPCR?

We thank the reviewer for expressing this concern, which reflects similar thoughts of the other reviewers: what does the EBV-read count=1 group represent? As this is a central point in our study, we have now aggregated several lines of evidence that support that the samples with EBV-read counts=1 are true positives with low viral load. We lay these arguments out in detail in the initial paragraph and would like to refer the reviewer to this section. At this point we would like to highlight two analyses in particular, which specifically address this reviewer’s concern:

- We now provide the genome coverage plot separately for individuals of EBV-read counts=1, and those with EBVread counts ≥ 2 (see new **Figure 1b**). It can be seen that the distribution of reads is highly similar, and no specific region can be identified to which the EBVread=1 counts would map. This is further reflected by largely equal coverage of all 94 EBV genes in the EBV-genome (new **Supplementary Table S1**), which again is not different for the groups of EBV-read count=1 and those with EBV-read counts of ≥ 2 (see also reviewer 2, main comment 3).
- Unfortunately, there is no way to access the AoU or UKB material directly for qPCR. However, we had access to 110 individuals from an in-house cohort for which GS and DNA aliquots were available. Extracting EBV-reads from those individuals mimicked the distribution in UKB and AoU (including a zero-skewed distribution and individuals with one or more EBV-reads). Running a clinically-validated qPCR assay on those samples (using *EBNA1* as target), we found that EBV-read count=1 samples are more often positive and, if so, with lower Cp values (indicative of higher viral load) than the EBV-read count=0 group. Similarly, samples with EBV-read counts =2-5 had more positive replicates, and again lower Cp values, than samples from the EBV-read count=1 group. We provide this data above, in **Figure 1** and as new **Extended Data Figure 2**.

The reviewer also asks a separate (though related) question: If we rule out that individuals with EBV-read count=1 are false-positives, why is the number of EBVread+ samples in the immunocompetent individuals still higher than in the two studies (qPCR, GS) which we cited? While we do not have a definite answer due to the lack of information, there are several hypotheses:

- First, as shown by our analysis of non-genetic factors, we see a large contribution of both ancestry and demographic factors to the probability of detecting EBV-reads. Therefore, if the participants of the UKB/AoU cohorts have different characteristics (i.e., are older, more smokers, more men) than the cohorts used in the qPCR analysis, this would increase the fraction of EBVread+ individuals. Unfortunately, none of the two cited papers used for comparison provides information on cohort demographics.
- Second, depending on the way of DNA extraction, relevant blood cell types might be differently represented in the biomaterial used for qPCR and GS, which would reflect a technical bias that might influence the results. Again, we lack the data to follow up on this hypothesis.
- Specifically for the qPCR study, an additional concern is that this data were from a diagnostic setting, in which generally stricter cut-offs are applied for determining a sample to be EBV-positive. As the individuals with EBV-read counts=1 have only low viral loads, they might have been more likely to be set to EBV-negative in this clinical assay.

We now provide this context in the Results section of the main paper:

“In both UKB and AoU, the fraction of EBVread+ individuals was higher than in smaller studies of immunocompetent individuals involving GS (14.0%)²⁴ or diagnostic quantitative PCR (qPCR, 11.03%)¹⁸. These differences might be attributable to differences in cohort composition and/or strict cut-offs used in clinical settings.”

2 -Likewise, in the All of Us cohort, EBV+ is given as 21.8%; how many of those samples were single-read only?

The distribution of EBVread+ samples in AoU is given in **Extended Data Fig. 3** (former **Supplementary Figure 2**). Specifically, the number of samples with EBV-read count=1 is 37,901, representing 51.8% of EBVread+ individuals across all ethnicities (n=73,137).

In the European subcohort of AoU (n=189,658), which is more closely resembling the analyzed UKB cohort, we had 19,017 individuals with EBV-read count =1, which is 56.8% of all EBVread+ individuals (n=33,472).

Taken together, in the overall AoU compared to UKB we do see (i) a higher fraction of EBVread+ individuals (21.8% vs. 16.2%), and (ii) also higher read count numbers in EBVread+ individuals (i.e., more of the EBVread+ individuals have at least 2 read counts; 48.2% to 38.1%). However, numbers are more similar when individuals are matched on ethnicity (17.6% vs. 16.2%), with the remaining difference likely reflecting the different cohort compositions, as described above in response to reviewer 2, main point 2b. In order to avoid redundancy, we kindly refer the reviewer to our reply above.

We now make this information more explicit in the main text and have also aggregated information on cohort characteristics in **Supplementary Table 2**.

3 -The above concerns also extend to the analysis performed examining samples with read counts of 2 or more vs those with only a single read. Again, one wonders whether the clear differences here were due to viral load, as the authors assert, or the fact that some of the single read samples were in fact EBV-.

This point reflects the fact that in the initial manuscript, it has been unclear what EBV-read count=1 is, and whether we truly measure a correlate/surrogate of EBV viral load. In the current version we now have solid evidence that the single-EBV-read individuals are true positives with low viral loads. We describe these lines of evidence in the introductory paragraph and additionally at other reviewer comments. In order to avoid redundancy, we would like to refer this reviewer to those answers.

4 -The results for the MHC region are extremely interesting and important, especially given the role of this region in autoimmune disease. It's unclear to me whether any non-classical HLA variants within the MHC were examined after accounting for the classical HLA associations, could the authors please clarify?

We apologize for this lack of clarity. No, we have not performed any examination of non-classical HLA variants within the MHC region. The main reason for that is that non-classical HLA alleles are currently not well captured by existing imputation methods. Also, the allelic diversity of non-classical HLA-alleles is lower than those of classical HLA alleles, and therefore best captured by SNPs in our analysis. Ideally, GRS analyses should include SNPs and HLA-alleles jointly, which would capture this diversity but still require some methodological developments. While we consider this out-of-scope of the present study, we have added short statement on this aspect in the Discussion & Methods sections:

Methods: "Non-classical HLA-alleles were not included due to the lack of established standards for imputing these alleles."

Discussion: "Finally, given the biological complexity of the MHC region and current challenges in HLA-allele imputation (PMID 41107551), some HLA-associations might have been missed, or mimicked by extended regions of LD."

5 -Regarding the imputation of HLA for the All of Us cohort, it would be helpful to see quality metrics, particularly as some of the associated alleles are rather rare. It's especially surprising that many of these are given at 4-field resolution. Was this reduced for analysis (e.g., to protein-coding resolution only)?

We agree with the reviewer that some more information regarding the HLA-alleles in the AoU cohort are required, in particular as the replication of HLA-alleles is a central finding of our study. We have revisited the Methods, where we have now added additional information, and also provide details in a **Supplementary Figure** (see below) and **Supplementary Table S6**. We have also performed additional changes and analyses. As a major change, we have

adapted the resolution of alleles in AoU to 2-field, as it is much more in agreement with UKB data. Additionally, we:

- compared the allele frequencies between HLA-alleles in UKB and in the European subcohort of AoU, which are highly concordant even for most rare alleles (see panel a)
- typed HLA-alleles in a subset of 900 individuals of AoU (150 individuals per ancestry, randomly sampled), and compared them to the imputed HLA-alleles. Correlations were calculated against the entire cohort (panel b) or by ancestry (panel c). Methodological details on how quality metrics were calculated are to be found in PMID 34611364.

As can be seen from the plots, the vast majority of HLA-alleles had $r^2 > 0.5$ (as far as typing data was available). We observed 12 HLA-alleles with $r^2 < 0.5$, all of which were rare (max. cohort allele frequency: 0.007). Two of them were among the 54 conditionally independent HLA-alleles (selected at positions 46 and 47 respectively), but had r^2 -values of 0.48.

Based on these data, we calculated correlation coefficients (r^2) and positive predictive values (PPVs). Our results reflect the current challenges associated with HLA-alleles, which are located in regions of high and extended LD and are therefore difficult to comprehensively characterize with current short-read methods.

In the manuscript, we provide the Figure above as part of the **Supplementary Note 12**. We also have added correlation coefficients / PPVs for all typed HLA-alleles in the new **Supplementary Table S6**. Finally, we also show the top-10 independent HLA-alleles from the conditional analysis, and their associations in both UKB and AoU, as **new panel b in Figure 2**.

6 - As currently written, it is not clear how the conditional analysis of HLA alleles was conducted. Please clarify the writing or add the relevant description to the Methods section.

Thanks for pointing this out - yes, this part was missing in the Methods section, we apologize for this oversight. We have now added a paragraph on the conditional analysis in the Methods section.

7 - Lines 215-220: Were all 54 independent HLA alleles that withstood conditional analysis testing identified in these comparisons?

After changing to a replication using two-field HLA-alleles, we observed some slight changes in the numbers: *“Of the 116 associated HLA-alleles, 106 were matched to HLA-alleles in AoU (Methods), and 100 of these showed nominal significance and a consistent effect direction in both datasets (Supplementary Table S6). For the 54 conditionally independent HLA-alleles, 46 were replicated among 52 that were available in AoU.”*

To make this information easier accessible to the readers, we have consolidated former **Supplementary Tables S5** (HLA and replication AoU) and **S6** (conditional analysis) into one **Supplementary Table S6**, which includes the unconditioned and conditioned analyses.

We have also included all r^2 -values that were available, into **Supplementary Table S6**. We would like to highlight that of the 116 genome-wide significant imputed HLA-alleles, HLA typing was available for 90, with an average r^2 -value of 0.88 (min: 0.33; max: 1).

8 - In the UKB GRS analysis, the conclusion for the HLA_MHC I-MS result described in the text does not seem to agree with the interpretation of Figure 4. Once HLA-A*02:01 is accounted for, the signal disappears. This is not surprising, since A*02:01 is a well-known protective marker for MS. However, Figure 4 only shows the significant association, without mention that it is entirely driven by A*02:01. Further, when DRB1*04:04 was accounted for in the RA model, there was no signal loss, suggesting that there may be additional/novel genetic factors driving the association. The authors might want to consider finding a way to distinguish between these two outcomes in Figure 4.

This is an excellent point. Following the revised analysis, we now see some slight changes in the P-values, including a nominally significant result for HLA_MHC I in MS when HLA-A*02:01 is excluded ($P=0.031$). Still, including HLA-A*02:01 yields a much more significant GRS association ($P=3.09 \times 10^{-5}$). We have modified the GRS panel in Figure 4 by adding a symbol to the MS-boxplot. This symbol is explained in the legend, stating that the result gets attenuated when HLA-A*02:01 is removed.

We also describe this in the Results section, as follows:

*“For MS, this effect was attenuated when HLA-A*02:01, which is associated with MS²⁶ and was among the most significant HLA-alleles in the GWAS, was excluded from the GRS ($P_{HLA_MHC} = 3.09 \times 10^{-5}$, $P_{excl\ HLA-A*02:01} = 0.031$).”*

Referee #3 (Remarks on code availability):

- The authors should be congratulated for their extremely thorough and comprehensive documentation of code used in this manuscript. It is very accessible and easy to follow. It includes informative descriptions and workbooks for interfacing with UKB and AoU. The context and commentary provided for each script are very helpful. Example data is provided. This is one of the more reproducible and well-documented workflows this reviewer has ever seen.

- My one comment would be just to go back through and double check that the input data files for each script are all properly described and/or have the correct repository location, with respect to the structure that will be hosted on Github. Most scripts do have this, but I found a few examples of some that may have been missed, see below:

- EBVread_host_control-0.1/Finemapping_Coloc/finemapping_coloc_forestplot.R:

```
o sumstats <-
```

```
read_tsv("C:/Users/srichter/Documents/Promotion/sumstats/zero_vs_1_18_sumstats.tsv.gz")
```

o Could not find this file. Please include this file or a description that demonstrates the formatting/data that it should contain

- EBVread_host_control-0.1/UKB-RAP_notebooks/2_2_coverage_plot/coverage_plot.R:

```
o per_b_coverage<-
```

```
read_tsv(file="/mnt/project/2_1_count_ebv_reads/per_base_coverage.tsv",  
col_names=c("CHR", "POS", "COV"))
```

o “mnt/project/” does not match the repository structure as provided

We thank the reviewer for their overall positive assessment of our code. The script *EBVread_host_control-0.1/Finemapping_Coloc/finemapping_coloc_forestplot.R* indeed had indeed lacked documentation regarding the parameters that potentially need modification. We therefore now modified this script. Paths starting with /mnt/project/ within the UKB-RAP scripts point to files that are stored within the project folder at DNAnexus. This is due to the analysis environment (JupyterLab) which mounts the project folder at /mnt/project/. To clarify this potentially confusing folder structure, we added a note to the README file of the UKB scripts.

Referee #4 (Remarks to the Author):

I co-reviewed this manuscript with one of the reviewers who provided the listed reports.

Referee #5 (Remarks to the Author):

Schmidt et al describe a novel study investigating genetic and non-genetic factors that contribute to latent EBV infection control, by undertaking large-scale analyses of the UK

Biobank and All of US. The authors provide their rationale for this study, namely that inefficient control of latent EBV infection is linked to the development of EBV-associated diseases, including EBV-associated malignancies, however there is a lack of data that can be used to study latent infection, and BioBanks often miss the viral load data. Some studies include serological data, but this only helps to identify previous infections or current lytic infections; it is difficult to use for latency. Instead, the authors use short-reads mapping to the EBV genome as surrogate measurements for increased viral load in latent infections. Overall, the study is novel and robust, and the paper is well-written, providing context both for the rationale of analyses and for the results. However there are still some major points that need to be addressed, please see below:

Detection of EBV-reads using GS data from UKB and AoU

1) The authors use the presence of EBVread+ as a proxy of latent EBV infection. The question is how they can distinguish between latent infection and a primary or lytic infection in the cohort. I would assume this was addressed when the authors excluded immunocompromised individuals (who are more likely to have a lytic infection) and excluded outliers with a higher number of EBV read+. However, the authors should explain this more explicitly.

This comment raises two important points: first, the double meaning of the term “latency”, and second, the question of which phase of the viral life cycle we actually measure. These comments are in line with similar points raised by multiple reviewers. We provide a detailed answer to both questions in the introductory paragraph of this rebuttal letter, and at prior instances in our replies to the other reviewers. While we would like to refer the reviewer to these points, we here summarize the main conclusions:

- 1) wording: Please see the introductory paragraph. Briefly, we had used the term “latent infection” to describe all processes after primary infection - without referring to the viral life cycle, so without discriminating between latent infection and lytic reactivation (and independent of any virus production). We have realized that this has caused confusion, so we have renamed the “latent infection” to “persistent EBV infection” across the entire manuscript, including the title.
- 2) correlation of viral reads with viral life cycle: In the response to comment 4 of reviewer 2, we address this comment in depth. Briefly, we now experimentally show that EBV-reads are a surrogate measure for EBV-DNA load in blood cells. It is established that EBV (and so EBV-DNA) hides in memory B cells in a latent stage, which is compatible with results obtained using paired GS / transcriptome data. Using EBV serology data as an orthogonal measure, we were able to show that EBV-reads, and therefore EBV-DNA load in blood cells, are also correlated with lytic activity to some degree. This is also in agreement with prior literature, including the germinal center concept of EBV infection (PMID 26424647).

2) Line 107 & Figure 1: The authors use serology data to corroborate EBV sequencing read detection. The EBV antibodies used, if these are the ones shown in extended Figure 2 (EA-D and ZEBRA are both acute phase and VCA depends on whether it is IgM or IGg, which is not specified), would not show non-controlled latent EBV infection, rather lytic/acute infection. How do the authors explain the use of all antibodies and what their analysis means. Maybe a sub-analysis would help to tease this out.

Before going into details, we would like to clarify how the serology data used in our study was obtained. The serology data was previously generated in a subcohort of UKB (PMID 35383168) and comprises measurements of four individual antibodies (ZEBRA, EA-D, EBNA1 and VCA-p18), all of which were IgG. According to the information provided in accompanying documents (<https://biobank.ctsu.ox.ac.uk/crystal/crystal/docs/infdisese.pdf>), "MFI thresholds for seropositivity for each antigen was determined by examining percentile plots and cut-offs were chosen to optimize sensitivity and specificity, based on the validation work. Based on the quantitative properties of the data, MFI values can also be analysed in categories (e.g. tertiles) among the sero-positives to evaluate dose-response relationships."

We now use these antibody measures for two purposes: 1) to validate that EBV-reads are specific to sero+ individuals and 2) to obtain information on the serological correlates that underlie our GS-based EBV measure. As point 2 has been mentioned by several reviewers, we provide an in-depth reply to this point as part of the introductory paragraph. However, we also briefly summarize those results here.

For 1), we used the previously established binary definition of EBV-seropositivity, which relied on elevated levels of at least two of the four individual antibodies, as suggested by PMID 35383168 and in UKB ("EBV seropositivity defined as positive if: two or more of antigen VCA p18 > 250, antigen EBNA-1 > 250, antigen ZEBRA > 100, antigen EA-D > 100 [unit: MFI]"). When assessing the overall sensitivity and specificity of EBVread+, as illustrated in **Figure 1d**, this provided support for the fact that our measure is highly specific. However, we agree with the reviewer that the information on the specific, individual antibodies is much more informative to understand what is captured by EBVread+. For the rest of the manuscript, we have therefore considered each antibody separately.

For 2), we have analyzed each of the four IgG measurements individually and compared their MFI values to the individual EBV-read counts. Specifically, we took those individuals in the *UKB-EUR*-cohort that had serology data available, and who were sero+ according to the definition above (n=7,338 individuals). We then compared the GS-based EBV-read counts with quantitative values for the four individual antibodies (IgG ZEBRA, EA-D, EBNA-1 and VCA-p18). For this we grouped MFI values into deciles, and calculated the fraction of EBVread+ individuals, overall and across different bins of EBVread counts. We also calculated mean MFI values per decile. The results are shown in the introductory paragraph and as **new Extended Data Figure 5** in the manuscript.

The strongest correlation of EBV+ status is observed with IgG against VCA-p18, which is indicative of prior EBV infection (both, primary infection and lytic reactivation) and completed virus production. Given the age distribution of the UKB participants, primary infection is unlikely. As VCA-p18 IgG-antibodies are (i) markers of past infections (24175209), which (ii) can be boosted by recurrent (or repeated) antigen contact (18556771), the correlation of EBVread+ with high IgG antibody levels against VCA-p18 might reflect ongoing lytic activity during persistent infection. Significantly increased IgG titers against VCA-p18 have also been observed in individuals with increased EBV-DNA in blood in independent, though small, studies (PMIDs 11953465, 35921373).

The association with IgG VCA-p18 is also observed at the genetic level, as the genome-wide significant loci are strongest correlated with quantitative IgG levels against VCA-p18 (now as **panel d** in main **Figure 2**, and **Extended Data Figure 6**). For more information, we kindly refer to point 2 in the initial paragraph.

Based on this data, we propose a model in which EBV-reads mainly stem from memory B cells that are latently infected with EBV, and a small pool of reactivating cells. The *in vivo* processes that underlie the presence of reactivating cells in peripheral blood, and their extent, are currently unknown, but the recent observation of B cells in latency II-III as well as lytic stages in peripheral blood of individuals with SLE (PMID 41223250) provides support that lytic reactivation in peripheral blood cells indeed exists. Additionally, EBVread+ is correlated with serological patterns that suggest lytic reactivation, a fact that is in line with models of the EBV life cycle that predict more reactivations when the pool of latently infected B cells increases (PMID 26424647). Together, these results now provide novel impulses that should prompt respective follow-up studies.

Non-genetic factors contributing to EBVread+

3) Line 150: “In contrast, increasing age, GS yield...” It is expected that the higher the genome sequencing yield is, the higher the chance of detecting EBV reads. The authors should not really use read counts for EBV positivity, but rather reads per million (RPM) to normalise for the differing sequencing depths in the cohorts.

In both UKB and AoU, the distribution of EBV-read counts was zero-inflated and, thereby, cannot be well modelled as an outcome of regression models available in standard GWAS-frameworks. We therefore decided to binarize EBV-reads for the main analysis and classify individuals with an EBV-read count of 0 as controls, and individuals with EBV-read counts ≥ 1 as cases, which allowed the use of logistic regression. In this binary scenario, a change from GS yield to RPM would not have changed the attribution of case-control status: individuals with 0 reads also have 0 RPM (and remain controls), and all individuals with non-zero reads have >0 RPM (and remain cases). As all our main analyses in the manuscript refer to this definition (cases: EBV-read counts=1-18 reads, controls 0 reads), we have not changed our definition of EBVread+. Still, differences in GS yield are a predictor of the case-control status, and it therefore remains reasonable to include GS yield as a covariate in the logistic regression analysis to avoid potential confounding (see e.g. Chapter 4.4.6 Control of Confounding in Vittinghoff, Eric, et al. Regression methods in biostatistics: linear, logistic, survival, and repeated measures models. New York, NY: Springer New York, 2005.).

However, the reviewer’s concern gets relevant when quantitative analyses on EBV-read count distribution are performed. This was not the scope of our study for the reasons described above, but might become relevant in future analyses. We therefore took a closer look into the relation between GS and RPM:

First, we plotted the distribution of individual RPM-values in each of the EBV-read count groups. As can be seen in the boxplots below, the RPM values are well grouped with the EBV-read count groups, with only limited overlap (see below; picture represent EBV-read count groups from 1-4 (left panel) for a better visibility, and an overall picture for EBV-read count groups up to 10 reads (middle panel)).

Second, we tested how the attribution of EBV-read counts would change quantitatively if we had used RPM. We observed that the usage of RPM instead of absolute EBV-read counts would have resulted in 1.1% of all samples with counts ≥ 2 having a lower RPM value than the highest RPM value among samples with EBV-read count=1. In the right panel we

illustrate this by plotting the correlation between a sample's rank (with random assignment of ranks in case of ties) based on absolute read counts (n_rank) and its rank when based on the yield-normalized RPM value (rpm_rank; for non-zero EBV read counts only; see plot below). This data suggests that for quantitative analyses, the usage of RPM might be more accurate, but the risk of wrong attribution of samples through the usage of GS yield is generally low.

4) Line 151-153: I don't follow the seasonality effect on EBVread+. Are the authors suggesting that viral load (in latent infections) is higher during winter? If so, is this an effect of other seasonal diseases (i.e. flu)? If it were the effect of other infections reactivating the virus, it would again point to lytic infections rather than latent. Could the authors please comment.

In both UKB and AoU, we robustly observe that more individuals are EBVread+ during winter. As the detection of EBV-reads reflects elevated viral DNA load in blood cells, this relates to the fact that more EBV-genomes are present in samples measured in winter as opposed to summer. While the reasons for this observation are currently unknown, we suggest two plausible hypotheses. First, as suggested by the reviewer, an increased rate of seasonal infections during winter, such as respiratory viruses, as well as increased environmental stress, might trigger more memory B cells to go into lytic reactivation, which is supported by literature (PMID 38281067). Second, it could also be due to differences in leukocyte composition of the blood, which has been shown to undergo a seasonality effect on immune cell populations, with all tested B cell subpopulations (incl memory B cells) showing significantly higher levels in winter (PMID: 27818087).

We would like to mention that the JCTF cohort, which was established from individuals infected with SARS-CoV-2, also had a substantially higher EBVread+ rate (39.2%) than expected based on cohort characteristics. As this data provides some first support for the seasonal infection model, we have added this as a statement to the Results section:

“Though this seasonality effect requires further investigation, a plausible hypothesis is that seasonal infections during winter, such as co-infections with respiratory viruses, might drive EBVread+. This would be consistent with observations of a higher rate of EBVread+ in the JCTF, whose participants had been infected with SARS-CoV-2 around the time of sampling (39.2% EBV-read+; Supplementary Note 5).”

We have also added a Supplementary Note 5 describing additional analyses that ensured that none of the other results for which the JCTF cohort was used, were confounded by severity of the SARS-CoV-2 co-infection.

Identification of common genetic variants associated with EBVread+

5) Line 189: “To ensure the robustness of our results...” The authors here show that irrespective of whether the “EBV positive case” is defined by just one read or more reads, the effect sizes of the EBVread+ GWAS lead variants correlate well. Could the authors repeat this (ie EBVread+=1 compared to EBVread+>=2) for other analyses as well, to show that by including the singleton reads, they are not interpreting low level contamination as EBV positivity?

Thank you for the suggestion. The concern on “what actually is the EBV-read count=1 group” has been raised by all other reviewers as well. We have therefore very rigorously looked into this and now provide additional evidence by five different analyses, showing that this group is not contamination, but rather low-level EBV viral load. These multiple lines of evidence are detailed in the introductory paragraph, but we summarize the main aspects here:

- read=1 coverage distribution matches overall coverage distribution,
- read=1 allelic variation distribution matches the allelic distribution in EBVread count samples of =>2 and differ from clearly contaminated plates
- qPCR results from a clinically validated assay confirm higher viral load in the group of EBVread=1 individuals when compared to those with EBVread=0
- transcriptome analyses show a higher fraction of EBV-transcript read counts in samples from EBVread=1 individuals compared to those with EBVread=0.
- GWAS using read=0 vs. read=1 yields similar results and effect estimates as analyses including higher EBV-read counts

As we feel that we provide robust support for the hypothesis of EBV-read count=1 individuals being true positives with low viral loads, we did not perform the suggested comparisons of different groups for all other downstream analyses.

6) Line 197-199: “In contrast, only weak evidence was generated for correlations of effect sizes to EBV sero-positivity (Fig 2c)” In the legend of figure 2 the authors state that seropositivity was defined as having 2 /4 antibodies exceeding thresholds, however as above it would be best to stratify depending on whether the antibodies are indicative of lytic or previous (latent) infections.

This important question has been addressed as part of the comments 1&2 for this reviewer, and also as part of the introductory paragraph. In order to avoid redundancy, we would, at this point, like to refer to those answers above. With the new analyses described above, we now have the data at hand to emphasize this distinction much more. Also, we would like to highlight that we have added the correlations of the EBV-read counts with MFI values for each of the four antibodies as **Extended Data Figure 5**, and the genetic correlations of the

genome-wide significant loci in **Extended Data Figure 6** (with raw data in **Supplementary Table S11**).

7) Lines 215-217: In the genetic study, the discovery cohort was UKB, and the AoU was used to replicate the results. In the previous section (non-genetic factors), the authors start with analysis of the AoU cohort and then UKB. I find this a bit confusing – is there a rationale for this?

Our rationale was to use one data set (e.g. UKB or AoU) for “discovery” analyses and the other data set to “replicate” the findings. This is expected to reduce overfitting and potentially avoid statistical phenomena, such as winners' curse, and an overall higher robustness of findings. Thanks to the reviewer we have realized that this was confusing. We now provide a study overview as **Supplementary Note 1** and additionally explain this at the respective sections in the text. Also, whenever possible, we have added the prefixes “UKB” or “AoU” to the names of the cohort in which the respective analyses was undertaken.

[Redacted text]

[Redacted figure]

Annotation and fine-mapping of associated non-MHM loci

8) Line 268-270: Some specific techniques need a bit more context, particularly for the large audience of a broad-theme journal. Just as an example: “Fine-mapping”, “credible sets of variants” might be common in statistical genetics, but they are not well known in other fields (i.e. EBV virology). I find this section very difficult to follow: in Supplementary Table S4 I can see 28 variants, not sets of variants?

We apologize for expecting too much domain-knowledge in some parts of the manuscript. When all comments were addressed, we went over the entire paper again, with the aim to identify such paragraphs and to reduce their complexity. Where possible, we generated a more high-level text in the main manuscript, while providing more detailed information as Supplementary Note. Also, we partially restructured entire sections. While we cannot list all modified sections here individually, we hope that these changes will have increased the overall readability.

Gene-based analyses suggest novel genes for involvement in host-EBV interaction

9) Line 290 : “.. we observed 83 genes classified as IEI” I think this is quite an interesting finding and I am wondering whether the authors can check from the Biobank and All of us data, whether these immune deficiencies are also associated with impaired control of other herpesviruses (CMV?) or in general, increased susceptibility to infections (ie other pathogens). In short, is this an EBV-specific phenomenon, or are the individuals that are more likely to not control well the EBV latent infection, overall more susceptible to other infections?

10) Also are these IEI genes involved in EBV-related cancers?

We thank the reviewer for these two questions, which we would like to answer jointly.

While our method of determining infections from GS reads can generally be extended to other viruses, a direct application of the pipeline using the current settings lacks specificity and sensitivity to reliably generate this information from AoU or UKB. Furthermore, in clinical practice, particular vulnerability to CMV is not assessed as routinely in patients with inborn errors of immunity unless this becomes clinically severe, or if/when the patients undergo allogeneic stem cell transplantation, where CMV infection is monitored as part of routine patient care.

We therefore approached the two points from a different perspective: we checked the 83 genes identified in our analysis against the phenotypes observed in patients carrying these IEIs. Specifically, we investigated in detail the 9 genes (for which monogenic germline defects causing IEIs are known) which we identified with a test-wide significant enrichment of common variants, as we consider them the strongest candidates for being causal genes involved in control of EBV viral load. It became apparent that germline lesions in some of these genes go along with particular susceptibility to severe EBV infections as published in recent studies, and are often associated with markedly increased risk of EBV-associated complications, in particular, lymphoma. As these results are important, we have summarized them in a new panel of **Figure 3** (replacing the *ERAP2* haplotype panel):

We also provide a detailed description (including the references) as **Supplementary Note 7:**

Among the nine IEI genes with test-wide significance in the MAGMA gene analysis, we found four to be associated with IEIs that have marked and well-documented susceptibility to EBV infection or chronic/ clinically more severe manifestations of EBV infection, including *IKZF3/Aiolos* deficiency (PMID 34155405), *NFKB1* deficiency (PMID 32278790), *CTLA4* deficiency (PMIDs 25213377, 25329329, 29729943) and, most profoundly, *CD70* deficiency (PMID 28011864, 28011863, 32603431). These four IEIs are also associated with increased susceptibility for lymphoma, often EBV-driven. This is in line with the clinically accepted notion that inability to clear an active EBV infection is associated with significant risk of lymphoma development. Regarding the risk for other infections, this is well documented for two genes, while it has not yet been shown for the two others (though it is plausible based on the pathomechanisms of the associated deficiencies).

For three additional genes, it is at least plausible that they would also go along with clinically more profound EBV infection: *DCLRE1B*, in which loss-of-function mutations cause a dyskeratosis congenita phenotype with bone marrow failure including B- and NK-cell lymphopenia as well as T-cell dysfunction (PMID 35007328), *IKZF1/Ikaros* (described as dominant-negative or gain-of-function mutations, respectively (PMIDs 29889099, 35333544, 2698193), and *IRF1* (whose deficiency that causes NK cell lymphopenia and decreased type 1 dendritic cells (PMID 36736301). Similarly, it seems plausible that these genes are involved in increased risk for both other herpes infections and associated cancers, though this has not yet been documented.

The IEIs caused by the two genes *CCR2* (PMID 38157855) and *ICOS* (PMID 12577056) have not yet been reported with any relation to EBV.

Polygenic architecture of EBVread+ and EBV-associated diseases

11) Line 364: “to IM, HL, MS, RA, non-Hodgkin-Lymphoma, SLE, and Sjogren disease”
How were these EBV-associated diseases chosen (aside IM, lymphomas and MS). Also is it not the case that these diseases are associated, for the most part, with an active EBV infection?

Thanks for pinpointing this lack of clarity. For the initial analyses we had chosen a set of seven diseases for which there is strong evidence that a prior infection with EBV (generally measured based on being EBVsero+) is a major risk factor, if not even causal (as for MS and IM). This selection was based on the current state of knowledge as summarized in PMID 36113467. At this point, the reasoning was to provide evidence for a biological relevance of the GS-based EBV measure in disease etiology.

In order to also identify potential diseases for which an association with prior EBV infection is less established (largely due to the lack of systematic assessment of EBV serology, and potentially the limited correlation between EBV serology and EBV viral load), we then performed a systematic phenome-wide association study (PheWAS), identifying IBD, hypothyroidism and T1D as additional diseases which display overlapping genetic background with EBV viral load.

We would like to highlight that our study generates novel hypotheses regarding a potential role of persistent EBV infection in those diseases, which require further follow up. In order to

more clearly state the reason for initial selection, we have added a short statement and include the reference at this point in the manuscript.

Two-Sample Mendelian Randomization (2SMR)

12) Line 445-446: “significant heterogeneity of effects was observed across SNPs, and the causal effects for these outcomes were driven by SNPs in the MHC region” .

I think it is problematic that the observed effects of EBVread+ on RA and T1D were driven by MHC variants. This violates core MR assumptions since MHC/HLA variants directly affect autoimmune disease risk independent of viral control mechanisms, suggesting shared genetic susceptibility rather than causal relations. Also, the evidence suggesting an association between EBV and T1D is weak.

We thank the reviewer for this excellent comment, referring to a prerequisite for Mendelian Randomization: ensuring that the core assumptions of MR are true. As correctly pointed out, this is challenging for any analysis involving HLA-alleles, due to their generally presumed pleiotropy.

While we think that we had already noted caution regarding the interpretation of the 2SMR results in the first version, we have now revisited the entire section. Major changes, and aspects we would like to stress, include:

- We had restricted the instrumental variables to variants that are associated with the exposure (EBVread+) at genome-wide significance
- In our sensitivity analyses we only found evidence for horizontal pleiotropy for T1D, using a set of SNPs that included some MHC SNPs, but also other variants outside of MHC.
- We had applied six different estimators which included two pleiotropy-robust estimators (MR-RAPS, MRPRESSO), and only report those traits for which results are consistent across all six.

Based on those results we conclude that there is some evidence for potential causal effects of EBVread+ (as exposure) on the outcomes RA and on T1D, however, results might as well be explained by a shared genetic architecture, e.g., through horizontal pleiotropy or a genetically-influenced common cause of EBVread+ and RA/T1D. We now present the results as an exploratory analysis and have toned down the claims in our manuscript. Also, we have removed the statement on causality from the Abstract and provide all estimators and sensitivity analyses in **Supplementary Tables S20 and S21**.

Discussion

13) The authors should also include amongst the limitations that the cohorts used in this study are heavily skewed towards European ancestry populations. This is relevant 1) because of the association with HLA, which varies dramatically across populations; 2) EBV-associated diseases prevalence varies a lot worldwide; 3) EBV viral strains are different across different populations.

We would like to point out that for most of the AoU analyses (in particular overall EBV-read+ rates, non-genetic factors and GRS), we had also included the non-European individuals of

AoU. Still, the reviewer is right regarding the fact that these diverse cohorts were not included for discovery analyses, but replication (largely due to the fact that AoU was only used as replication for the genetic findings from UKB, which is majorly European). We therefore now extend this important aspect in our discussion section:

“Third, despite the partial transferability of HLA-based GRS across ancestries, the results of the discovery analyses were mostly generated in European individuals. This might have influenced the identity of associated HLA-alleles, and may limit the generalizability of the findings with respect to different EBV strains and EBV-associated diseases, which vary in terms of global distribution and prevalence. Replication of the GWAS findings and downstream analyses in non-European ancestries are required.”

Minor comments

Methods

14) Lines 749&772: why did the authors use different criteria for mapping the EBV and HHV7 reads ie soft clipped bases and length of alignment?

For each of the two viruses, aggregated reads were graphically displayed using IGV. Based on visual inspection, the parameters were first chosen to avoid unspecific mapping of EBV. Thereafter, the same filter criteria were applied to HHV7, but additional unspecific mapping remained. Therefore, filters for sequencing reads were set to stricter criteria for HHV-7 as compared to EBV. We have now added this information in **Supplementary Note 6**, where HHV7 detection is described.

15) Line 748&749: “Only reads where both, forward and reverse, reads mapped to the EBV genome were retained.” How can there be EBVread+=1, if both mates in a pair must map to EBV to be retained? Do the authors mean one EBV fragment rather than read?

In our study we used the term “EBVread” as a term for a mate pair (i.e., both forward and reverse reads of one pair). We understand that this is confusing, but feel that this term best captures that GS-based measure: an “EBVread” as referred to in our study means “one sequenced fragment” which is composed of “one read pair”.

To make this clear and avoid confusion, we now make this more explicit in the Methods:

*“Only read-pairs where both forward and reverse reads, respectively, mapped to NC_007605.1, were retained. Within pairs, reads were removed if they had more than 20 soft-clip bases, less than 120 bases matching the reference or were duplicates (see **Supplementary Note 2**). Finally, if at least one read of a read-pair remained, this was counted as one “EBVread”.”*

Referee #5 (Remarks on code availability):

I reviewed the attachments provided with the code. I did not attempt to install and run the analyses. The code seems well organised, with different directories for different analyses and some documentation available.

Thank you.

Referee #6 (Remarks to the Author):

I co-reviewed this manuscript with one of the reviewers who provided the listed reports.

Point-by-point rebuttal letter for reviewer comments:

Referee #1 (Remarks to the Author):

I thank the authors for their detailed responses to my previous review. The authors have done substantial additional work to address my major concerns and, on the whole, am satisfied with their responses.

Thank you.

My one additional concern surrounds the validation studies of the EBV reads = 1 phenotype. Although the majority of their new analyses support the conclusion that these are true positives, the qPCR validation studies show surprisingly high rates of positivity in the EBV=0 samples (figure 1 panels e & f). In particular, the validation-2 cohort shows essentially equal proportions of positive replicates in the EBV=0, 1 and 2 read-count groups. At this stage, I don't know how much more work can be done to address any potential mis-classification, and I certainly don't want to delay publication further, but I do feel the authors need to address this issue further in discussion and describe how this may influence their results.

We have added a statement on this aspect and its implications, in the Discussion. The paragraph reads:

„First, due to the standard depth of human GS, most individuals had an EBV-read count of zero, and many had an EBV-read count of exactly 1. For statistical analyses, we binarized the phenotype into low/high EBV viral load, based on absolute EBV-read count numbers, and compared EBV-read count 0 vs. 1 and higher. Given the limited resolution, some individuals with presumed high viral load might actually have low viral load. However, this potential mis-classification is unlikely to have impacted the overall conclusions, which are supported by our sensitivity analyses and are similar to those of a recent study, which used a different definition for increased viral load (Nyeo et al. 2026).“

Otherwise, I am very impressed with the quality of this manuscript and the additional experiments/analysis.

Thank you.

Referee #1 (Remarks on code availability):

I did not review and revised code

Referee #2 (Remarks to the Author):

The authors have done an excellent job in responding to previous reviewer comments and in revising the manuscript. The revised study is a technically strong and innovative correlation analysis of large health research databases with detection of EBV DNA in genome sequencing from blood samples. The authors demonstrate the feasibility, validity and utility of this type of study, providing a path forward for analysis of multiple other viral and genetic biomarkers.

While the technical aspects of the study are well justified, one could argue that the ultimate conclusion is that EBV viral loads are elevated in individuals that are either immunosuppressed (e.g. HIV or steroids), or have some type of immune-inflammatory or autoimmune disorder. These conclusions are somewhat expected given what is already known about EBV biology. While the methodological approach to large data analysis is technically innovative and exemplary, the biomedical insights may not be very impactful since it is already known that EBV control is affected by other immune disrupting events, including immunosuppressive agents, other viral infection and autoimmune-inflammatory disease.

Nevertheless, the study is among the first to demonstrate the analysis of GWAS data sets can be explored for EBV and other biomarkers to correlate with previously unappreciated disease connections.

Thank you.

Very minor: Line 369-370. Seems to be a fragment sentence “..54 tissues available in GTEx V8,.”

Thank you - we have deleted the fragmented sentence (as it was not required).

Referee #3 (Remarks to the Author):

The authors have done an admirable job in addressing reviewer concerns, including substantial new analyses. All critiques have been adequately addressed and I have no additional concerns.

Thank you.

Referee #3 (Remarks on code availability):

All code concerns have now been addressed.

Thank you.

Referee #4 (Remarks to the Author):

I co-reviewed this manuscript with one of the reviewers who provided the listed reports.

Referee #4 (Remarks on code availability):

All comments were satisfactorily addressed.

Thank you.

Referee #6 (Remarks to the Author):

I co-reviewed this manuscript with one of the reviewers who provided the listed reports.

Referee #6 (Remarks on code availability):

I reviewed the code, but did not attempt to re-run the analysis. The code looks clear, and the folder includes README files and instructions.

Thank you.